# Investigation of the relationship between the spatial gradient of total electron content (TEC) between two nearby stations and the occurrence of ionospheric irregularities

Teshome Dugassa[1,2], John Bosco Habarulema[3,4], and Melessew Nigussie[5]

[1]Ethiopian Space Science and Technology Institute, Department of Space Science and Application, Addis Ababa, Ethiopia
[2]Bule Hora University, College of Natural and Computational Science, Department of Physics, Bule Hora, Ethiopia
[3]South Africa National Space Agency, Space Science, Hermanus, South Africa
[4]Department of Physics and Electronics, Rhodes University, Grahamstown, South Africa
[5]Washera Geospace and Radar Science Laboratory, Physics Department, Bahir Dar University, Bahir Dar, Ethiopia

**Correspondence:** Teshome Dugassa (tdugassa2016@gmail.com), John Bosco Habarulema (jhabarulema@sansa.org.za), Melessew Nigussie (melessewnigussie@yahoo.com)

**Abstract.** The relation between the occurrence of ionospheric irregularities and the spatial gradient of total electron content (TEC) derived from two nearby located stations (ASAB:4.34°N, 114.39°E and DEBK: 3.71°N, 109.34°E, geomagnetic), located within the equatorial region, over Ethiopia, during the post-sunset hours was investigated. In this study, the Global Positioning System (GPS) based derived TEC during the year 2014 obtained from the two stations were employed to inves-

5 tigate the relationship between the gradient of TEC and occurrence of ionospheric irregularities. The spatial gradient of TEC ($\Delta TEC/\Delta lon$) and its standard deviation over 15 min, $\sigma(\Delta TEC/\Delta lon)$, were used in this study. The rate of change of TEC derived indices ($ROTI$, $ROTI_{ave}$) were also utilized. Our results revealed that most of the maximum enhancement/reduction in $\Delta TEC/\Delta lon$ are noticeable during the time period between 19:00 LT and 24:00 LT hours. In some cases, the peak values in the spatial gradient of TEC are also observed during daytime and post-midnight hours. The intensity level of $\sigma(\Delta TEC/\Delta lon)$

observed after post-sunset show similar trends with $ROTI_{ave}$, and was stronger/weaker during equinoctial/solstice months. The observed enhancement of $\sigma(\Delta TEC/\Delta lon)$ in the equinoctial season shows an equinoctial asymmetry where the March equinox was greater than September equinox. During post-sunset period, the relation between the spatial gradient of TEC obtained from two nearby located Global Navigation Satellite System (GNSS) receivers and equatorial electric field (EEF) was observed. The variation in the gradient of TEC and $ROTI_{ave}$ observed during the evening time period show similar trends

with EEF with a delay of about 1-2 hrs between them. The relationship between $\sigma(\Delta TEC/\Delta lon)$ and $ROTI_{ave}$ correlate linearly with correlation coefficient of C = 0.7975 and C = 0.7915 over ASAB and DEBK, respectively. The majority of the maximum enhancement/reduction in the spatial gradient of TEC observed during the evening time period may be associated with ionospheric irregularities/equatorial plasma bubbles. In addition to latitudinal gradients, the longitudinal gradient of TEC has a significant contribution to the TEC fluctuations.

**Key words**: Spatial gradient of TEC, $ROTI_{ave}$, ionospheric irregularities

# 1 Introduction

The ionosphere, which consists of free electrons and ions, frequently experiences irregular electron density. After sunset, the ionospheric plasma interchange instabilities present in the equatorial/low-latitude ionosphere generate large-scale depletions in the ambient electron density which leads to the formation of plasma density irregularities that affect radio communication and navigation system (Basu and Basu, 1981). The generation of the plasma irregularities can be related to the decrease in plasma production immediately after sunset and the fast recombination rate in the E-region ionosphere, which results in a steep electron density gradient. The large enhancement of F-region vertical plasma drift in the evening hours due to the presence of enhanced eastward electric field is a critical driver which control the generation of plasma density irregularities (Fejer, 1991; Fejer et al., 2008). This pre-reversal enhancement (PRE) vertical plasma drift moves the F-region to higher altitudes (Abdu et al., 2009). When the altitude of F-region is high enough to overcome recombination effects, the Rayleigh-Taylor (R-T) instability mechanism initiates growth in plasma fluctuations. The R-T instability is considered primarily responsible for the generation of ionospheric plasma density irregularities or plasma bubbles in the equatorial and low-latitude region (Rao et al., 2006a; Fejer et al., 1999). Kelley (2009) reported that the existence of equatorial plasma bubbles (EPBs) is attributed to the instability of the R-T plasma which is triggered by the intensification of the eastward equatorial electric field just before its reversal. The characteristics of ionospheric scintillation and ionospheric irregularities over the equatorial and low-latitude region in different longitudinal sectors during different solar and geomagnetic activities have been studied (e.g., Burke et al., 2004; Paznukhov et al., 2012; Oladipo and Schuler, 2013a; Seba and Tsegaye, 2015). Various instruments such as all-sky imager (Wiens et al., 2006), and very high-frequency radar observation (e.g., Otsuka et al., 2009; Ajith et al., 2016) have been used to study the behavior of ionospheric irregularities and related scintillations. Recently, Global Navigation Satellite System (GNSS) signal analysis is an important tool to study the behavior of ionospheric irregularities (e.g., Pi et al., 1997; Nishioka et al., 2008; Watthanasangmechai et al., 2016; Magdaleno et al., 2017) because of its growing application in civilian and military applications.

The inhomogeneity of ionospheric electron distribution can cause sudden, rapid and irregular fluctuations of the amplitude and phase of the received signals, known as ionospheric scintillation (Wernik and Liu, 1974). This inhomogeneity, i.e, spatial plasma density/TEC gradient, varies significantly at low-latitude region because of geomagnetic storms, equatorial spread F (ESF) and Appleton anomaly. As the GNSS signals pass through the ionosphere, the ionospheric irregularities also cause the delay of signals. The classification of the spatial electron density/TEC gradients can be given as latitudinal (north-south) and longitudinal (east-west) (Jakowski et al., 2004). It is normally found that the spatial plasma density gradients can be represented by means of TEC changes per latitude or longitude (TECU/deg) or by their changes in distance (TECU/km). In addition to causing an integrity threat for life-safety application to air traffic management (Luo et al., 2002; Lee et al., 2011; Rungraengwajiake et al., 2015), the ionospheric TEC gradient is also unfavorable from communication, and surveillance system which depends on trans-ionospheric signal propagation (Foster, 2000). Radicella et al. (2004) and Nava et al. (2007) also presented the contribution of the horizontal gradients of vertical TEC to positioning error. The characteristics of horizontal ionospheric density gradients and their effects on trans-ionospheric radio wave propagation have been studied at different

latitudes (Jakowski et al., 2005; Radicella et al., 2004). It has been reported that the majority of large/steep TEC gradients are associated with equatorial plasma bubbles (Pradipta and Doherty, 2016). Rao et al. (2006a) estimated ionospheric spatial gradient from F-region peak electron density (NmF2) data using a chain of radio soundings. Based on the GNSS data acquired by the dense distribution of receivers over Brazilian longitude sector, Cesaroni et al. (2015) highlight the relationship between intensity and variability of TEC gradients and the occurrence of ionospheric scintillation.

Previous studies have attempted to explain the relationship between the latitudinal (N-S) gradient of TEC surrounding the anomaly region and ionospheric scintillation over different sectors (Mendillo et al., 2001; Valladares et al., 2001; Rao et al., 2006b; Ray et al., 2006; Muella et al., 2008). Mendillo et al. (2001) pointed out that equatorial ionization anomaly (EIA) strength at sunset is the best available precursor for pre-midnight ESF. Using latitudinal distribution of TEC measurements at about 20:00 LT, Valladares et al. (2001) observed a high crest-to-trough ratio prevalent to ESF days. Recently, Seba et al. (2018) investigated the relationship between equatorial ionization anomaly and night time ESF over East Africa longitudinal sector using data from ground-based Global Positioning System (GPS) stations and a horizontal meridional neutral wind model. To identify signals which severely suffer from the ionospheric gradient, Ravi Chandra et al. (2009) and Rungraengwajiake et al. (2015) used rate of change of TEC (ROT) and rate of change of TEC index (ROTI). From the definition, however, ROTI mixes both the spatial and temporal gradients of TEC variations. To show the relation between EIA and ESF, Seba et al. (2018) used ROTI and crest-to-trough ratio. Even though, the characteristics of ionospheric irregularities/plasma bubbles over equatorial/low-latitude region of Africa under different solar and geomagnetic activities was discussed (Seba and Tsegaye, 2015; Seba and Nigussie, 2016; Mungufeni et al., 2016; Kassa and Damtie, 2017; Olwendo et al., 2018; Bolaji et al., 2019; Dugassa et al., 2019), a limited number of studies have been carried out over the region on the relationship between TEC gradient and the occurrence of ionospheric irregularities.

Gradient of electron density is one of the candidates which affects the growth rate of the R-T instability and hence the generation of ionospheric irregularities (Sultan, 1996). The magnitude of Pederson conductivity which can be estimated using electron density also has an effect on the growth rate. Over the African region, however, there is a lack of instrumentation (radar, ionosondes and/or incoherent scatter) to directly measure the gradient of electron density and investigate its relationship with occurrence of ionospheric irregularities. It is generally known that TEC is the integral of electron density, and the spatial gradient of TEC between close-by stations would help us to examine its relation with ionospheric irregularity at those stations. Investigating the relationship between the spatial gradient of TEC and the occurrence of ionospheric irregularity using ground GPS-TEC receiver from two nearby located stations is the aim of the current study. In this work, closely located GPS stations will also help to study the relationship between the gradient of TEC and electric field during post-sunset. The relation between the daytime eastward equatorial electric field derived from the equatorial electric field (EEF) model and the daytime equatorial electrojet (EEJ) obtained from ground-based magnetometer measurements were also discussed. The study is the first of its kind in the African sector to present the relation between the spatial gradient of TEC and the occurrence of ionospheric irregularities. The gradients of plasma density might be considered as an important parameter in the modeling of ionospheric irregularities and mitigating positioning errors on GNSS based applications.

## 2 Data and analysis method

The GNSS data used for this study were obtained from University NAVSTAR Consortium (UNAVCO) database (http://www.unavco.org/). We used data from two receiver stations located in the East African region at Debark (Geog. Lat. 13° N, Geog. Long. 37.65° E, Geomag Lat. 4.13° N) and Asab (Geog. Lat. 13° N, Geog. Long. 42.65° E, Geomag. Lat. 4.85° N) for the period 2014. The Receiver-Independent Exchange (RINEX) observation files obtained from the IGS website were processed by the GPS-TEC application software developed at Boston College (Seemala and Valladares, 2011). The TEC analysis software uses the phase and code values for both L1 and L2 GPS frequencies to eliminate the effect of clock errors and tropospheric water vapor to calculate relative values of slant TEC (Sardón and Zarraoa, 1997; Arikan et al., 2008). In order to avoid the multipath effects, different authors have used observation data above certain cutoff mask ranging from 15° to 35° (Chu et al., 2005; Mushini and Pokhotelov, 2011). In the current study, an elevation cutoff mask of 30° was used for all the VTEC computed. Table 1 gives the list of all the stations for which data has been used in this study.

There are two independent ways of estimating the TEC gradient values using ground-based GPS receiver data (e.g., Lee et al., 2007, 2010). The first method uses a pair of closely-spaced receiver stations, looking at the same GPS satellite to calculate the difference in TEC values between the two neighboring ionospheric piercing points (IPP) at any given time. The second method uses a single GPS receiver station to infer the spatial TEC gradient values based on the observed temporal rate of change in TEC. In the current study, we have applied the first method to obtain the spatial gradient of TEC. Using the computed VTEC determined from the two receiver stations, the spatial gradient of TEC (difference of TEC between two stations per longitudinal separation) was computed for every time and then we analyzed its diurnal, monthly and seasonal variations. The two stations are located nearly along the same geographic latitude with longitudinal separation of about $\sim 5°$ or corresponding spatial separation of 535.7 km. Stations with the same latitude were selected to examine only the contribution of the longitudinal gradient of TEC to the generation of ionospheric irregularities expressed by ROTI. In the competition of the spatial gradient of TEC, the 1 minute VTEC values for all satellites in view were averaged. The spatial gradient of TEC utilized in this study was computed using Eq. (1) (Lee et al., 2007; Ravi Chandra et al., 2009; Cesaroni et al., 2015).

$$Spatial\ gradient\ of\ TEC(t_i) = \frac{VTEC_{asab}(t_i) - VTEC_{debk}(t_i)}{\Delta lon} = \frac{\Delta VTEC}{\Delta lon} \tag{1}$$

where i = 1 to 1440, and $\Delta lon$ represents the difference in the longitude between the two stations. To detect the presence of plasma density irregularities, using ground and space-based TEC/plasma density measurements different authors presented different techniques. For example, Pi et al. (1997) applied the standard deviation of ROT over 5 min to characterize the occurrence of ionospheric irregularities. For in situ electron density measurements, Su et al. (2006) and Huang (2018) used the standard deviation of ion density variations over 10 s as indicator of ESF occurrence. Cesaroni et al. (2015) used the standard deviation of gradient of TEC to indicate the variability of the gradients. In this study, we used the standard deviation of spatial gradient of TEC over 15 min, $\sigma(\Delta TEC/\Delta lon)$, to examine the relationship between spatial gradient (zonal or E-W gradients) of TEC and occurrence of ionospheric irregularities.

The time variation of TEC also known as rate of change of TEC (ROT and ROTI), which is a measure of large-scale ionospheric irregularities (Aarons et al., 1997) were used in this study. These indices are a good proxy for the phase fluctuation, and can be used to characterize all the known features of equatorial spread F (ESF) (Mendillo et al., 2000). The rate of change of TEC (ROT) is given by

$$ROT = \frac{TEC_k^i - TEC_{k-1}^i}{t_k^i - t_{k-1}^i} \tag{2}$$

where $i$ is the visible satellite and $k$ is the time of epoch and ROT is in units of TECU/min. The ROTI is defined as the standard deviation of ROT over a 5-min period and mathematically given by Eq. (3) (Pi et al., 1997; Bhattacharyya et al., 2000; Nishioka et al., 2008). Usually, ROTI > 0.5 TECU/min indicates the presence of ionospheric irregularities at scale lengths of a few kilometers (Ma and Maruyama, 2006).

$$ROTI = \sqrt{\langle ROT^2 \rangle - \langle ROT \rangle^2} \tag{3}$$

Oladipo and Schuler (2013b) employed the idea of Mendillo et al. (2000) to obtain a new index called $ROTI_{ave}$ index given in Eq. (4). $ROTI_{ave}$ index is the average of ROTI over 30 min interval for a satellite and then averaged over all satellites in view. The index gives the average level of irregularities over half an hour. Recently, $ROTI_{ave}$ has been applied to demonstrate and explain the level of ionospheric irregularities over the low-latitude/equatorial region of Africa (Oladipo et al., 2014; Bolaji et al., 2019; Dugassa et al., 2019). In this study, the rate of TEC fluctuation index (ROTI) and $ROTI_{ave}$ (Pi et al., 1997; Oladipo and Schuler, 2013b; Oladipo et al., 2014) were used to observe the occurrence of ionospheric irregularities.

$$ROTI_{ave}(0.5hr) = \frac{1}{N} \sum_{n=1}^{N} \sum_{i=1}^{k} \frac{ROTI(n, 0.5hr, i)}{k} \tag{4}$$

where $n$ is the satellite number, $0.5hr$ is half an hour (0, 0.5, 1,... 23.5, 24 UT), $i$ is the 5 min section within half an hour (i = 1, 2, 3, 4, 5, 6), $N$ is the number of satellites observed within half an hour and $k$ is the number of $ROTI$ values available within half an hour for a particular satellite. According to Oladipo and Schuler (2013b), the value of $ROTI_{ave} < 0.4$, $0.4 < ROTI_{ave} < 0.8$ and $ROTI_{ave} > 0.8$, respectively represents the background fluctuation, existence of phase fluctuation, and severe phase fluctuation activities. These threshold values were used to observe the relation between the occurrence of ionospheric irregularities and the spatial gradient of TEC.

The magnetic data used in this study are obtained from International Real-Time Magnetic Observatory Network (INTER-MAGNET) and Africa-Meridian B-field Education and Research (AMBER) magnetometers installed in Addis Ababa (AAE, 9.0° N, 38.8° E, 0.2° N, geomagnetic) and Adigrat (ETHI, 14.3° N, 39.5° E, 6.0° N, geomagnetic), respectively. Both of the

instruments provide one-minute values of the northward (X), eastward (Y), vertical (Z) components of the Earth's magnetic field, from where the horizontal component (H) is computed using Eq. (5).

$$H = \sqrt{X^2 + Y^2} \tag{5}$$

To avoid different offset values of different magnetometers, the nighttime baseline values in the H component (Eq. 6) are first obtained for each day and subtracted from the corresponding magnetometer data sets to obtain the hourly departure of H denoted $\delta H$ expressed by Eq. (7). The baseline value was defined as the average of the H component night time (23:00 - 02:00 LT) value of the Earth's magnetic field.

$$H_o = \frac{H_{23} + H_{24} + H_{01} + H_{02}}{4} \tag{6}$$

where $H_{23}$, $H_{24}$, $H_{01}$, and $H_{02}$ are respectively the hourly values of H at 23:00, 24:00, 01:00 and 02:00 in local time (LT).

$$\delta H(t) = H(t) - H_o \tag{7}$$

where t is the time in hours ranging from 01:00 to 24:00 LT. The hourly departure $\delta H$ is then corrected for the non-cyclic variation using (Eq. 8). This correction was proposed previously by Rastogi et al. (2004) who defined non-cyclic variation as a phenomenon in which the value at 01:00 LT is different from that of local midnight (24:00 LT).

$$\Delta c = \frac{\delta H_{01} - \delta H_{24}}{23} \tag{8}$$

The hourly departure of H ($\delta H$) corrected for the non-cyclic variation corresponding to the magnetometer data set gives the solar quiet variation (Sq) values as shown in Eq. (9):

$$Sq(t) = \delta H(t) + (t - 1) * \Delta c \tag{9}$$

where t = 1 to 1440.

The equatorial electrojet current (EEJ) produces a strong enhancement in the H-component magnetic field measured by magnetometers located within $\pm 5°$ of the magnetic equator. Measurements of this magnetic field perturbation in equatorial magnetometers could provide a direct measure of the daytime equatorial electrojet (EEJ) and have strong relationships with dayside vertical velocity ($\boldsymbol{E} \times \boldsymbol{B}$ drift) (Anderson et al., 2004, 2006; Yizengaw et al., 2012). The equatorial stations respond primarily to the EEJ and also to the ring current and the global quiet time Sq current system. However, ground magnetometers just outside the extent of the EEJ ($\sim 6° - 9°$, off the dip equator) exhibit exact response to the ring and Sq currents, but near-zero response to the EEJ. To obtain the contribution of H-component field to the EEJ current, we subtract the H-component

value recorded at the off the equator ($\sim 6° - 9°$ geomagnetic) from H-component value measured at the magnetic equator, using Eq. (10). The subtraction has been made to remove the contribution of the ring current and global Sq dynamo from the H-component.

$$\Delta H = \delta H_{AAE} - \delta H_{ETHI} \tag{10}$$

where $\delta H_{AAE}$ and $\delta H_{ETHI}$, respectively, show the hourly departure of H over Addis Ababa and Adigrat, .

**Table 1.** Location information and the type of data used in this study.

| Name of stations | Code | Geo. lon | Geo. lat | Geom. lon | Geom. lat | Data |
|---|---|---|---|---|---|---|
| Asab, Eritrea | ASAB | $42.65°$ E | $13°$ N | $114.34°$ E | $4.85°$ N | GPS-TEC |
| Debark, Ethiopia | DEBK | $37.65°$ E | $13°$ N | $109.24°$ E | $4.13°$ N | GPS-TEC |
| Addis Ababa, Ethiopia | AAE | $38.77°$ E | $9.04°$ N | $110.47°$ E | $0.18°$ N | Magnetometer |
| Adigrat, Ethiopia | ETHI | $39.46°$ E | $14.28°$ N | $111.06°$ E | $5.80°$ N | Magnetometer |

    The other data source used in this study is the Real-time model of the Ionospheric Electric Fields (http://geomag.org/models/ PPEFM/RealtimeEF.html). The Prompt Penetration Electric Field Model (PPEFM) (Manoj and Maus, 2012) is a transfer function model which models the daily variations coming from the solar wind, which are mapped in the interplanetary electric field (IEF) data. Eight years IEF data from the ACE satellite, radar data from Jicamarca Unattended Long-Term studies of the

Ionosphere and Atmosphere (JULIA) system, and magnetometer data from the CHAMP satellite have been used to derive the transfer function. By using the real-time data from the ACE satellite, the transfer function models the current variations in the equatorial ionosphere. To calculate the best estimates of equatorial electric field, the model takes time and location as input parameters. The model outputs provide the electric field generated as a result of the convective electric field, quiet time electric field, and both. In the present study, we have used the background quiet-time electric field to examine the relation between the

equatorial electric field (EEF) and the spatial gradient of TEC derived from the two nearby stations. To have a rough idea about the PPEFs during storm days, we have also used the quiet-time and penetration electric field.

    The real-time model of electric field has been used for different case studies over different sectors to observe the influence of prompt penetration electric field (PPEF) on the variations of total electron content and the occurrence of ionospheric irregularities (Nayak et al., 2017; Dugassa et al., 2019). However, this model has not been applied yet to explain the electrodynamic

phenomena over African low-latitude region. To use the PPEF model in this region, we presented its relationship with equatorial electrojet (EEJ), an indicator of the eastward electric field, during the daytime period over the equatorial region of Africa based on ground-based magnetometer measurements (Rastogi and Klobuchar, 1990; Anderson et al., 2002; Yizengaw et al., 2014). It has been reported that the strength of EEJ before sunset has a correlation with the generation of ESF during nighttime

period preceded by a rise in the F-region (Dabas et al., 2003; Uemoto et al., 2010; Ram et al., 2007). The relation between the EEF obtained from the real-time electric field model and ΔH was determined. The EEF derived from the real-time electric field model was used in this study to explain the influence of the equatorial electric field on nighttime variations of the spatial gradient of TEC and the occurrence of ionospheric irregularities. The temporal resolution of EEF was 5 min and that of $\Delta H$ was 1 min. To make their resolution consistent, 5 min average of $\Delta H$ of each of the selected quiet days were computed. In this study, ΔH derived from the H-component of the geomagnetic field of the two stations during quiet days of the year 2012 were used. In this year, we have a large number of magnetometer measurements relative to other years. From each month of the year 2012, the five quiet international days (total of 38) obtained from (http://wdc.kugi.kyoto-u.ac.jp/qddays/index.html) were selected to show the correlation between $\Delta H$ and EEF. Since the EEJ is a daytime phenomenon, only the daytime values of EEF and $\Delta H$ during (07:00 to 17:00 LT) were examined.

## 3    Results and Discussions

### 3.1    Relation between the day-time Equatorial Electrojet (EEJ) and Equatorial Electric Field Model (EEFM)

Figure 1a presents the diurnal variation of EEF and EEJ current signature of H-component of geomagnetic field on 26 March 2012. As can be seen in Figure 1a, during the daytime period (07:00 - 17:00 LT), the ΔH and EEF show similar trends. The relationship between the strength of daytime EEJ derived from ΔH and EEF obtained from equatorial electric field model is shown in Figure 1b. To show the performance of the EEF model over the East African sector, we have presented the relationship between ΔH and EEF for five (5) international quiet days of each month of the year 2012. As depicted in Figure 1b, during the daytime period, the ΔH correlates positively and linearly with EEF with the correlation coefficient, C = 0.60. Manju et al. (2012) obtained an excellent agreement with observations at the Indian and South American sectors. Different techniques have been utilized to estimate the ionospheric electric field (e.g., Hysell and Burcham, 2000; Anderson et al., 2002; Alken et al., 2013; Dubazane and Habarulema, 2018). Anderson et al. (2002) proposed ΔH deduced from ground-based magnetometers as a proxy of equatorial electrojet current. They reported that the vertical plasma drifts observed from Jicamarca incoherent scatter radar (ISR) has a positive and linear relation with ΔH and henceforth the ΔH was widely taken as a proxy substitute for the EEF. Anderson et al. (2006) and Yizengaw et al. (2011) also reported a strong relation between the dayside vertical velocity ($\boldsymbol{E} \times \boldsymbol{B}$ drift) and $\Delta H$. Alken et al. (2013), on the other hand, estimates the EEF using CHAMP satellite derived latitudinal current profiles of daytime EEJ along with $\Delta H$ measurements from ground magnetometer stations and they showed that pair of magnetomer stations one located near magnetic equator and the other at off-equator capture the day-to-day strength of the EEJ.

The daytime eastward equatorial electric field in the ionospheric E-region plays an important role in equatorial ionospheric dynamics. It is responsible for driving the EEJ current system, equatorial vertical ion drifts, and the equatorial ionization anomaly. The EEJ is a strong ionospheric current along the magnetic equator driven by the dayside eastward electric field. Studies also show that the daytime electrodynamics play a decisive role in the initiation of post-sunset ESF (e.g., Mendillo

et al., 2001; Valladares et al., 2001, 2004). The connection between the occurrence of ESF during the evening sector preceded by the rapid rise in F-layer and the strength of EEJ before sunset has been presented (Dabas et al., 2003; Burke et al., 2004; Kelley, 2009; Uemoto et al., 2010; Ram et al., 2007). Sreeja et al. (2009) reported observational evidence for the plausible linkage between the daytime EEJ related electric field variations with the post-sunset F-region electrodynamics. Furthermore, Hajra et al. (2012) indicate that the afternoon/evening time variation of the eastward electric field as revealed through EEJ seems to play a dominant role in dictating post-sunset resurgence of EIA and consequent generation of spread-F irregularities. Since the equatorial electric field derived model (EEF) correlate moderately with $\Delta H$ over East Africa longitudinal sector, we could use the real-time EEF model over the equatorial/low-latitude region of Africa to explain some special features of ionospheric phenomena like plasma density irregularities and the positive/negative spatial gradient of TEC between the two stations.

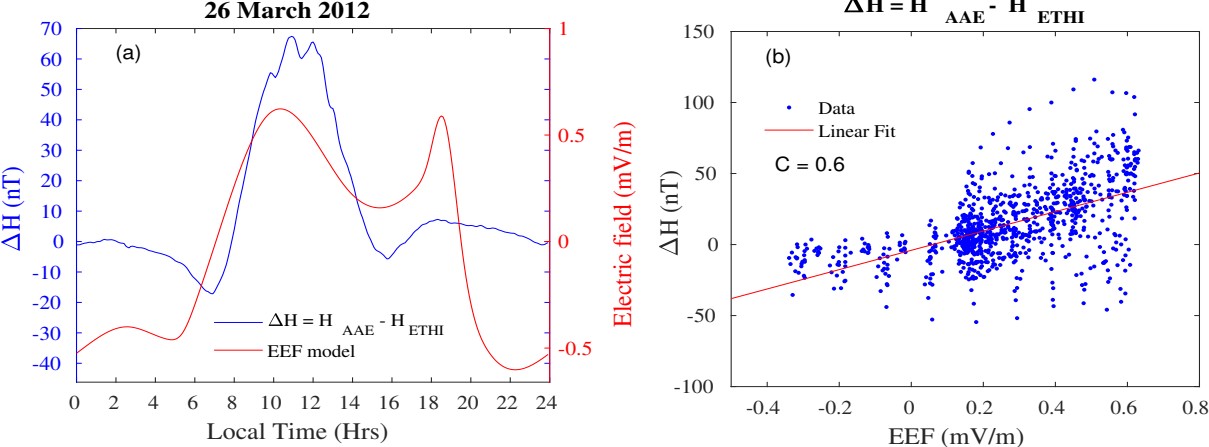

**Figure 1.** (a) Example showing the diurnal variation of Equatorial Electric field (EEF) Model (red curve) and $\Delta H$ (blue curve) during 26 March 2012 and (b) The correlation between the equatorial electrojet (EEJ) and quiet-time equatorial electric field model (EEF) during day-time (07:00-17:00 LT) period for quiet days of year 2012. The red line shows the linear fit of data points.

## 3.2 Relation between the equatorial electric field (EEF), the spatial gradient of TEC and occurrence of ionospheric irregularity

Figure 2 (a-l) shows the diurnal variation of the quiet-monthly mean of the spatial gradient of TEC (blue curves) and EEF (along $\sim 40°$ E) (red curves) in the year 2014. We superimpose EEF and spatial gradient in TEC to observe the effect of EEF on the variability of the gradient in TEC and/or occurrence of ionospheric irregularities. In the computation of the spatial gradient of TEC (using Eq. 1), negative/positive values in the gradient of TEC may be observed during the nighttime or daytime. Both the negative and positive differences of TEC between the two stations show the gradient of TEC. A positive/negative gradient in TEC denotes an enhancement/reduction in TEC or electron density over ASAB relative to DEBK. Gradient in TEC is positive when TEC over ASAB is greater and is negative when TEC over DEBK is greater. This difference may be attributed to different

physical processes, like neutral winds and plasma drift. In this study, the term maximum enhancement/reduction in the gradient of TEC (in terms of magnitude) were used when the nighttime value of gradient of TEC was larger than the daytime value. There were also cases when the gradient in TEC during the daytime was greater than nighttime values. It can be seen from Figure 2 (red curves) that around evening hours, enhancement in EEF were observed in the equinoctial months and was relatively

weak during the June and December solstices. It has been stated that an enhanced eastward electric field will be produced from the electrodynamical interaction of the eastward thermospheric wind with the geomagnetic field around the dip equator at the sunset terminator when longitudinal gradient conductivity exists between the high-conducting dayside ionosphere and the low-conducting night side ionosphere (Batista et al., 1986; Heelis et al., 1974). Most of the enhancement/reduction in the TEC gradient was observed in the pre-midnight (19:00 - 24:00 LT) and postmidnight (24:00 - 06:00 LT) but after 1-2 hr of the

post-sunset enhancement of the equatorial electric field. During the nighttime period, the maximum enhancement/reduction in the spatial gradient of TEC was found mostly in the range between 5.0 TECU/deg and -5.0 TECU/deg. A variation in the spatial gradient of TEC observed in the pre-midnight may be due to the plasma bubbles (Ratnam et al., 2018). In some days, the spatial gradient of TEC observed during the daytime was relatively small compared to the evening time hours. The maximum enhancement/reduction in the gradient of TEC and the peak in the EEF observed during the pre-midnight period

was significant during the equinoctial months. After post-sunset period, the maximum enhancement/reduction in the gradient of TEC in solstice months was small compared to equinoctial months, when PRE electric field observed in the evening period was minimum. Yoshihara et al. (2005) confirmed the larger ionospheric gradients during summer and followed by autumn. The ionospheric gradients are less during winter as compared to summer and autumn. The enhancement/reduction in the gradient of TEC observed in the evening period could be related to the PRE in zonal electric field.

Figures 3 (a - d) shows the diurnal variation of the spatial gradient of TEC (blue curve) and $ROTI_{ave}$ (red and black curves) over ASAB and DEBK stations. The $ROTI_{ave}$ values in each panel was greater than 0.4 TECU/min, a threshold value showing the presence of irregularities in the pre-midnight hours. Likewise, maximum enhancement/reduction in the gradient of TEC was observed during the pre-midnight and post-midnight periods. It is evident from Figure 3 during post sunset period that the pattern of ROTI (observed in both stations) and the spatial gradient in TEC show a kind of similar trend. Different researchers

used the concept of ionosphere spatial gradient based on multi-GNSS observations within a small scale region to provide corrections and integrity information to the Ground-Based Augmentation System (GBAS) (Rungraengwajiake et al., 2015; Saito and Yoshihara, 2017). They attribute the large ionosphere spatial gradient to the TEC enhancements and the ionosphere irregularities. Saito and Yoshihara (2017) associated spatial gradient in ionospheric TEC with plasma bubbles. Rungraengwajiake et al. (2015) analyzed plasma bubbles at postsunset equinox time and observed the higher scales in east-west gradient

compared with north-south gradients for GBAS system, however, Cesaroni et al. (2015) reported that the north-south gradient in TEC correlates well with ionospheric scintillation than the east-west gradient of TEC. The plasma density variability, either the spatial and/or temporal, causes not only the GNSS-based positioning error but also radio wave scintillation.

The maximum enhancement/reduction in the gradient of TEC and the associated ionospheric irregularity during the post-sunset period can be explained by ionospheric electrodynamics. It is well known that Earth's equatorial ionosphere presents

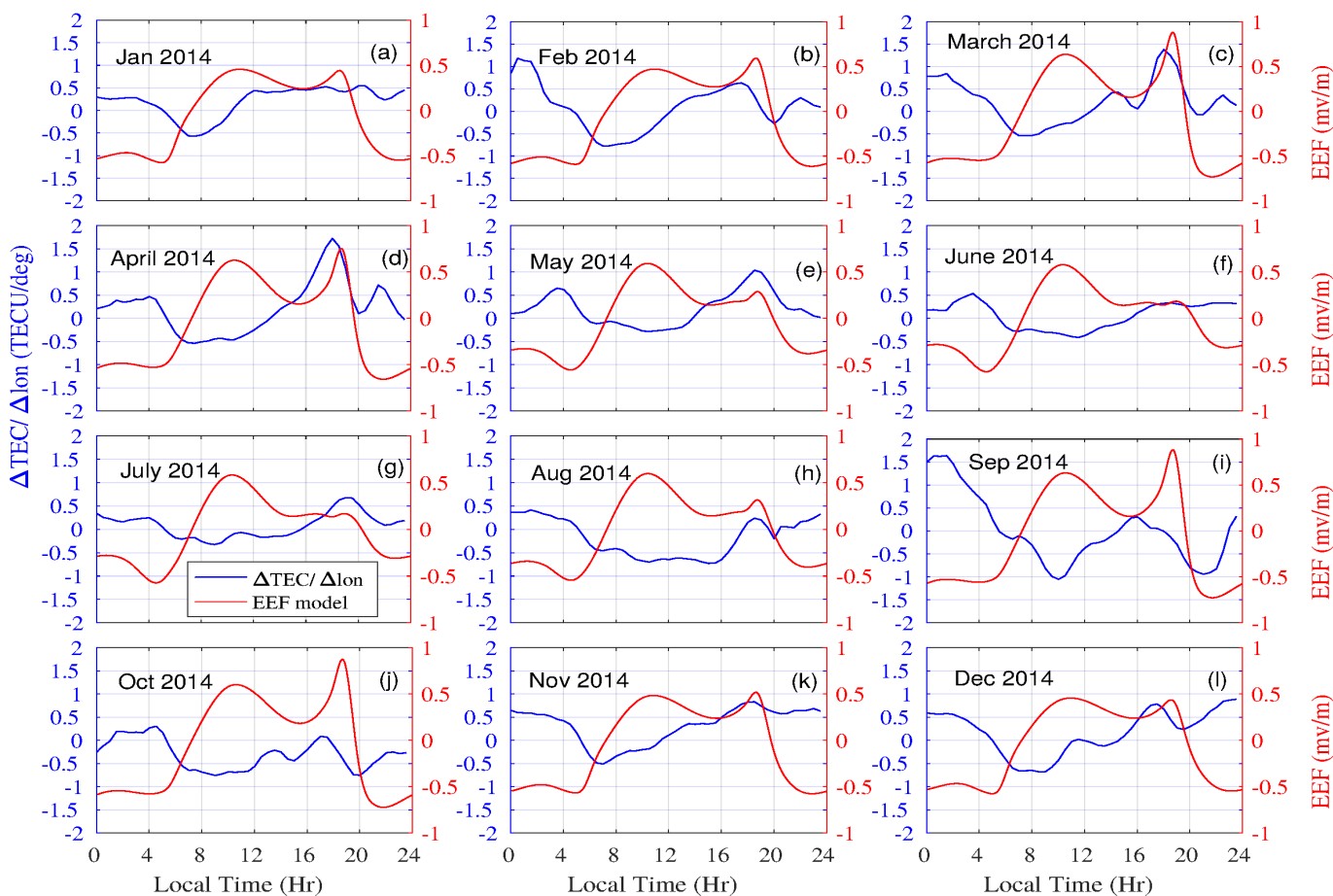

**Figure 2.** Comparison of Quiet-Monthly Mean of EEF derived from real-time electric field model at about ($\sim 40°$ E) and spatial gradient of TEC between ASAB and DEBK in the year 2014.

temporal and spatial variations. The electrodynamics of low-latitude ionosphere after sunset is influenced by F-region dynamo which is governed by a longitudinal gradient of the electrical conductivity and thermospheric zonal wind (Crain et al., 1993). Anderson et al. (2004) showed that the scintillation activity is related to the maximum $\boldsymbol{E} \times \boldsymbol{B}$ drift velocity between 18:30 LT and 19:00 LT. Mendillo et al. (2001) have pointed out that the best available precursor for the pre-midnight equatorial spread F (ESF) is the equatorial ionization anomaly (EIA) strength at sunset, which is in turn influenced by the magnitude of PRE. Using differential TEC profiles, TEC (at 18:00 hr) - TEC (at 20:00 hr), Valladares et al. (2004) explained that the PRE of the vertical drift would re-energize the fountain effect. The postsunset EIA produces a large plasma density gradient from the trough region to the crest region. Takahashi et al. (2016) observed the steepest TEC gradient with a difference of 30-50 TECU from the inside to outside plasma bubbles.

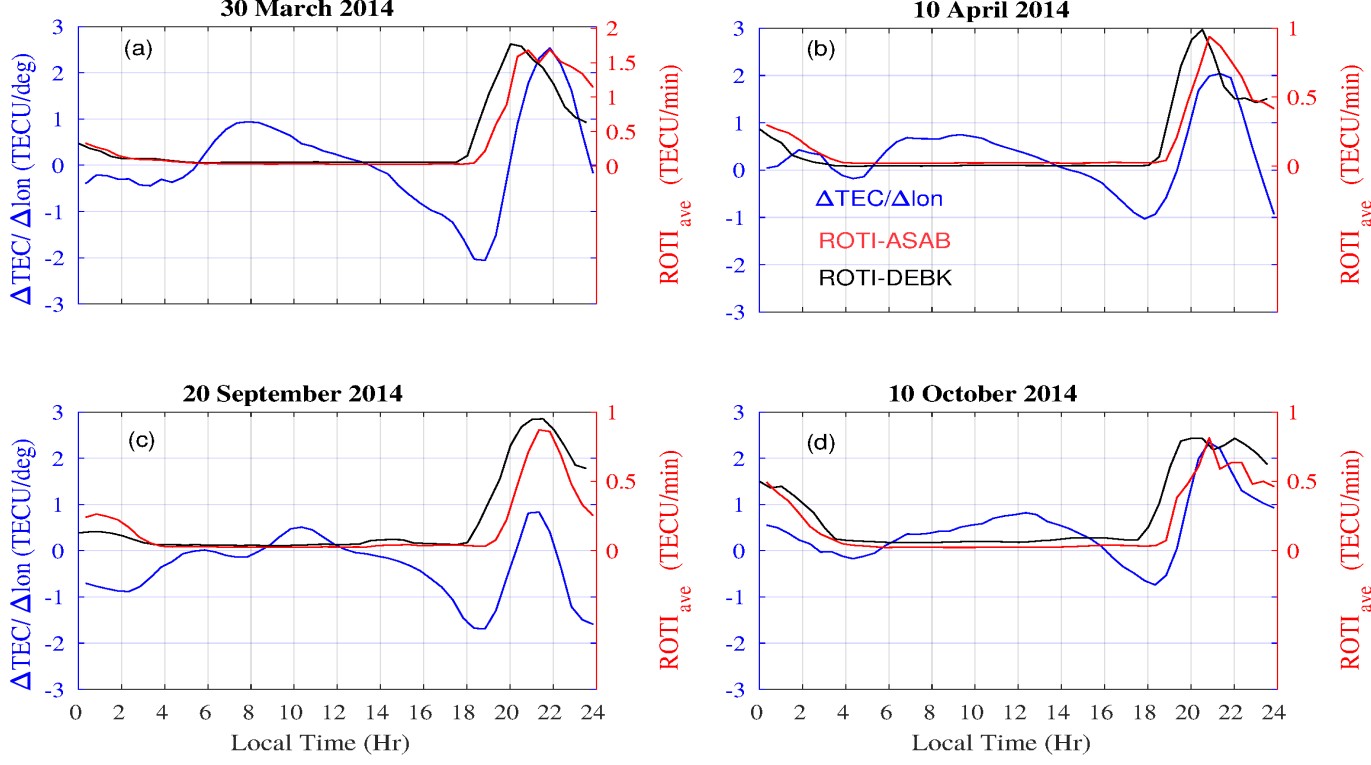

**Figure 3.** Typical examples of diurnal variation in the spatial gradient of TEC (blue curve) and the ROTI over ASAB (red curve) and DEBK (black curves) on a) 30 March 2014, b) 10 April 2014, c) 20 September 2014 and d) 10 October 2014.

In the evening sectors, the vertical drift enhancement is of particular significance as it is the major drivers for the generation of ESF (Farley et al., 1970; Woodman, 1970; Basu et al., 1996; Fejer et al., 1999; Martinis et al., 2005). Tulasi Ram et al. (2006) reported that the rapid enhancement of post-sunset of the zonal electric field leads to a large vertical plasma drift ($\boldsymbol{E} \times \boldsymbol{B}$), thereby lifting the F-layer to higher altitudes resulting in a condition conducive for the generation of ESF. Ionospheric
5   irregularities are mostly observed over equatorial/low-latitude region an hour or two hours after the PRE. Rastogi and Woodman (1978) showed that ESF can appear at any time of the night other than the post-sunset period following the abnormal reversal of the vertical F-region drifts to an upward direction, with a delay of about 1-2 hr. As illustrated in Figure 2, the influence of post-sunset enhancement in the zonal electric field on the maximum enhancement/reduction in the spatial gradient of TEC during the post-sunset period can be seen. This could indicate the maximum enhancement/reduction in the spatial gradient of
10   TEC and the occurrence of ionospheric irregularities have some degree of relationship.

Figures 4 (a-h) indicate representative cases showing the diurnal variation of $ROTI_{ave}$ over ASAB (red curve) and DEBK (black curve) and $\sigma(\Delta TEC/\Delta lon)$ (blue curve) during the occurrence of ionospheric irregularities (left panel) and during their absence (right panel). It is clearly observed from Figs. 4 (a-d) that intensity level of $ROTI_{ave}$ was greater than 0.4 TECU/min indicating the presence of ionospheric irregularities. Figures. 4 (e-h), on the other hand, indicate examples when the presence

of occurrence of ionospheric irregularities are absent, where the value of $ROTI_{ave}$ was less than 0.4 TECU/min. $ROTI_{ave} \geq$ 0.4 TECU/min indicates the presence of ionospheric irregularities (Oladipo and Schuler, 2013b; Oladipo et al., 2014). As can be seen from Figure 4 (blue curves), the intensity level of $\sigma(\Delta TEC/\Delta lon)$ observed in the evening period was higher when irregularities were present (Fig. 4, left panels) than when irregularities were absent (Fig. 4, right panels). It is evident from the

figures that the strength of $\sigma(\Delta TEC/\Delta lon)$ observed on the nighttime period was greater than the daytime value, as $ROTI_{ave}$ does. The post-sunset plasma bubble irregularities are generated at the bottom side of the F-layer by the sunset enhancement of the zonal electric field called pre-reversal enhancement caused by the combined action of an eastward thermospheric wind and the longitudinal gradient in ionospheric conductivity that exist along sunset terminator (Rishbeth, 1971; Fejer et al., 1999). It is well documented that plasma bubble development depends on the linear growth rate for generalized R-T instability process, the

flux tube integrated Pedersen conductivity that controls the non-linear development, and density perturbations that are needed to act as a seed to trigger the instability growth. The PRE is generated through interaction of zonal neutral wind in the F-region and conductivity gradient caused by the terminator. The density gradient affects the R-T instability growth rate thus, the generation of irregularities (Ossakow, 1981; Mendillo et al., 1992). Background electron density and its distribution in the ionosphere affects the formation of ionospheric irregularities. The intensity level of $\sigma(\Delta TEC/\Delta lon)$ obtained from two nearby located

stations observed near sunset terminator, related with the longitudinal gradient of ionospheric conductivity, could indicate the presence/absence of large-scale ionospheric irregularities. Cesaroni et al. (2015) reported a strong relationship between the standard deviation of the gradient of TEC and the occurrence of ionospheric scintillation.

Figure 5 (a-d) illustrates typical examples of the diurnal variation of $\sigma(\Delta TEC/\Delta lon)$ and $ROTI_{ave}$ during some of selected geomagnetic storm periods. During the study period days with the disturbance storm time index (Dst) $\leq -50$ nT value

(http://wdc.kugi.kyoto-u.ac.jp/) were considered. To reflect the effect of storms (during the study period) on the variation of ROTI and spatial gradient, four sample storm periods: (a) 17-21 February 2014, (b) 10-14 April 2014, (c) 25-29 August 2014 and (d) 10-14 September 2014 were selected. These storm days are categorized as moderate (-100 nT $\leq$ Dst $\leq$-50 nT) and strong magnetic storms ($Dst \leq -100nT$) (Loewe and Prölss, 1997; Echer et al., 2013). Figure 5 (iv) and (v), shows the temporal variation of the Dst and z-component of interplanetary magnetic field (IMF Bz) during the storm periods. The storm

observed during 17-21 February was highly complex and had multiple main and recovery phases resulting from a series of Earth-directed Coronal Mass ejections (CMEs) (see, Ghamry et al., 2016, for details). The intense storm of February 17 started with sudden storm commencement (SSC) at 09:00 UT. The storm event was characterized by southward turning of IMF Bz with a magnitude of 13 nT and a depressed Dst index of minimum value -120 nT at 09:00 UT on 19 February which was followed by the recovery phase on February 21. The storm event during 10-14 April 2014 is moderate with maximum negative

excursion of Dst ∼-87 nT at 10:00 UT on 12 April, corresponding to a negative (southern) IMF Bz ∼ -8.5 nT. On 12 April 2014, a change in polarity of z-component of IMF from southward to northward at about ∼ 20:00 UT was observed. Maximum negative excursion of Dst ∼ -79 nT at 19:00 UT and IMF Bz ∼ -13 nT at 15:00 UT was observed during storm event 26-29 August 2014. During 10-14 September 2014 storm period, whereas, minimum Dst about ∼-88 nT at 00:00 UT and IMF Bz ∼ -11 nT at 22:00 UT was observed. On 12 September 2014, the z-component of IMF turned southward at about 20:00 UT.

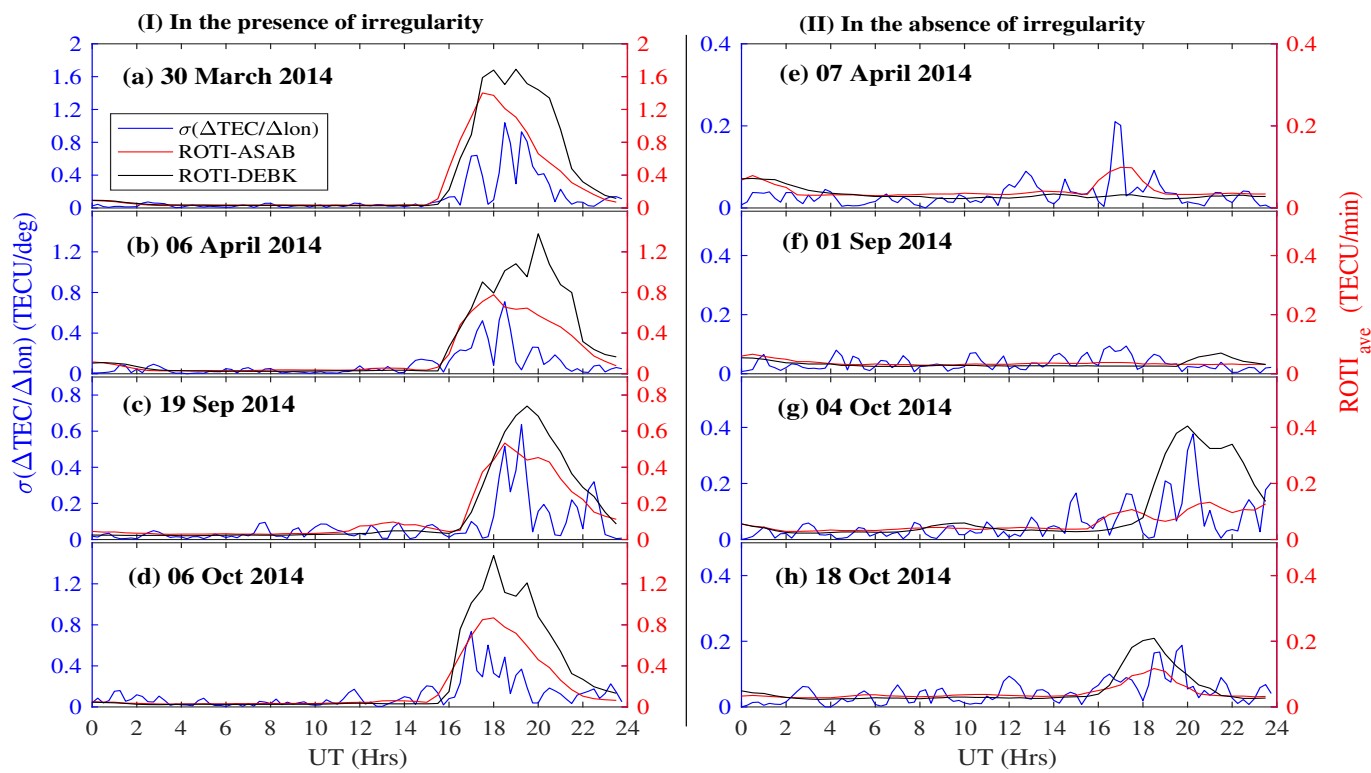

**Figure 4.** Typical examples of the diurnal variation of $\sigma(\Delta TEC/\Delta lon)$ (blue curve) and $ROTI_{ave}$ over ASAB (red curve) and DEBK (black curves) (I) in the presence of ionospheric irregularities on (a) 30 March 2014, (b) 06 April 2014, (c) 19 September 2014 and (d) 06 October 2014 (II) in the absence of ionospheric irregularities on (e) 07 April 2014, (f) 01 September 2014, (g) 04 October 2014 and (h) 18 October 2014. Local time (LT) = UT + 3 hr.

To present the effect of the storms on the variation of ionospheric irregularities and spatial gradient of TEC, the quiet monthly mean of ROTI and $\sigma(\Delta TEC/\Delta lon)$ were used as a background condition (indicated by red curves) in Fig. 5 (a-d), (i)-(iii). The intensity level of ROTI and $\sigma(\Delta TEC/\Delta lon)$ observed after postsunset period during the storm main phase (on 19 February and 12 April 2014) (see, Fig. 5 (a) and (b), (i) - (iii)) was less than the quiet monthly mean, indicating the suppression effect of
5 the storms on occurrence of ionospheric irregularities. On 12 September 2014 storm main phase, however, the intensity level of ROTI and $\sigma(\Delta TEC/\Delta lon)$ observed after was greater than the quiet monthly mean, indicating the triggering effect. During the second storm main phase on 20 February 2014, the intensity level of both ROTI and $\sigma(\Delta TEC/\Delta lon)$ was higher than the quiet monthly mean values. During storm main phase of 27 August 2014, on the other hand, significant effect of the storm on the occurrence of ionospheric irregularities and spatial gradient was not observed.
10 When the presence of ionospheric irregularity is observed over both stations ($ROTI_{ave} \geq 0.4$ TECU/min), the magnitude of $\sigma(\Delta TEC/\Delta lon)$ observed in the post-sunset period shows enhancement (for example, 12 September 2014 and 12 April 2014); and when the occurrence of ionospheric irregularities is suppressed ($ROTI_{ave} < 0.4$ TECU/min), the mag-

nitude of $\sigma(\Delta TEC/\Delta lon)$ shows reduction (for example, 19 February 2014). During storm days, when the presence of ionospheric irregularities is observed and suppressed, the spatial gradient of TEC $\sigma(\Delta TEC/\Delta lon) > 0.7$ TECU/deg and $\sigma(\Delta TEC/\Delta lon) \leq 0.4$ TECU/deg were observed, respectively. The magnitude of $\sigma(\Delta TEC/\Delta lon)$ observed during night-time period could indicate the presence/absence of ionospheric irregularity. In the presence of ionospheric irregularities, the

enhancement/reduction in the spatial gradient of TEC observed during post-sunset period during geomagnetic quiet/disturbed conditions was higher than when ionospheric irregularities are suppressed (see. Figs. 4 and 5). The triggering/suppression in the occurrence of ionospheric irregularities during geomagnetic storms was related to the enhancement/reduction in the gradient of TEC, $\sigma(\Delta TEC/\Delta lon)$ (see Figs. 5). As can be seen from Figs. 5, the geomagnetic storm appears to show a similar effect on the spatial gradient of TEC as it has on ionospheric irregularities. The intensity level of both $ROTI$ and $\sigma(\Delta TEC/\Delta lon)$

observed during post-sunset hours was affected by geomagnetic storms. The magnitude of the spatial gradient of TEC show enhancement/reduction during 12 September 2014 and 19 February 2014 storm days, respectively. In the same way, the intensity level of ROTI (a proxy of ionospheric irregularities) during the storms was above/below the threshold value, respectively.

The effect of geomagnetic storms on the occurrence of ionospheric irregularities could be related to the magnitude and polarity of the z-component of interplanetary magnetic field (IMF Bz) (Biktash, 2004) or local time at which the maximum

negative excursion of Dst occurs (e.g., Aarons and DasGupta, 1984; Aarons, 1991). During magnetic disturbances when the IMF Bz is southward, the penetration of magnetospheric to low-latitude results in an eastward disturbance electric field on the dayside and westward electric field disturbance on the nightside, with the effect being maximum in the dusk sector which augments with evening prereversal enhancement (Fejer and Scherliess, 1998). When the IMF Bz turns northward, however, the polarity of the low-latitude electric fields are reversed (Kelley et al., 1979). On 19 February 2014 (Fig. 5 (a, v) and 12

April 2014 (Fig. 5 (b, v), a northward turning of IMF Bz occurs around $\sim 15:00$ UT and $\sim 19:00$ UT, respectively. When the northward turning of IMF Bz occurs around the dusk, the overshielding electric field might develop and cause a decrease of the upward vertical plasma drift, and hence suppress the occurrence of ionospheric irregularities. Based on local time at which the maximum negative excursion of Dst occurs, Kassa and Damtie (2017) indicated the suppression effect of storm occurred on 19 February 2014 on the occurrence of ionospheric irregularities. In the case of 12 September 2014, however,

a southward turning of IMF Bz occurs around dusk sector ($\sim 19:00$ UT). In this period of transient southward turning of IMF Bz, the undershielding electric field might cause eastward perturbation which reinforce the original PRE electric field, thus increase the vertical plasma drift. This leads to the triggering effect of the storm on the occurrence of irregularities. The enhancement/reduction in the intensity level of the spatial gradient of TEC indicated by $\sigma(\Delta TEC/\Delta lon)$ during the storm periods respond to the storm similar to ROTI. During storm recovery phases (for example, 13-14 April 2014 and 13-

14 September 2014), the triggering/suppression in the occurrence of ionospheric irregularities also appears to be related to enhancement/reduction in the gradient of TEC.

Geomagnetic storm was reported to either trigger or inhibit the occurrence of ionospheric irregularities (Aarons, 1991; Martinis et al., 2005; Oladipo and Schuler, 2013a; Kassa and Damtie, 2017). Different suggestions have been reported on how the geomagnetic storm affect the PRE electric field and hence the generation of ionospheric irregularities (e.g., Aarons, 1991;

Biktash, 2004). During geomagnetic storms, the prompt penetration electric field (PPEF) and the disturbance dynamo electric

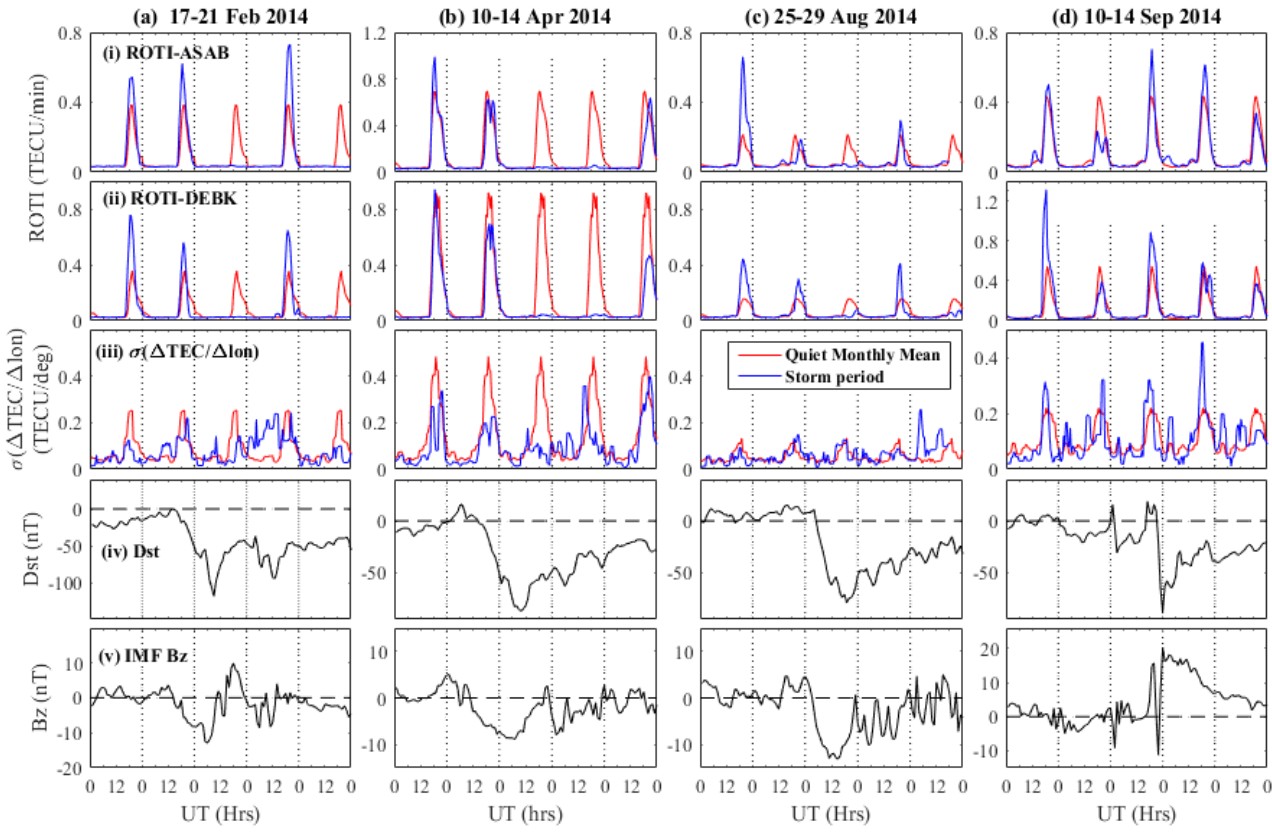

**Figure 5.** Variation of ROTI over (i) ASAB and (ii) DEBK, and (iii) $\sigma(\Delta TEC/\Delta lon)$ during geomagnetic storms (blue curves) (a) 17-21 February 2014 (b) 10-14 April 2014 (c) 26-29 August 2014 and (d) 10-14 September 2014 with respect to the quiet monthly mean (red curves). Local time (LT) = UT +3 hr.

field (DDEF) are the two major electric field sources which change the magnitude/polarity of the background electric field. The PPEF direction is eastward/westward during daytime/nighttime, whereas the DDEF has opposite polarity with PRE electric field and PPEF. They affect the occurrence of ionospheric irregularities by modulating the background electric field. In the absence of observations over the region of interest, the Prompt Penetration Equatorial Electric Field Model (PPEFM) (Manoj and Maus, 2012) can be utilized to observe the effect of PPEFs. To indicate the mechanism that affects the occurrence of ionospheric irregularities, we presented the temporal variation of equatorial electric field over 37°E during geomagnetic storm conditions for (a) 19 February 2014 (b) 12 April 2014 (c) 27 August 2014 and (d) 12 September 2014 (see Fig. 6). The blue curve represents the background quiet-time electric field. The red curve represents the total electric field (PPEF superimposed on the background quiet-time electric field). On 19 February 2014 (Fig. 6 (top panel)), it can be clearly noted that the total electric field observed near sunset (∼ 18:00 LT, indicated by black arrow) is below the background electric field. It must be

mentioned here, that the reduced PRE may have contributions from westward disturbance dynamo (DDEF) electric field. On this storm day, in addition, the Bz component of interplanetary magnetic field during dusktime turns northward related to over-shielding electric field which suppresses the background electric field. As a result, the post-sunset ionosphere drifts downward leading to the suppression in the ionospheric irregularities (represented by ROTI) and reduction in the spatial gradient of TEC

(represented by $\sigma(\Delta TEC/\Delta lon)$) relative to quiet background condition. The inhibition (triggering) effect of 12 April 2014 (12 September 2014) storm events on the occurrence of ionospheric irregularities and spatial gradient was not evident from storm-time electric field, EEF. This could be due to difference in the intensity of the storm. While the storm on 19 February 2014 is strong (Dst = -120 nT), it is moderate on the other events (-87 nT, -79 nT, and -88 nT) on 12 April, 26 August and 12 September 2014, respectively. In addition, the mechanism can be explained by the local time at which the maximum

negative excursion of the ring current energy occurs (Aarons and DasGupta, 1984; Aarons, 1991). During 12 April 2014 and 12 September 2014 storm main phase, the peak negative excursion of Dst occurs around daytime (postmidnight) period, respectively. The occurrence of ionospheric irregularity observed was inhibited (triggered) consistent with Aarons (1991), and the spatial gradient of TEC, too. The triggering/inhibition effect of the storm on the generation of ionospheric irregularities could also be related to the magnitude and direction of z-component of interplanetary magnetic field (Biktash, 2004). It is

evident from Figures 4 and 5 that the spatial gradient of TEC, $\sigma(\Delta TEC/\Delta lon)$, obtained from two closely located stations could show the presence and absence of ionospheric irregularities.

Figures 7 (a-d) respectively, show the annual variation of $ROTI_{ave}$ (over ASAB and DEBK), spatial gradient of TEC ($\Delta TEC/\Delta lon$) and its standard deviation $\sigma(\Delta TEC/\Delta lon)$ in the year 2014. The intensity level of $ROTI_{ave}$, $\Delta TEC/\Delta lon$, and $\sigma(\Delta TEC/\Delta lon)$ is indicated in the color bar. As stated by Oladipo and Schuler (2013b), the value of $ROTI_{ave} \geq 0.4$

TECU/min shows the presence of ionospheric irregularity. The occurrence of ionospheric irregularities at the two stations, as indicated by intensity level of $ROTI_{ave}$, was predominantly observed in the pre-midnight periods, mainly between 19:00 LT and 24:00 LT. The large-scale ionospheric irregularities, which are responsible for the scintillation of trans-ionospheric signals at GNSS frequencies, are more pronounced during post-sunset hours. The observed phase fluctuation shows monthly variations and there is also a seasonal trend in the occurrence of ionospheric irregularity. Strong and weak ionospheric irregularities are

observed in March equinox and in June/July solstices, respectively.

It can be seen from Figure 7c that positive/negative values in the gradient of TEC were observed. Maximum enhancement/reduction in the gradient of TEC was observed mostly during the post-sunset (18:00 - 24:00 LT) and post-midnight (24:00 - 06:00 LT) period. Equation (1) was applied to all days (364 days) of the year 2014 in computing the spatial gradient of TEC. Out of the total observed daily maximum value of the gradient of TEC, about 194 days (in percent about 53%) of

them fall in this time period. There were also cases where the maximum enhancement/reduction in the values of the gradient of TEC was observed in the early morning period. In Figure 7d the diurnal, monthly and seasonal variation in the standard deviation of spatial gradient of TEC, $\sigma(\Delta TEC/\Delta lon)$, was clearly observed and its variation show similarity with variation of $ROTI_{ave}$. Maximum enhancement in $\sigma(\Delta TEC/\Delta lon)$ was observed in the evening time period, 19:00 - 24:00 LT. The seasonal variation in $\sigma(\Delta TEC/\Delta lon)$ also appears frequently in equinoctial months, but rarely in solstice months. Such kind

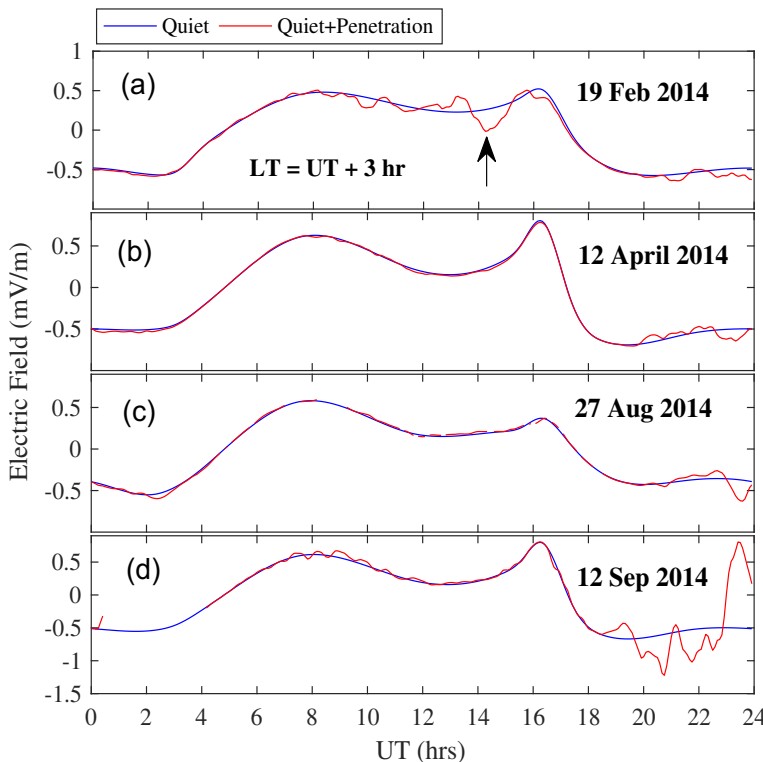

**Figure 6.** Prompt penetration electric field observed during storm days of (a) 19 February 2014 (b) 12 April 2014 (c) 27 August 2014 and (d) 12 September 2014. The arrow indicates time of PRE. Local time (LT) = UT +3 hr.

of variation could be related to the magnitude of $E \times B$ drift. Cesaroni et al. (2015) also found a seasonal variation of the TEC spatial gradients and they reported that it is larger during the equinoctial seasons than in the solstice seasons.

Figures 7 (e-h) respectively, show the daily maximum values of $ROTI_{ave}$ (over ASAB and DEBK), spatial gradient of TEC ($\Delta TEC/\Delta lon$) and standard deviation of spatial gradient of TEC $\sigma(\Delta TEC/\Delta lon)$, respectively, in the year 2014. As can be
5  observed from Figures 7 (e-h)(right panel), the daily maximum value of $ROTI_{ave}$, $\Delta TEC/\Delta lon$ and $\sigma(\Delta TEC/\Delta lon)$ shows monthly and seasonal variations, and an equinoctial asymmetry is also observed. The daily maximum value of $\Delta TEC/\Delta lon$ and $\sigma(\Delta TEC/\Delta lon)$ shows similar trends with the daily maximum value of $ROTI_{ave}$ observed over ASAB and DEBK stations. The trend they show has similarity with the time of occurrence of maximum enhancement/reduction, monthly and seasonal variations. Moreover, the seasonal variation observed in both variables exhibits equinoctial asymmetry, where the
10  March equinox was greater than September equinoxes. The mechanism of generation of the enhancement in vertical drift just after sunset was detailed by Farley et al. (2008). The magnitude of peak vertical drift is known to control the seasonal and day-to-day variations in the occurrence of equatorial spread F (Manju et al., 2009; Tulasi Ram et al., 2006).

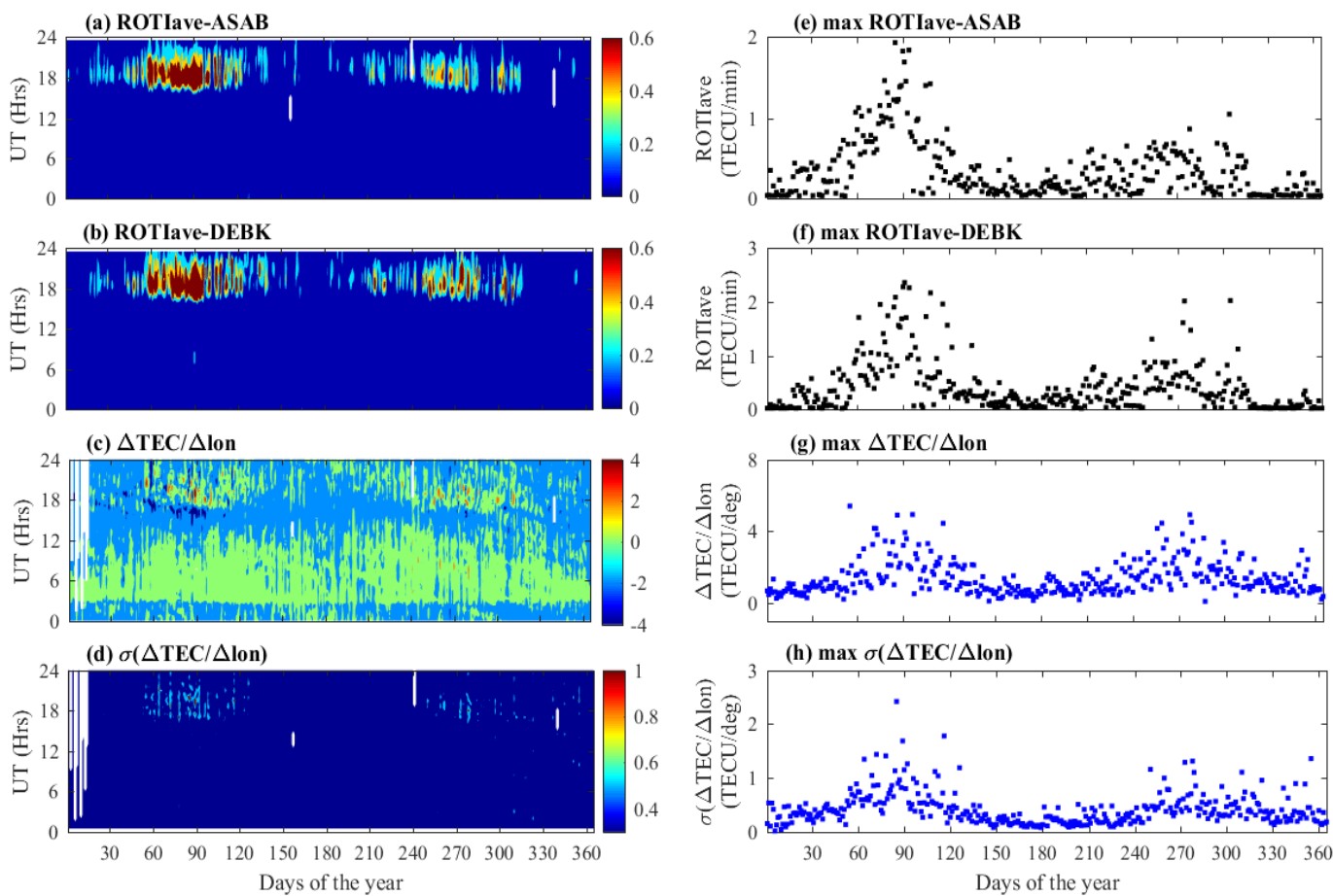

**Figure 7.** Annual and daily maximum value variation of (a,b) $ROTI_{ave}$ over Asab (ASAB), (c,d) $ROTI_{ave}$ over Debark (DEBK), (e,f) spatial gradient of TEC ($\Delta TEC/\Delta lon$), (g,h) standard deviation of gradient of TEC $\sigma(\Delta TEC/\Delta lon)$ in year 2014. The $ROTI_{ave}$ in TECU/min and $\Delta TEC/\Delta lon$ in TECU/deg is indicated in color bar. Local time (LT) = UT + 3 hr.

Figure 8 depicts the quiet-monthly mean of $ROTI_{ave}$ (over ASAB and DEBK) (red and black curves) and $\sigma(\Delta TEC/\Delta lon)$ (blue curve) in the year 2014. The enhancement/reduction in the intensity of $\sigma(\Delta TEC/\Delta lon)$ show similar trends with $ROTI_{ave}$, and was stronger/weaker during equinoctial/solstice months. Equinoctial asymmetry both in $ROTI_{ave}$ and $\sigma(\Delta TEC/\Delta lon)$ was also evident from Fig. 8, where March equinoxes were stronger than September equinoxes. As expected, the TEC spatial gradients are also found to be larger during the equinoctial seasons than in the solstice seasons.

Figure 9 shows the relationship between the standard deviation of the spatial gradient of TEC $\sigma(\Delta TEC/\Delta lon)$ and $ROTI_{ave}$ (over ASAB and DEBK) in the year 2014. The daily maximum values of $\sigma(\Delta TEC/\Delta lon)$ and $ROTI_{ave}$ were considered to examine the correlation. The correlation coefficient between $\sigma(\Delta TEC/\Delta lon)$ and $ROTI_{ave}$ is about 0.7915 (in ASAB) and 0.7975 (in DEBK), respectively. Studies indicate that the gradient of TEC can be computed from a pair of

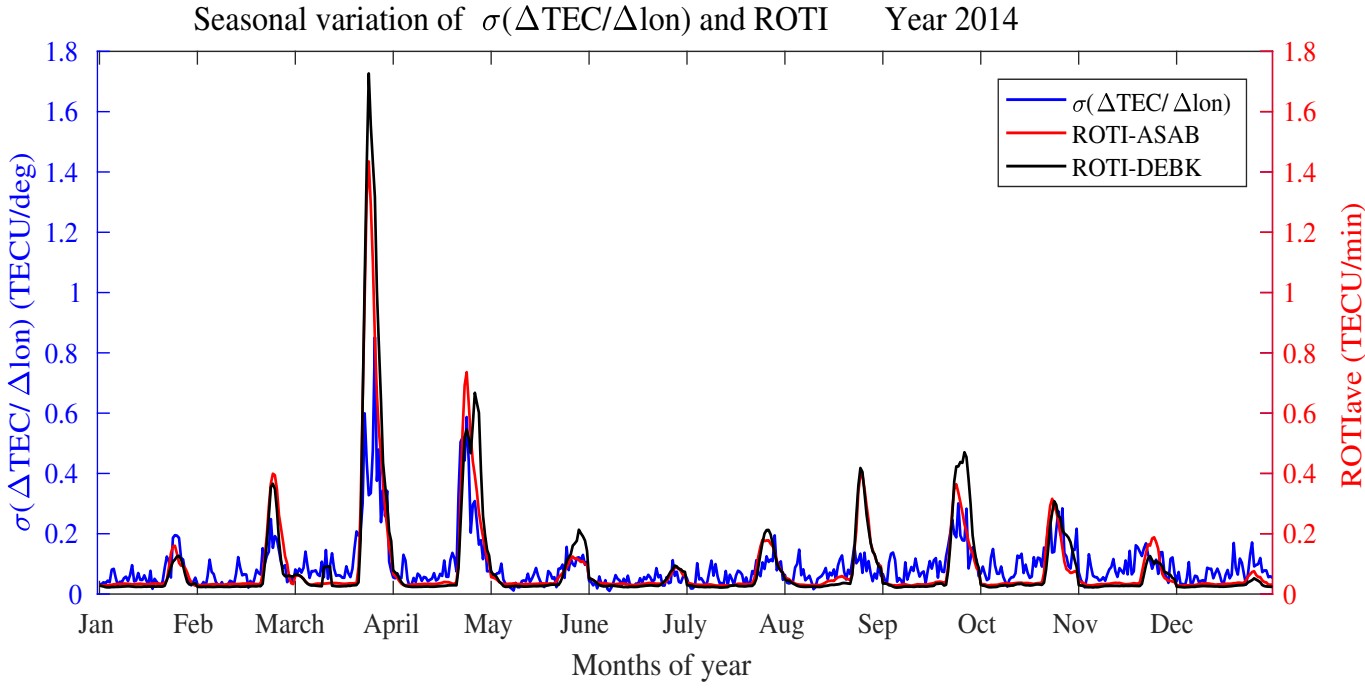

**Figure 8.** Seasonal variation of $ROTI_{ave}$ over ASAB (red curve), $ROTI_{ave}$ over DEBK (black curve), and $\sigma(\Delta TEC/\Delta lon)$ (blue curve) in the year 2014

closely-spaced receiver stations ($< 2^o$) such that the two receivers share the same GPS satellite. In our case, however, the two stations are separated by $5^o$. The moderate correlation obtained might be attributed to the wider longitudinal separation ($5^o$) between the two stations. The other factor for the moderate correlation between the gradient of TEC and the occurrence of ionospheric irregularities might be the way ROTI was computed (since ROTI contains both the spatial and temporal varia-

tion in TEC). It is well known that ROT is the combination of the spatial and temporal gradients. However, by giving less attention to the spatial gradient effect, previous authors often used $\Delta TEC/\Delta t$ to examine the fluctuation in TEC. It is not only the temporal variation of TEC that contribute to the fluctuation in the phase and amplitude of the signals but also the spatial gradient of TEC. The computed correlation coefficient between the TEC gradient and ROTI, here, gives an indication of the contribution of the spatial gradient of TEC to ROTI (or ROT) usage. This can give the case where the spatial gradient

of TEC between two nearby located stations can be used as an indicator of occurrence of ionospheric irregularities. Every night time enhancement/reduction in the gradient of TEC may not be a guarantee to indicate the occurrence/non-occurrence of ionospheric irregularities. However, there are cases which show the occurrence of irregularities over both stations (ASAB and DEBK) when the night time enhancement/reduction in the TEC gradient were observed. Hua and Chunbo (2009) discussed the relation between ROTI index, ionospheric TEC gradient, and vertical TEC. Cesaroni et al. (2015) also described

the importance of the information provided by the TEC gradients variability and the role of the meridional TEC gradients in driving scintillation. By comparing the zonal and the meridional components of average and standard deviation of $\Delta TEC$,

Cesaroni et al. (2015) reported that the North-South (N-S) gradients of TEC are significantly larger than their East-West (E-W) counterparts, regardless of the season. For GNSS Ground-Based Augmentation System, Saito and Yoshihara (2017) observed extreme spatial gradient in ionospheric total electron (about 3.38 TECU/km) associated with plasma bubbles. It is suggested that when scintillation events are investigated ionospheric TEC gradient is also one of the considerable parameters.

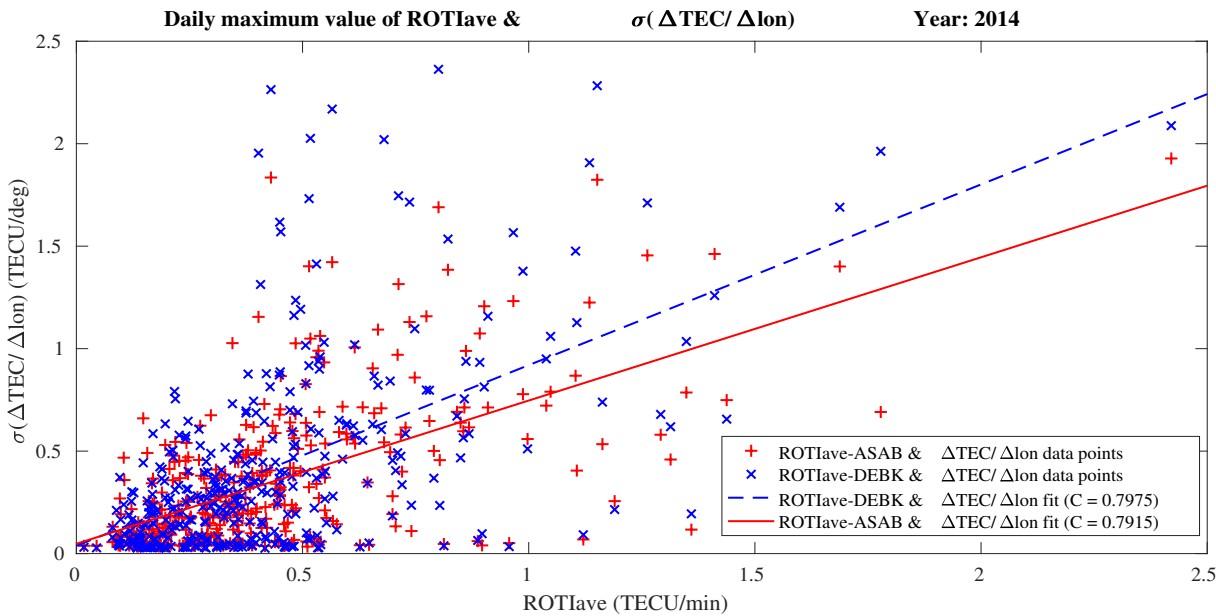

**Figure 9.** Relation between the daily maximum value of $\sigma(\Delta TEC/\Delta lon)$ and the daily maximum value of $ROTI_{ave}$ over ASAB (red, +) and DEBK (blue, x) in the year 2014. The blue broken and solid red lines indicate the linear fit between the daily maximum value of $\sigma(\Delta TEC/\Delta lon)$ and $ROTI_{ave}$ for ASAB and DEBK, respectively.

5    Figure 10 presents the percentage occurrence of ionospheric irregularities over ASAB (blue) and DEBK (red) in the year 2014. The estimation of the percentage occurrence of ionospheric irregularities was made for quiet days of each months of year 2014. The percentage occurrence of irregularities was calculated by counting the number of days in a month with $ROTI_{ave} \geq 0.4$ TECU/min and dividing by the number of days in a month for which the data are available, and multiplied by $100\%$ (Oladipo et al., 2014). Since the two stations are close to each other, the occurrence of ionospheric irregularities observed over both stations does not show major differences. Two peaks of irregularity occurrence were observed around the middle of the equinoxes (i.e., in March and September) at both stations. This could be related to the alignment of the magnetic field lines with a geographic meridian (Burke et al., 2004; Tsunoda, 2005, 2010). The seasonal variation of ionospheric irregularities exhibits an equinoctial asymmetry in its occurrence especially at the two peaks (i.e., in March and September), where March equinox was greater than September equinox. The maximum $ROTI_{ave}$ observed over this station in the year 2014 was about 1.8 TECU/min in March 2014 and the minimum level of $ROTI_{ave}$ was observed on December Solstice.

Based on a few station observations, earlier studies indicated the equinoctial asymmetry in the occurrence of L-band scintillations which was attributed to differences in the meridional winds during two equinoxes (e.g., Nishioka et al., 2008; Maruyama et al., 2006; Otsuka et al., 2006). Nishioka et al. (2008) analyzed the occurrence characteristics of plasma bubbles using GPS-TEC obtained all over the globe and found equinoctial asymmetry in their occurrence. They suggested that equinoctial asymmetry could be due to the asymmetric distribution of integrated conductivities during analyzed periods. Using three ionosonde observations, Maruyama et al. (2006) reported that meridional wind is the key factor for the equinoctial asymmetry. Using multi-instrument observations, Sripathi et al. (2011) examined the equinoctial asymmetry in scintillation occurrence in the Indian sector and they suggested the asymmetry in the electron density distribution and meridional winds as a possible causative mechanisms. Manju et al. (2012) also reported equinoctial asymmetry in ESF occurrence and they discussed the possible role of asymmetric meridional winds. Manju and Haridas (2015) observed a significant asymmetry in the threshold height between the vernal equinox and autumn equinox and underlines the distinct differences in the role of neural dynamics in ESF triggering during the two equinoxes. Based on scintillation index ($S_4$) and GPS-TEC derived indices, the seasonal and equinoctial asymmetry in the occurrence of ionospheric irregularities over equatorial/low-latitude region of African was presented (Susnik and Forte, 2011; Paznukhov et al., 2012; Oladipo and Schuler, 2013b; Oladipo et al., 2014; Seba and Tsegaye, 2015; Mungufeni et al., 2016). By employing the horizontal wind model (HWM14), Seba et al. (2018) recently reported that the difference in the wind pattern between March and September is one of the factors for the equinoctial asymmetry. The local time and seasonal trends of occurrence of ionospheric irregularities observed in this study are similar to those reported in the previous studies (Aarons, 1993; Basu et al., 1988; Olwendo et al., 2013; Amabayo et al., 2014; Seba and Tsegaye, 2015). The equinoctial asymmetries in the occurrence of ionospheric irregularities observed in our case might also be due to the direction of the meridional winds during equinoxes over the stations.

In terms of diurnal, monthly, and seasonal behavior the enhancement/reduction in the spatial gradient of TEC and the occurrence of ionospheric irregularities appears to show similar trends. And, it is evident from the above result that the spatial gradient of TEC between two nearby located stations where the two receivers lie nearly along the same latitudes convey insight into the relation between large-scale ionospheric irregularity occurrence and spatial gradient in TEC.

## 4  Conclusions

In this study, we presented for the first time the relationship between the spatial gradient of TEC between two nearby located stations (ASAB and DEBK) and the occurrence of ionospheric irregularities over Ethiopia, an equatorial region, using ground based GPS-TEC observations. The following observations are a summary of our analysis. The daytime equatorial electrojet (EEJ) derived from H-component of geomagnetic field and the real-time electric field (EEF) model (Manoj and Maus, 2012) correlates linearly and positively with correlation coefficient of C = 0.6 in the year 2014. Most of the peak enhancement/reduction value of $\Delta TEC/\Delta lon$ and $\sigma(\Delta TEC/\Delta lon)$ were observed about 1-2 hrs later from post-sunset enhancement of equatorial electric field (EEF), which indicates that EEF and the spatial gradient of TEC have a strong relationship. In terms of month and season, the nighttime pattern of the spatial gradient of TEC ($\Delta TEC/\Delta lon$) and its standard

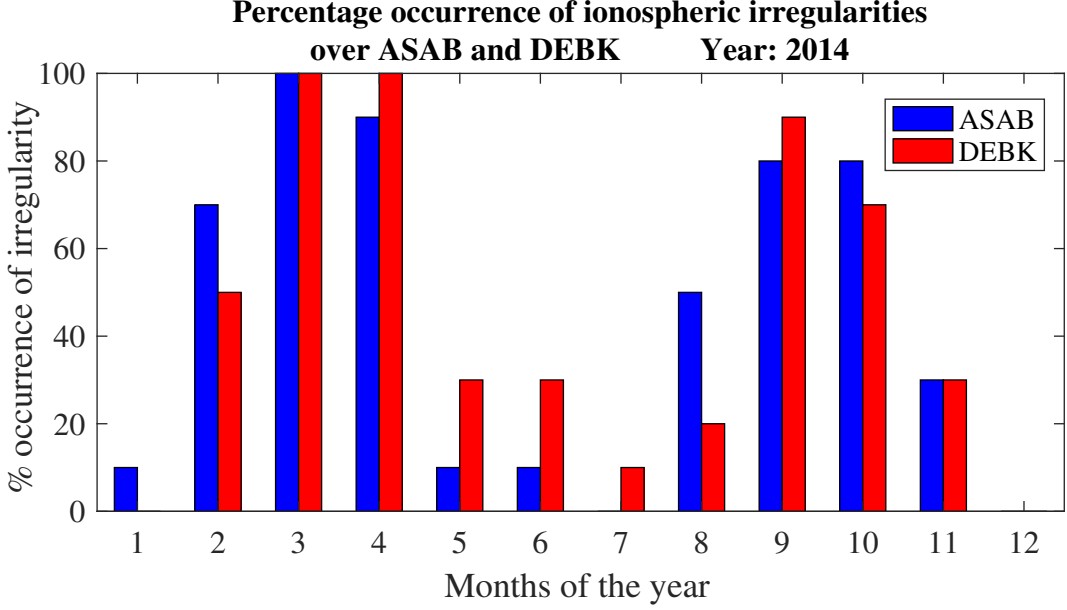

**Figure 10.** Percentage of occurrence of ionospheric irregularities over ASAB (blue) and DEBK (red) stations during quiet days of each month in the year 2014 based on $ROTI_{ave}$ index.

deviation $\sigma(\Delta TEC/\Delta lon)$ show a similar trend with $ROTI_{ave}$. The relation between the standard deviation of the spatial gradient of TEC, $\sigma(\Delta TEC/\Delta lon)$, and ionospheric irregularity occurrence indicated by $ROTI_{ave}$ were also presented. The correlation coefficient between $\sigma(\Delta TEC/\Delta lon)$ and $ROTI_{ave}$ was about 0.7975 (in ASAB station) and 0.7915 (in DEBK station). Both $\sigma(\Delta TEC/\Delta lon)$ and $ROTI_{ave}$ show maximum enhancement/reduction during equinoctial/solstice months.

5  Equinoctial asymmetry was also observed in both $\sigma(\Delta TEC/\Delta lon)$ and $ROTI_{ave}$, where March equinox was greater than September equinox. The intensity level of $\sigma(\Delta TEC/\Delta lon)$ was stronger/weaker when the occurrence of ionospheric irregularity is present/absent. When the occurrence of ionospheric irregularities was suppressed (for example, during geomagnetic disturbed conditions), the nighttime peak value of $\sigma(\Delta TEC/\Delta lon)$ was smaller. Based on the above results, the strength of spatial gradient of TEC between the two nearby located stations lying along the same geomagnetic latitudes could indicate the

10  presence of large-scale ionospheric irregularities. During post-sunset period on equinoctial months (March and April, 2014), the relation between the spatial gradient of TEC/electron density and the zonal electric field was observed. The threshold value of the gradient of TEC and its standard deviation $\sigma(\Delta TEC/\Delta lon)$ and the minimum longitudinal separation between two stations that could lead us to predict the occurrence of ionospheric irregularities are not addressed in the current study and this will be considered in the future investigation.

## Acknowledgments

We thank Ethiopian Space Science and Technology Institute (ESSTI) for facilitating conditions to do this research. We also acknowledge the administration and staff the Space Science Directorate of the South African National Space Agency (SANSA) for the support during the research visit of the first author to the institution. The authors would like to express their gratitude to

the International GNSS Service (IGS) for providing the GPS data (ftp://cddis.gsfc.nasa.gov). We thank the Cooperative Institute for Research in Environmental Sciences (CIRES) team for real-time PPEEFM model at http://geomag.org/model/PPEFM/ RealtimeEF.html and the World Center for Geomagnetism, (WDC) Kyoto (http://wdc.kugi.kyoto-u.ac.jp0), the International Service of Geomagnetic Indices (ISGI). We are also grateful to the online AMBER (http://magnetometers.bc.edu/index.php/) and INTERMAGNET (http://www.intermagnet.org/) for freely providing magnetometer data. Melessew Nigussie work has

been supported by Air Force Office of Scientific Research, Air Force Material Command USAF under Award No.FA9550-16-1-0070. T. Dugassa, thanks Bule Hora University for permitting study leave.

## Data availability

The data used in this study were obtained from ftp://cddis.gsfc.nasa.gov, http://geomag.org/models/PPEFM/RealtimeEF.html, http://magnetometers.bc.edu/index.php/, http://www.intermagnet.org/, http://wdc.kugi.kyoto-u.ac.jp0, and http://isgi.unistra.

fr/data_download.php.

## Author contributions

T. Dugassa performed the analysis and drafted the manuscript with the support of J.B. Habarulema and M. Nigussie provided with constructive scientific advices.

## Competing interests

The authors declare that they have no conflict of interest.

## Review statement

This paper was edited by Erdal Yiğit and reviewed by two anonymous referees.

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
