# Peer review of "Investigation of the relationship between the spatial gradient of total electron content (TEC) between two nearby stations and the occurrence of ionospheric irregularities"

_Annales Geophysicae, 2018_

## Referee Comment (RC1) · Anonymous Referee #1 · 6 Jan 2019

General comments

The study attempts to show how the difference between the TEC of two close GNSS stations can be used as a precursor of ionospheric irregularities in the post sunset period over both stations. The study being the first of its kind in the African sector is worthy of interest couple with the fact that it is well written and it gives insight about a possible relation between electric field and irregularities in the post sunset period. However, the authors first need to give a good justification why they want to use TEC gradient between two stations as a proxy of irregularities. Giving the fact that ROTI which can be easily estimated is already an indicator of irregularities why use the gradient between two stations to do same work? What if the constraints "over the same latitude, but separated by a longitude of about 5 degree" is removed what happens to the relation?

Specific comments

The first issue I have with this paper is the way the relationship is established between TEC gradient and ROTI. Presenting variations of two quantities side by side does not give us any quantitative information about the nature of the relation. One will wish to know how that relation has been quantified in terms of correlation or ratio between the quantities. (For example we know in quantitative manner how ROTI relates to Scintillation index S4 and how this relationship varies depending on certain factors). It would have been interesting to have such quantitative information in this study. The abstract mentioned "observe the correlation between the spatial gradient of TEC and the occurrence of ionospheric irregularities" but there is no such correlation in the whole texts. Another concern is that the study never really specified or took into consideration the quietness and/or disturbed nature of the days used and if it did little information is given on this. We know the study covers year 2014 which is about 365 days. But we do not know how many days were used. If they were days without data, how many days were quiet and disturbed. As such, one is forced to assume that the relation between TEC gradient and ROTI was given for all dyasof year 2014 and was unaffected by magnetic activity. This could be misleading given the fact that both TEC gradient and ROTI are severely affected by magnetic activity. True the authors talk about quiet time but failed to tell us how and which criteria were used to segregate/ isolate these days and how many of them were used in the computation of TEC gradient and ROTI in both stations simultaneously. In establishing a relation between equatorial electric field and spatial gradient, the authors used only four days in year 2014. I don't think this is sufficient enough to show any kind of relationship between both quantities. In justifying the magnetic data gap (the H component of the Earth's magnetic field) in 2014, the authors performed a correlation between the equatorial electrojet ($\Delta H$) and equatorial

electric field (EEF) for quiet days in 2012. Again no information is given on these days and how they were selected and how many they were. Another problem here is that the authors assumed that the correlation between ($\Delta$H) and EFF as obtained in 2012 (0.7) will be the same in 2014. (they actually did the correlation to justify the use of EEF in 2014 in the absence of $\Delta$H data). We have 2 years between 2012 (Average F10.7 $\sim$ 120 s.f.u) and 2014 (Average F10.7 $\sim$ 146 s.f.u). Based on this I believe the solar effect will affect the correlation and this need to be mentioned if not evaluated for the sake of accuracy. (I am sure there will be variations even in the modeled EEF values in 2012 and 2014).

Listing of technical corrections

Abstract Line 9. Change correlation to relation. I didn't see any correlation study between both variables in this work. Line 11 maximum positive/depletions Why not use maximum enhancement and reduction. The spatial gradient will either be positive of negative. A negative gradient means reduction in electron density. Let's avoid using the word depletion since it can be mistaken for TEC depletion. Line 15-16. The spatial gradient of TEC between the two nearby stations could be used as an indicator of the occurrence of ionospheric irregularities. Is it over both stations or it is a general statement?

1. Introduction

Page 2 Line 8. Attests Line 14. Remove mechanism Line 18. ESF write in full. First time used. Line 30. GPS write in full. The GPS scintillation index, S4 is not an instrument. The GPS is. Line 31. Global Navigation Satellite System (GNSS). Use either GNSS or GPS.

Page 3 From lines 1-2, a mention of some work done over Africa has been made. However nothing was said about the scope of such studies, their limitations/gaps and how they relate to this study. Kindly address. Line 26. "and see". Change to as well as study Line 27 "A closely found" change to closely located Line 28. I am not

comfortable with the word 'longitudinal'. Change to spatial for uniformity with title. Line 27-28. What is the justification for the study of the relation between longitudinal (in this case spatial) gradient of TEC derived from two GPS receivers and occurrence of ionospheric irregularities still using GPS? Line 29. Same as in line 28.

Page 4 2. Data and methods Line 2-3 Kindly read that statement and adjust for easy flow. Line 5. Why year 2014 only? Is there any particular justification for the choice of this year? Line 6. Remove "of" Lines 7-8. Change the first average to "mean" Line 10 change were to "was" Line 10 "then analyzed to show the possible indicator of". This statement is not correct. Adjust Line 11 "The spatial gradient of TEC between the two nearby stations are located nearly along the same". Adjust statement. May be you should delete "are". Line 14. Any reference for equation 1? Line 16-17. I am not satisfied with your definition of $\Delta H$ the way it is and the way you associate it to the EEJ in these particular lines. In addition you need to add how the H was processed and corrected for baseline value and non cyclic variations. Line 22. . . .is a transfer function model which to models the daily variations. . .Check the sentence.

Page 5 Line 1. "which are mapped from interplanetary electric field (IEF) data". Change to . . .which are mapped in the interplanetary electric field (IEF). Line 5. I think you need to clearly explain the various options that the model provides and then proceed to tell us exactly which of the three options you used and why. Line 16, add "s" to station. Line 23. Put a comma after reliable. Line 24. Change "from the model" to 'it' Line 32. Was ROTI introduced to quantify the ROT measurements or ionospheric irregularities? Clarify please.

Page 6 Line 6. Adjust to (Ma and Maruyama, 2006). Line 23. I thought the scope of the study was 2014. Why use data from 2012? Have you accounted for the yearly variation and solar activity influence in juxtaposing your 2012 and 2014 data? Please could you clarify this? 3 Results and Discussions Line 22. For some of selected. Remove of Lines 22-23. How did you select those quiet days? How many where there? What is the temporal resolution of both $\Delta H$ and EEF. Is the correlation obtained from "some

selected quiet days" be an adequate representation of all other quiet days in year 2012? Check Figure caption in Figure 1 and harmonize. Let's know whether you use some days or quiet days of month of year 2012.

Page 7 Lines 2-5. You gave us a beautiful description of how $\Delta$H can be derived between the two magnetometers just for you to come and tell us that the data were not available for year 2014. I think it should have been the other way round. Line 3. Correct Adegrat to Adigrat Line 4-5. To solve this data gap we used the daytime information of equatorial electric field derived from the real-time prompt penetration electric field model as an option. Did you use the real time model of Ionospheric electric fields of the real-time prompt penetration electric field model? In Page 6 lines 21-23 the authors claimed the relation is for some selected quiet days of months of year 2012 but in Page 7, lines 5-7 they state that the same relation is for year 2012. This may be extremely misleading.

We need a clear explanation on how you wish to use correlation result for 2012 to support some of your results in 2014. You must be aware of the solar activity influence on vertical drift. It will not be totally accurate to say that the correlation in 2012 will be the same as those in 2014. May be you did that as an indication of something. You need to clarify.

Page 8 Line 5. Be consistent. Is it Figure or Fig? (check in all texts and harmonize). Line 7. Replace "but lags' with " but after ''. ... Line 8. The depletions in the gradient. I am not comfortable with the word depletion. Kindly use the reduction in the gradient. Line 9. ....maximum positive of the spatial gradient of... Replace with " the peak of spatial gradient" or "the maximum spatial gradient". Line 11. Change depletion with reduction Lines 16-19. Why over Asab only?

Page 9 30 March 2014, 10 April 2014, 20 September 2014, 10 October 2014. Where these days selected randomly? The caption in Figure 3 should be self explanatory and should tell us the stations (Asab and Debark) that were used for the ROTI.

Page 10 Line 1. Do you have any reference for this? Line 1. From the figure,…. Which Figure? Specify. Lines 2-3. A convincing and quantitative way to demonstrate inferences in lines 3-4 is by performing correlation between spatial gradient and irregularities. Lines 7-8. An ionosphere gradient of 518 mm/km was discovered, generated by a plasma bubble. Read the statement and rephrase. Line 14. (see., Fig. 5). Change to as seen in Figure 5. Line 17. Change "a" by "the" Line 18. Change "indicates" to "shows" Line 19. ……..in section (2)… which section 2? Change to as stated earlier. Line 23. Put 's' to period Line 23-24. Equation (1) was applied to all days of the year 2014? Including disturbed days? This is where it is important to separate disturbed days from quiet ones. We know that gradients can be significant during geomagnetic storms. Line 28 – 32. Most of the observed features have not been discussed and plausible answers not given to explain them. Line 32. Change depletions to reductions. Figure 4. a) Diurnal variation of the spatial gradient of TEC over ASAB and DEBK , b) Daily maximum value of the spatial gradient of TEC variation, c) Diurnal variation of ROTIave over ASAB station and d) Daily maximum value variation of ROTIaave over ASAB station in the year 2014. Check this Figure caption and adjust according to your Figures (e) and (f) are missing.

Page 11 Lines 1-2. If you can show it don't say it. Lines 10-11. "The trend they show has similarity with" The trend is already a similarity. Adjust the statement. The caption of Figure 4 is misleading. Please check and let it conform with what you have in the texts. Why not add a correlation plot between spatial gradient and ROTI over each station? This is a better way of obtaining quantitative information between both quantities.

Page 12 Line 3. What about Debark? Why is it not presented? Besides, is this Figure for quiet and disturbed periods? How did you segregate the effect of transient disturbances? Line 26. Basu et al., the year is missing.

Page 13 Line 6. Change "has not been seen" to something suitable. 4 Conclusions

Page 14 Lines 2-3. This is inconclusive and cannot feature in this section given the fact the relation between EEF and TEC gradient was investigated for just for 4 days (Figure 2).

5. Acknowledgments Page 14, line 6. Remove and. Page 14, lines 7-8. We acknowledge http://www.geomag.org/models/PPEFM/RealtimeEF.html for providing the data the Prompt penetration equatorial electric field model. Give proper acknowledgement please.

Page 14, line 8. Provide adequate acknowledgement for using the AMBER data (Visit AMBER website for adequate acknowledgement).

References Page 15. Line 31-32. Incomplete reference. Page 16. Line 16-20. Arrange references chronologically. Page 16. Line 36. Adjust the initials. Page 17. Line 1-5. Arrange references chronologically. Also consider the reference in P.16 line 36-37 in the chronological arrangement.

Please also note the supplement to this comment: https://www.ann-geophys-discuss.net/angeo-2018-131/angeo-2018-131-RC1-supplement.pdf

---

## Author Comment (AC1) · 26 Feb 2019

**Response to reviewers' comments on the manuscript angeo-2018-131-RC1:**

**Spatial gradient of total electron content (TEC) between two nearby stations as indicator of occurrence of ionospheric irregularity**

Authors: Teshome Dugassa, John Bosco Habarulema, and Melessew Nigussie

[Figure]

The authors thank the reviewer for his comments that helped to improve the quality of this work. The comments are addressed as shown below.

**Reviewer 1**

**General comments**

1. **The study attempts to show how the difference between the TEC of two close GNSS stations can be used as a precursor of ionospheric irregularities in the post-sunset period over both stations. The study being the first of its kind in the African sector is worthy of interest couple with the fact that it is well written and it gives insight about a possible relation between electric field and irregularities in the post sunset period. However, the authors first need to give a good justification why they want to use TEC gradient between two stations as a proxy of irregularities.**

   **Response**:
   It is well known that ROTI is a proxy for the occurrence of ionospheric irregularities and scintillations. The relationship between the spatial gradient of TEC and ROTI was established to observe how the gradient of TEC affect radio signals, hence give information that the horizontal electron density gradient as an important parameter to predict ionospheric scintillation. After establishing this relationship showing that both parameters give information about ionospheric irregularities, TEC gradient method may be an alternative as it is a simple computation of establishing the difference. The motivation why we used TEC gradient between two stations as a proxy of
irregularities are explained (Page 2: Lines 22-35 and Page 3: Lines 1-23).

2. **Giving the fact that ROTI which can be easily estimated is already an indicator of irregularities why use the gradient between two stations to do the same work?**

   **Response**:
   Our main interest in establishing the spatial gradient of TEC as indicator of irregularity is due to the following reason:

   It is well known that pre-reversal enhancement (PRE) is a postsunset phenomenon. PRE can uplift the ionosphere and create a conducive condition for irregularity formation. This implies the magnitude of the zonal electric field is an important parameter for real-time prediction. It is also k own that PRE is due to the spatial gradient of electron density near solar-terminator. We know TEC is the integral of electron density. So the TEC gradient would help us to estimate the strength of the zonal electric field (Page 3, Lines 7 - 15).

3. **What if the constraints "over the same latitude, but separated by a longitudinal of about 5 degree" is removed? What happens to the relation?**

   **Response**:
   We thank the reviewer for the important point raised. We have a plan to investigate the optimum separation between two GNSS receivers that can provide the best correlation between TEC gradient and irregularity. This will be done in a separate work.

**Specific comments**

1. **How does the relationship between TEC gradient and ROTI is established? The study never really specified or took into consideration the quietness and/or disturbed nature of the days used.**

   **Response**:

   To show the relationship between the TEC gradient and ROTI, the 10 quiet international days in the year 2014 were used. In this study period, there are about 364 days fully available in both stations simultaneously. The quiet international days are obtained from http://wdc.kugi.kyoto-u.ac.jp/qddays/index.html. In total, about 120 quiet days were used in investigating the relationship between TEC gradient and ROTI. The absolute value of the spatial gradient of TEC was used to observe the correlation between the gradient of TEC and ROTI (Page 14, Figure 6). Moreover, in the revised version of the manuscript, we also considered the effect of a geomagnetic storm in developing the relationship between TEC gradient and the occurrence of ionospheric irregularities (Page 12, Figure 4).

2. **In establishing a relation between equatorial electric field and spatial gradient, the authors used only four days in year 2014. Is this sufficient enough to show any kind of relationship between both quantities?**

   **Response**:

   In the previous version of the manuscript, we used four quiet days as a typical example to observe a relation between equatorial electric field and spatial gradient of TEC which is not sufficient enough. In the revised version of the manuscript, however, we modified Figure 2 by adding more data/months to establish the relationship between the equatorial electric field and spatial gradient. In the updated graphs, we took the monthly-quiet mean of the equatorial electric field and TEC gradient of the year

2014 (Page 10, Figure 2).

3. **In Justifying the magnetic data gap (the H component of the Earth's magnetic field) in 2014, the authors performed a correlation between the equatorial electrojet $\triangle H$ and equatorial electric field (EEF) for quiet days in 2012. No information is given on these days and how they were selected and how many they are?**

**Response**:
Five quietest international days of each month of the year 2014 were utilized to perform the correlation between the equatorial electrojet $\triangle H$ and the equatorial electric field (EEF). These days are obtained from http://isgi.unistra.fr/data_download.php. Out of 60 days, only 38 days of data were available simultaneously in both magnetometer measuring stations (Page 8, Figure 1b).

4. **The authors assumed the correlation between $\triangle H$ and EEF as obtained in 2012 (0.7) will be the same in 2014.**

**Response**:
We did the correlation between $\triangle H$ and EEF to justify the performance of the model over the equatorial region of Africa using available magnetometer data (the year 2012). As can be seen from their relation, the day time EEF model and day time $\triangle H$ correlated positively. Since the performance of the model was good, we used the real-time electric field model for the year 2014 in the absence of $\triangle H$ data to observed the possible relation between electric field and irregularities/TEC gradient in the post-sunset period. Actually, the two selected years have a different solar condition. However, the average value of F10.7cm for the quiet days in the two years does not show significant variation (Page 8, Figure 1b).
**Listing of technical corrections**

**Abstract**

1. **Line 9: Change correlation to relation. I did not see any correlation study between both variables in this work.**

   **Response**: Corrected (Page 1, Line 9).

2. **Line 11: Maximum positive/depletions. Why not use maximum enhancement and reduction. The spatial gradient will either be positive or negative. A negative gradient means reduction in electron density. Let's avoid using the word depletion since it can be mistaken for TEC depletion.**

   **Response**: Corrected (Words and phrases stated maximum positive/depletions was corrected to maximum enhancement and reduction). (Page 1, Line 10).

3. **Line 15-16: The spatial gradient of TEC between the two nearby stations could be used as an indicator of the occurrence of ionospheric irregularities. Is it over both stations or it is a general statement?**

   **Response**: Over both stations (We did correlation for both stations) (Page 1, Line 15 - 18).

   **Introduction**

   **Page 2**

4. **Line 8: Attests**

   **Response**: Corrected (Page 2, Line 7).

5. **Line 14. Remove mechanism**

   **Response**: Corrected (Page 2, Line 11).

6. **Line 18. ESF write in full. First time used.**

   **Response**: Corrected (Page 3, Line 2).

7. **Line 30. GPS write in full. The GPS scintillation index, S4 is not an instrument. The GPS is.**

   **Response**: Corrected (Page 1 , Line 5).

8. **Line 31. Global Navigation Satellite System (GNSS). Use either GNSS or GPS.**

   **Response**: Corrected (Page 2, Line 19).

   **Page 3**

9. **From lines 1-2, a mention of some work done over Africa has been made. However nothing was said about the scope of such studies, their limitations/gaps and how they relate to this study. Kindly address.**

   **Response**: Modified (Page 3 Lines 1- 10)

10. **Line 26. "and see". Change to as well as study.**

    **Response**: Corrected (Page 3 , Line 10 ).

11. **Line 27 "A closely found" change to closely located**

    **Response**: Corrected (Page 3, Line 14).

[Figure]

12. **Line 28. I am not comfortable with the word 'longitudinal'. Change to spatial for uniformity with title.**

    **Response**: Corrected (Page 3, Line 16).

13. **Line 27-28. What is the justification for the study of the relation between longitudinal (in this case spatial) gradient of TEC derived from two GPS receivers and occurrence of ionospheric irregularities still using GPS?**

    **Response**: Modified (Page 3 Lines 1-23)

14. **Line 29. Same as in line 28.**

    **Response**: Corrected.

    **Page 4**

    **Data and methods**

15. **Line 2-3: Kindly read that statements and adjust for easy flow.**

    **Response**: Corrected (Page 3, Lines 25-33; Page 4 , Lines 1-9).

16. **Line 5: Why 2014 only? Is there any particular justification for th choice of this year?**

    **Response**: It is because this year had sufficient simultaneous data for both stations for statistical values to be reliable.

17. **Line 6: Remove "of"**

    **Response**: corrected.

[Figure]

18. **Lines 7-8: Change the first average to "mean".**

    **Response**: Corrected.

19. **Line 10: change were to "was".**

    **Response**: Corrected.

20. **Line 10: "then analyzed to show the possible indicator of". This statement is not correct. Adjust**

    **Response**: Adjusted.

21. **Line 11: "The spatial gradient of TEC between the two nearby stations are located nearly along the same". Adjust statement. May be you should delete "are".**

    **Response**: Corrected.

22. **Line 14: Any reference for equation 1?**

    **Response**: corrected (Page 4, Line 8).

23. **Line 16-17: I am not satisfied with your definition of $\Delta H$ the way it is and the way you associate it to the EEJ in these particular lines. In addition you need to add how the H was processed and corrected for baseline value and non cyclic variations**.

    **Response**: modified (Page 5, Lines 10-25; Page 6, Lines 1-14).

24. **Line 22: ...is a transfer function model which to models the daily variations**...**Check the sentence**.

    **Response**: corrected.

    **Page 5**

25. **Line 1: "which are mapped from interplanetary electric field (IEF) data". Change to ... which are mapped in the interplanetary electric field (IEF).**

 **Response**: corrected.

26. **Line 5: I think you need to clearly explain the various options that the model provides and then proceed to tell us exactly which of the three options you used and why.**

 **Response**: corrected.

27. **Line 16: add "s" to station.**

 **Response**: Corrected.

28. **Line 23: Put a comma after reliable.**

 **Response**: Corrected.

29. **Line 24: Change "from the model" to 'it'**

 **Response**: corrected.

30. **Line 32: Was ROTI introduced to quantify the ROT measurements or ionospheric irregularities? Clarify please.**

 **Response**: modified.

 **Page 6**

31. **Line 6: Adjust to (Ma and Maruyama, 2006).**

 **Response**: adjusted.

32. **Line 23: I thought the scope of the study was 2014. Why use data from 2012? Have you accounted for the yearly variation and solar activity influence in juxtaposing your 2012 and 2014 data? Please could you clarify this?**

**Response**:
Since magnetometer data during the year 2012 was available simultaneously both at dip equator (Addis Ababa) and off-equator (Adigrat) we used these data to observe the performance of equatorial electric field model over equatorial region of Africa. The model correlates moderately over equatorial region of Africa, and hence we used the model over the region, during the year 2014 (Page 5, 6 ).

**Results and Discussions**

33. **Line 22: For some of selected. Remove of**

**Response**: corrected

34. **Lines 22-23. How did you select those quiet days? How many where there? What is the temporal resolution of both $\Delta H$ and EEF. Is the correlation obtained from "some selected quiet days" be an adequate representation of all other quiet days in year 2012? Check Figure caption in Figure 1 and harmonize. Let's know whether you use some days or quiet days of month of year 2012.**

**Response**:
The quiet days used in this study are the international five quiet days of each month of year 2014 obtained from Kyoto website. Only 38 quiet days available. The temporal resolution of $\Delta H$ is 1 min and EEF is 5 min. We corrected the resolution of $\Delta H$ to 5 min resolution. We modified the

correlation between the EEF and $\Delta H$ using the quiet days of year 2012. The graph is replotted. Caption of Figure 1 is modified (Page 8, Figure 1).

**Page 7**

35. **Lines 2-5. You gave us a beautiful description of how $\Delta H$ can be derived between the two magnetometers just for you to come and tell us that the data were not available for year 2014. I think it should have been the other way round.**

    **Response**: Modified (Page 5, Lines 7-25; Page 6, Lines 1-11)

36. **Line 3. Correct Adegrat to Adigrat.**

    **Response**: corrected.

37. **Line 4-5. To solve this data gap we used the daytime information of equatorial electric field derived from the real-time prompt penetration electric field model as an option. Did you use the real time model of Ionospheric electric fields of the real-time prompt penetration electric field model?**

    **Response**: We have used the real time model of Ionospheric electric fields of the real-time prompt penetration electric field model.

38. **In Page 6 lines 21-23 the authors claimed the relation is for some selected quiet days of months of year 2012 but in Page 7, lines 5-7 they state that the same relation is for year 2012. This may be extremely misleading.**

    **Response**: The relation is modified by considering the international quietest days of months of year 2012.

39. **We need a clear explanation on how you wish to use correlation result for 2012 to support some of your results in 2014. You must be aware of the solar activity influence on vertical drift. It will not be totally accurate to say that the correlation in 2012 will be the same as those in 2014. May be you did that as an indication of something. You need to clarify.**

**Response**:
The correlation between the equatorial electric field model and $\Delta H$ in the year 2012 was made to observe the performance of the model over the equatorial region of Africa, East Africa. We choice the year 2012 just because of availability of magnetometer data at both stations (Addis Ababa and Adigrat). The performance of the model over this sector was moderate (C = 0.6).

**Page 8**

40. **Line 5: Be consistent. Is it Figure or Fig? (check in all texts and harmonize).**

**Response**: We used Figure/Fig ways in referring figures according to the style of the journal.

41. **Line 7. Replace "but lags' with " but after ''...**

**Response**: corrected.

42. **Line 8: The depletions in the gradient. I am not comfortable with the word depletion. Kindly use the reduction in the gradient.**

**Response**: modified.

43. **Line 9: ...maximum positive of the spatial gradient of ... Replace with " the peak of spatial gradient" or "the maximum spatial gradient".**

**Response**: corrected.

44. **Line 11: Change depletion with reduction.**

**Response**: corrected.

45. **Lines 16-19: Why over Asab only?**

**Response**: Graph of Debark also added (Page 11 and 12, Figure 3 and 4).

**Page 9**

46. **30 March 2014, 10 April 2014, 20 September 2014, 10 October 2014. Where these days selected randomly?**

**Response**:
Over equatorial/low-latitude regions of Africa, since a high rate of occurrence of irregularities observed in the equinoctial seasons, and we selected the equinoctial seasons of the year 2014. And, these days are quiet days of the selected months.

47. **The caption in Figure 3 should be self explanatory and should tell us the stations (Asab and Debark) that were used for the ROTI.**

**Response**: The caption in Figure 3 modified (Page 11, Figure 3).

**Page 10**

48. **Line 1: Do you have any reference for this?**

**Response**: Yes (Page 9, Line 31).

49. **Line 1: From the figure, ... Which Figure? Specify.**

    **Response**: corrected.

50. **Lines 2-3: A convincing and quantitative way to demonstrate infer-
    ences in lines 3-4 is by performing correlation between spatial gradi-
    ent and irregularities.**

    **Response**: Modified and plotted (Page 14, Figure 6).

51. **Lines 7-8: An ionosphere gradient of 518 mm/km was discovered,
    generated by a plasma bubble. Read the statement and rephrase.**

    **Response**: Adjusted.

52. **Line 14: (see., Fig. 5). Change to as seen in Figure 5.**

    **Response**: corrected.

53. **Line 17: Change "a" by "the"**

    **Response**: corrected.

54. **Line 18: Change "indicates" to "shows"**

    **Response**: corrected.

55. **Line 19: ...in section (2)... which section 2? Change to as stated
    earlier.**

    **Response**: corrected.

56. **Line 23: Put 's' to period**

    **Response**: corrected.

57. **Line 23-24: Equation (1) was applied to all days of the year 2014? Including disturbed days? This is where it is important to separate disturbed days from quiet ones. We know that gradients can be significant during geomagnetic storms.**

    **Response**: The correlation between the TEC gradient and ROTI was done only for the quiet days (Page 11, Figure 3; Page 14, Figure 6).

58. **Line 28 - 32: Most of the observed features have not been discussed and plausible answers not given to explain them.**

    **Response**: Modified (Page 13, Lines 7-13, Page 14, Lines 1-13)

59. **Line 32: Change depletions to reductions.**

    **Response**: corrected.

60. **Figure 4: a) Diurnal variation of the spatial gradient of TEC over ASAB and DEBK , b)Daily maximum value of the spatial gradient of TEC variation, c) Diurnal variation of ROTIave over ASAB station and d) Daily maximum value variation of ROTIaave over ASAB station in the year 2014. Check this Figure caption and adjust according to your Figures (e) and (f) are missing.**

    **Response**:: Figures captions are adjusted (Page 13, Figure 5).

    **Page 11**

61. **Lines 1-2. If you can show it don't say it.**

    **Response**:: Modified (Page 13, Figure 5).

62. **Lines 10-11. "The trend they show has similarity with" The trend is already a similarity. Adjust the statement.**

**Response**: Adjusted.

63. **The caption of Figure 4 is misleading. Please check and let it conform with what you have in the texts.**

   **Response**: The caption of the figure is adjusted (Page 13, Figure 5).

64. **Why not add a correlation plot between spatial gradient and ROTI over each station? This is a better way of obtaining quantitative information between both quantities.**

   **Response**: Correlation plot between spatial gradient and ROTI over each stations are plotted (Page 14, Figure 6).

   **Page 12**

65. **Line 3: What about Debark? Why is it not presented? Besides, is this Figure for quiet and disturbed periods? How did you segregate the effect of transient disturbances?**

   **Response**:
   Percentage of occurrence of ionospheric irregularities for Debark is plotted (Page 16, Figure 7). The figures are both for quiet and disturbed periods. We didn't segregate the effect of transient disturbances, since it is not the objective of the study. This may be worked in the future.

66. **Line 26: Basu et al., the year is missing.**

   **Response**: corrected.

   **Page 13**

67. **Line 6: Change "has not been seen" to something suitable.**

    **Response**: corrected (Page 16, Line 15).

    **Page 14**

    **Conclusions**

68. **Lines 2-3: This is inconclusive and cannot feature in this section given the fact the relation between EEF and TEC gradient was investigated for just for 4 days (Figure 2).**

    **Response**: Modified. Quiet-monthly mean of EEF and TEC gradient were used to observe the relation between them (Page 10, Figure 2).

    **Acknowledgments**

69. **Line 6: Remove and.**

    **Response**: corrected.

70. **Lines 7-8: We acknowledge http://www.geomag.org/models/PPEFM/ RealtimeEF.html for providing the data the Prompt penetration equatorial electric field model. Give proper acknowledgement please.**

    **Response**: acknowledged correctly (Page 17, Lines 7-8).

71. **Line 8: Provide adequate acknowledgement for using the AMBER data (Visit AMBER website for adequate acknowledgement).**

    **Response**: acknowledged adequately (Page 17, Lines 8-9 ).

15    preceded by the rapid rise in F-layer and the strength of EEJ before sunset has been presented (Dabas et al., 2003; Burke et al., 2004; Kelley, 2009; Uemoto et al., 2010; Ram et al., 2007). Sreeja et al. (2009) reported observational evidence for the plausible linkage between the daytime EEJ related electric field variations with the postsunset F-region electrodynamics. Furthermore, Hajra et al. (2012) indicate that the afternoon/evening time variation of the eastward electric field as revealed through EEJ seems to play a dominant role in dictating postsunset resurgence of EIA and consequent generation of spread-F

20    irregularities. Even though ΔH and equatorial electric field derived model (EEF) do not correlate strongly enough over East Africa longitudinal sector, we might use the model derived equatorial electric field over equatorial/low-latitude region of Africa to explain some special features of ionospheric phenomena like plasma density irregularities and the enhancement/reduction of the spatial gradient of TEC between the two stations.

**3.2    Relation between the equatorial electric field (EEF), the spatial gradient of TEC and occurrence of ionospheric**

25             irregularities

Figure 2a-l show the diurnal variation of the quiet-monthly mean of EEF (along $\sim 40°$ E) and the quiet-monthly mean of the gradient of TEC in the year 2014. The red and blue curve indicate the EEF and the TEC gradient, respectively. The maximum enhancement/reduction in the gradient of TEC and the peak value in the EEF observed during the pre-midnight period was prevalent in the equinoctial months. After the post-sunset period, the maximum enhancement/reduction in the value of the

30    gradient of TEC on solstice months was small compared to equinoctial months. The EEF start rising nearly after 16:00 LT and show enhancement in the evening local time ($\sim$18:00 LT). Most of the maximum enhancement/reduction of the gradient of TEC was observed in the pre-midnight (19:00 - 24:00 LT) and postmidnight (24:00 - 06:00 LT) but after 1-2 hr of the post-sunset enhancement of the equatorial electric field. However, the spatial gradient of TEC observed during the daytime

[Figure]

**Figure 1.** (a) Examples showing the diurnal variation of Equatorial Electric field Model (EEF) (Red panel) and $\Delta H$ (blue panel) during 26 March 2012 and (b) The correlation between the equatorial electrojet (EEJ) and quiet-time equatorial electric field model (EEF) during day-time (07:00-17:00 LT) period for quiet days of year 2012. The red line shows the linear fit of data points.

was relatively small compared to the evening time hours for most of the days. The maximum enhancement/reduction in the gradient of TEC observed in the evening period could be related to the pre-reversal enhancement in a zonal electric field.

Figures 3a - d indicate examples showing the diurnal variations of the spatial gradient of TEC and ROTI over ASAB and DEBK stations. The blue, black and the red curves, respectively, indicate the spatial gradient of TEC, and the phase fluctuation index (ROTI) over Asab and Debark. In the post-sunset hours, after 18:00 LT, the pattern of ROTI observed in both stations and the gradient of TEC show a similar trend. The enhancement in the gradient of TEC and the occurrence of irregularities in the post-sunset period could be explained by the presence of ionospheric electrodynamics. The post-sunset period electrodynamics is influenced by F-region dynamo which is governed by a longitudinal gradient of the electrical conductivity and thermospheric zonal wind (Crain et al., 1993). Anderson et al. (2004) showed that the scintillation activity is related to the maximum $E \times B$ drift velocity between 18:30 and 19:00 LT. Mendillo et al. (2001) have pointed out that the best available precursor for pre-midnight ESF is the EIA strength at sunset, which in turn influenced by the magnitude of PRE. Using differential TEC profiles, TEC (at 18:00 hr) - TEC (at 20:00 hr), Valladares et al. (2004) explained that the PRE of the vertical drift would reenergize the fountain effect.

It has been reported that the eastward component of electric field manifested by the vertical plasma drifts over equator and intensified around/shortly after sunset before reversing to westward is one of the most important parameters responsible

[Figure]

**Figure 2.** Comparison of Quiet-Monthly Mean of EEF derived from real-time electric field model at about ($\sim 40^\circ$ E) and spatial gradient of TEC between ASAB and DEBK in the year 2014.

for driving many interesting ionospheric phenomena, like the Appleton density anomaly and the stability of the nighttime ionosphere (e.g., Horvath and Essex, 2003; Abadi et al., 2015). In the evening sectors, the vertical drift enhancement is of particular significance as it is the major drivers for the generation of ESF (Farley et al., 1970; Woodman, 1970; Basu et al., 1996; Fejer et al., 1999; Martinis et al., 2005). Tulasi Ram et al. (2006) reported that the rapid enhancement of post-sunset of
5  the zonal electric field leads to a large vertical plasma drift ($\boldsymbol{E} \times \boldsymbol{B}$), thereby lifting the F-layer to higher altitudes resulting in a condition conducive for the generation of ESF. Vertical drifts are taken as the key parameters determining the dynamics of ionospheric F-region, and the occurrence of ESF. Ionospheric irregularities are mostly observed over equatorial/low-latitude region an hour or two hours after the PRE. Rastogi and Woodman (1978) showed ESF can appear at any time of the night other than the post-sunset period following the abnormal reversal of the vertical F-region drifts to an upward direction, with
10  a delay of about 1-2 hr. From the figure, the post-sunset enhancement in the zonal electric field as shown in the equatorial electric field model has a profound effect on the enhancement of the spatial gradient of TEC during the post-sunset period. This might indicate that the spatial gradient of TEC as an indicator of the occurrence of ionospheric irregularities. Different researchers used the concept of ionosphere spatial gradient based on multi-GNSS observations within a small scale region to provide corrections and integrity information to the Ground-Based Augmentation System (GBAS) and they attribute the large

[Figure]

**Figure 3.** An example showing diurnal variation of the patterns of the spatial gradient of TEC and the ROTI over ASAB and DEBK stations on a) 30 March 2014, b) 10 April 2014, c) 20 September 2014 and b) 10 October 2014.

ionosphere spatial gradient to the TEC enhancements and the ionosphere irregularities (Rungraengwajiake et al., 2015; Saito and Yoshihara, 2017). Saito and Yoshihara (2017) reported that ionosphere gradient can be generated by plasma bubbles.

The patterns of the spatial gradient of TEC and ROTI during geomagnetic storm days were illustrated in Fig. 4a-d. These storm days are categorized as moderate magnetic storms (-100 nT $\leq$ Dst $\leq$-50 nT) (Echer et al., 2013). When the presence of
5   ionospheric irregularities are observed (ROTIave$\geq$ 0.4 TECU/min), the magnitude of spatial gradient of TEC in the post-sunset period is enhanced (during 13 September) and when the presence of ionospheric irregularities are suppressed (ROTIave$\leq$ 0.4 TECU/min), the magnitude of TEC gradient in the nighttime period is reduced (19 February and 27 August). During these seasons, however, the magnitude of TEC gradient shows enhancement in the daytime. Using phase fluctuation index (fp), Kassa and Damtie (2017) reported the inhibition effect of a storm of 19 September on the occurrence of ionospheric irregularities
10  over Ethiopia.

Figures 5a and b show ionospheric irregularity occurrence using the phase fluctuation index ($ROTI_{ave}$) at ASAB and DEBK in the year 2014, respectively. The $ROTI_{ave}$ values are indicated in the color scale. The occurrence of ionospheric irregularities at the two stations, as indicated by $ROTI_{ave}$ value, is a post-sunset phenomenon. The implication of this is that the large-scale

[Figure]

**Figure 4.** An example showing diurnal variation of the patterns of the spatial gradient of TEC and the ROTI over ASAB and DEBK stations during geomagnetic storm days (a) 19 February 2014 (b) 12 April 2014 (c) 27 August 2014 and (d) 13 September 2014.

ionospheric irregularities, which are responsible for the scintillation of trans-ionospheric signals at GNSS frequencies, are more pronounced during post-sunset hours. The observed phase fluctuation shows monthly variations and there is also a seasonal trend in the occurrence of ionospheric irregularity (see., Fig. 7). Maximum irregularities are observed in March equinox months and minimum in June/July. During this period, the occurrence of phase fluctuation showing irregularities is observed mainly

5  between 19:00 LT and 24:00 LT. As stated by Oladipo and Schuler (2013b), the value of $ROTI_{ave} \geq 0.4$ shows the presence of ionospheric irregularity.

Figure 5c shows the annual variation of the spatial gradient of TEC between DEBK and ASAB in the year 2014. In the computation of the spatial gradient of TEC (using Eq. 1), negative/positive values in the gradient of TEC were observed. The negative/positive values of TEC gradient depends on the value of TEC at the reference station and it denotes an increase/a

10  decrease in the value of TEC gradient. The maximum enhancement and reduction in the value of the gradient of TEC were observed mostly during the post-sunset (18:00 - 24:00 LT) and postmidnight (24:00 - 06:00 LT) period. Equation (1) was applied to all days (364 days) of year 2014 in computing the spatial gradient of TEC. Out of the total observed daily maximum value of the gradient of TEC, about 194 days (in percent about 53 %) of them fall in this time period. There was also a case

where the maximum enhancements and reductions in the value of the gradient of TEC were observed in the early morning period.

[Figure]

**Figure 5.** Ionospheric irregularity occurrence at (a) Asab (ASAB), Eritrea (b) Debark (DEBK), Ethiopia. (c) spatial gradient of TEC between ASAB and DEBK, in year 2014. Daily maximum value of $ROTI_{ave}$ (d) over ASAB, (e) over DEBK and (f) Daily maximum (blue) and minimum (red) values of spatial gradient of TEC, in year 2014. The $ROTI_{ave}$ value in TECU/min is indicated in color code.

The maximum enhancement/reduction in the gradient of TEC observed during post-sunset and postmidnight period showing monthly and seasonal variations are the other features noticed from Figs. 5c and f. The maximum peak/reduced values of the spatial gradient of TEC observed in equinoctial months are greater than solstice months. Equinoctial asymmetry in the occurrence of TEC gradient was also noticed. TEC gradient in March equinox was greater than that in September equinox. The minimum values in the spatial gradient of TEC observed mostly during post-sunset and post-midnight periods show the same trend with that of the maximum enhancement in the gradient of TEC. They also show an equinoctial asymmetry, TEC gradient in March equinoxes were greater than the one obtained for September equinoxes.

Figures 5d and e present the daily maximum values of the phase fluctuation index ($ROTI_{ave}$) over ASAB and DEBK stations, respectively and Fig. 5f shows the daily maximum and minimum value of the gradient of TEC in the year 2014. It is clearly observed from the figures that the enhancement in $ROTI_{ave}$ and gradient of TEC shows monthly and seasonal

variations, and also equinoctial asymmetry is observed. The daily maximum value of the spatial gradient of TEC between the two stations shows similar trends with the daily maximum value of $ROTI_{ave}$ observed over ASAB and DEBK stations. The trend they show has similarity with the time of occurrence of maximum enhancement, monthly and seasonal variations. Moreover, the seasonal variation observed in both variables exhibits equinoctial asymmetry, where the March equinox was greater than September equinoxes. The mechanism of generation of the enhancement in vertical drift just after sunset was detailed by Farley et al. (2008). The magnitude of peak vertical drift is known to control the seasonal and day-to-day variations in the occurrence of equatorial spread F (Manju et al., 2009; Tulasi Ram et al., 2006).

Figure. 6 shows the relation between the spatial gradient of TEC and ROTI over ASAB and DEBK computed in the time period of 19:00-24:00 LT for geomagnetic quiet days of the year 2014. For every quiet day of the year 2014, an average of $\Delta TEC/\Delta lon$ and average of ROTI in the time period between 19:00 LT and 24:00 LT were computed. In this time period, a peak value in TEC gradient about 2.5 TECU/deg and ROTI about 1 TECU/min were observed. In the computation of TEC gradient, a positive and negative value in the TEC gradient were observed. The positive/negative in the TEC gradient means a maximum enhancement/reduced in the plasma density relative to each other. To analyze the effect of the horizontal gradient of TEC on navigation and communication during geomagnetic storms, Radicella et al. (2004) considered the absolute value of TEC gradient. Here, we took the absolute value of the spatial gradient of TEC to describe the relationship between the TEC gradient and ROTI. The correlation coefficient between $\Delta TEC/\Delta lon$ and ROTI is about 0.58 (in ASAB) and 0.53 (in DEBK), respectively. It is well known that ROT is the combination of the spatial and temporal gradients. However, by giving less attention to the spatial gradient effect, previous authors often use $\Delta TEC/\Delta t$ to explain the TEC fluctuation determination. It is not only the temporal variation of TEC that contribute to the fluctuation in the phase and amplitude of the signals but also the spatial gradient of TEC. The computed correlation coefficient between the TEC gradient and ROTI, here, gives an indication of the contribution of the spatial gradient of TEC to ROTI (or ROT) usage. This can give the case where the spatial gradient of TEC between two nearby located stations used as an indicator of occurrence of ionospheric irregularities. Cesaroni et al. (2015) described the importance of the information provided by the TEC gradients variability and the role of the meridional TEC gradients in driving scintillation. By comparing the zonal and the meridional components of average and standard deviation of $\Delta TEC$, Cesaroni et al. (2015) reported that the North-South (N-S) gradients of TEC are significantly larger than their East-West (E-W) counterparts, regardless of the season. Saito and Yoshihara (2017) associated extreme ionospheric total electron gradient with plasma bubbles for GNSS Ground-Based Augmentation System and they obtained a largest ionospheric gradient of about 3.38 TECU/km.

Figure 7 presents the percentage occurrence of ionospheric irregularities over ASAB (blue) and DEBK (red) in the year 2014. The observation of this percentage occurrence is for all days of the year 2014 including both quiet and disturbed periods. The percentage occurrence of irregularities was calculated by counting the number of days in a month with $ROTI_{ave} \geq 0.4$ TECU/min and dividing by the number of days in a month for which data are available, and multiplied by 100 % (Oladipo et al., 2014). The two peaks of irregularity occurrence were observed around the middle of the equinoxes (i.e., in March and September) at both stations. This could be related to the alignment of the magnetic field lines with a geographic meridian (Burke

[Figure]

**Figure 6.** Relation between the spatial gradient of TEC and ROTI index derived from TEC over ASAB (blue) and DEBK (red) during 19:00-24:00 LT period of the magnetic quiet days of year 2014. The black and magnta lines indicate the linear fit between the spatial gradient of TEC and ROTI for ASAB and DEBK, respectively.

[revised manuscript text omitted]

---

## Referee Comment (RC2) · Anonymous Referee #2 · 21 Mar 2019

Review of manuscript # angeo-2018-131 Title: Spatial gradient of total electron content (TEC) between two nearby stations as indicator of occurrence of ionospheric irregularity Author(s): Teshome Dugassa, John Bosco Habarulema, and Melessew Nigussie

The current manuscript is an attempt to look into the spatial gradient of TEC between two nearby low-latitude GPS stations separated by 5 degrees in longitude, as a proxy for ionospheric irregularities. I appreciate the idea behind the study. The authors compare the spatial gradient to ROTI-index which is considered a standard now. However, authors have not been able to prove that the spatial gradient actually manifests ionospheric irregularities. It needs a bit more work and a lot of clarity. So I would

recommend the paper for a major revision. Below are detailed comments.

1. Figures 2 and 3 needs a bit more clarification. I understand that the blue curves in Figure 3 shows the spatial gradient and the blue curve in Figure 2 shows enhancement in spatial gradient. What does the authors mean by enhancement/depletion in spatial gradient? Please mention how the enhancement/depletion in spatial gradient is calculated in Figure 2. I am just confused if the blue line in Figure 2 represent spatial gradient in TEC or enhancement in spatial gradient in TEC. The two sentences in lines 3-4 on page 8 are cofusing. Considering Figures 2 and 3 are the most important figures of this manuscript they must be explained properly. 2. Page 8 Line # 9-10: The authors state: "The spatial gradient of TEC observed during the day time was relatively small compared to the evening time values for most of the days." If you take a look at Figure 3c, the daytime peak and the post sunset peak of the spatial gradient are almost of same values.

3. Also the authors need to explain what do the increase or decrease in the spatial gradient in TEC mean physically. For example, in case of ROTI, it is very straight forward. If you look at Figure 3, the day time ROTI stays around zero. Postsunset, the ROTI values increase showing fluctuations in TEC (hence density). In case of ROTI, a standard value of 0.5 is considered to identify ionospheric irregularities.

However, in case of spatial gradient of TEC, you can see daytime fluctuations as well. So what you see postsunset, may not entirely be due to fluctuations in TEC or irregularities. It may have a significant contribution due to zonal plasma drift. How do you eliminate that possibility?

4. If you look at Figures 3b and 3d, the peaks of the red and blue curve matches, but their values say a different story. The value of ROTI-index in Figure 3b is higher than that in Figure 3d. But the values of the spatial gradient show an opposite trend. The value of spatial gradient in Figure 3b is lower than that in Figure 3d. So in terms of ROTI, the ionospheric scintillation is stringer in Figure 3b than 3d. But in terms of

spatial gradient, it looks opposite. How to explain it.

5. Figure 3(c): It has almost similar +ve and -ve phases around 16-22 hours. What do the negative phase of the spatial gradient mean physically?

6. The authors here have just shown 4 cases where ionospheric irregularities were present. They also need to show cases when there were no ionospheric irregularities.

7. Figure 4 (a-c): The authors need to modify the color codes of the three figures to make it clear. Right now, the minimum limit of ROTI in Figure 4(a-b) is set at 0.5 which is considered as the onset of ionospheric irregularities. Set it to zero so that a clear picture can be seen.

8. The manuscript do not show any evidence yet, to prove the spatial gradient can be used as an indicator of ionospheric irregularities. It is in a stage where it actually investigates the relationship between spatial gradient of TEC and ionospheric irregularities. So I will suggest the authors suitably change the title of the manuscript. Technical:

9. Page 8: line 3: Figure 2a and d should be Figure 2 (a-d).

---

## Author Comment (AC2) · 12 Apr 2019

**Response to reviewers' comments on the manuscript angeo-2018-131-RC2: Spatial gradient of total electron content (TEC) between two nearby stations as indicator of occurrence of ionospheric irregularity**

Authors: Teshome Dugassa, John Bosco Habarulema, and Melessew Nigussie

The authors thank the reviewer for his/her comments that helped to improve the quality of this research. The comments are addressed as shown below.

**Reviewer 2**

[Figure]

**1. Figures 2 and 3 needs a bit more clarification. I understand that the blue curves in Figure 3 shows the spatial gradient and the blue curve in Figure 2 shows enhancement in spatial gradient. What does the authors mean by enhancement/depletion in spatial gradient? Please mention how the enhancement/depletion in spatial gradient is calculated in Figure 2. I am just confused if the blue line in Figure 2 represent spatial gradient in TEC or enhancement in spatial gradient in TEC. The two sentences in lines 3-4 on page 8 are confusing. Considering Figures 2 and 3 are the most important figures of this manuscript they must be explained properly.**

**Response:**

In both of the graphs (Figures 2 and 3), the blue curves show the diurnal variation in the gradient of TEC. In Figure 2, the authors try to relate the electric field derived from equatorial electric field model and the spatial gradient in TEC. Whereas, Figure 3 was presented to show the trends of the time variation of ROTI and the gradient in TEC. In both figures, the maximum enhancement/reduction in the gradient of TEC was observed.

On the computation of the spatial gradient of TEC between the two nearby located stations (ASAB and DEBK), the gradient of TEC may be either positive/negative value. Both the negative and positive values obtained from the differences of TEC between the two stations show the gradient of TEC. It is obvious that the positive/negative gradient in TEC is obtained when the minuend is larger/smaller than the subtrahend. A positive gradient in TEC denotes a higher in TEC/ electron density over ASAB relative to DEBK and a negative gradient in TEC indicates a lower in TEC/electron density over ASAB than DEBK, i.e, a higher in TEC/electron density over DEBK than ASAB.

In the manuscript, we have used the term maximum enhancement /reduction in the gradient of TEC (in terms of magnitude) when the nighttime value of gradient of TEC was larger than the daytime value. Since fluctuation in ionospheric plasma density was

prevalent during the nighttime hours, we have used the state of the night time gradient of TEC compared to daytime to describe the relationship between the gradient in TEC and the occurrence ionospheric irregularities.

The authors modified the term enhancement/depletion in the spatial gradient of TEC to maximum enhancement/reduction in the gradient of TEC. In the current study, there was no quantitative method that we have used to calculate the maximum enhancement/reduction in the gradient of TEC. We rather described the state of the gradient of TEC (which shows maximum positive/negative in TEC/electron density during nighttime relative to the daytime) by observing the values of gradient of TEC. The gradient in TEC was calculated by taking the difference of TEC between two stations (ASAB and DEBK) and divided by their longitudinal differences. If the gradient of TEC during nighttime is positive/ negative values, the state of gradient of TEC show enhancement/reduction in gradient of TEC. (Page 4, 9).

**2. Page 8 Line 9-10: The authors state: "The spatial gradient of TEC observed during the day time was relatively small compared to the evening time values for most of the days." If you take a look at Figure 3c, the daytime peak and the post sunset peak of the spatial gradient are almost of same values.**

**Response:**

The statement was modified as, "most of the day time value of spatial gradient in TEC was relatively small in magnitude compared to the night values."

**3. Also the authors need to explain what do the increase or decrease in the spatial gradient in TEC mean physically. For example, in case of ROTI, it is very straight forward. If you look at Figure 3, the day time ROTI stays around zero. Postsunset, the ROTI values increase showing fluctuations in TEC (hence density). In case of ROTI, a standard value of 0.5 is considered to identify ionospheric irregularities.**

**However, in case of spatial gradient of TEC, you can see daytime fluctuations as well. So what you see postsunset, may not entirely be due to fluctuations in TEC or irregularities. It may have a significant contribution due to zonal plasma drift. How do you eliminate that possibility?**

**Response:**

The increase/decrease in the spatial gradient in TEC shows the difference in the value of TEC/plasma density observed over the stations. The increase/decrease in the gradient in TEC indicates the higher/smaller in TEC/plasma density over ASAB relative to DEBK. The maximum enhancement/reduction in the gradient of TEC mostly observed in the nighttime period. Even though the two stations were located near each other, they do exhibit difference in TEC/plasma density. The difference might be attributed to zonal thermospheric wind and ExB drift. (Page 10-11)

At a basic level, there are two independent ways of estimating the TEC gradient values using ground based GPS receiver data (Lee et al., 2006). The first method uses a pair of closely-spaced receiver stations. The second method uses a single GPS receiver station to infer the spatial TEC gradient values based on the observed temporal rate of change in TEC. The two methods actually have their own merit and demerits. In our study, we have used the first method (i.e., station-pair method). This station-pair method gives us an instantaneous estimate of the TEC gradient along a fixed direction determined by the line segment connecting the two stations. It is true that the zonal drift might have an effect on TEC gradient. As the two stations used to get TEC gradient are close to each other, the same irregularity have a chance to be observed at these stations almost at the same time and hence difference of TEC at these stations can eliminate the contribution of zonally drifting irregularity on TEC gradient variation. However, in case of the single station-method we must note that the background ionospheric plasma drift/circulation speed can potentially inflate as well as deflate the estimated TEC gradient values obtained (Datta-Barua et al., 2010).

**4. If you look at Figures 3b and 3d, the peaks of the red and blue curve matches, but their values say a different story. The value of ROTI-index in Figure 3b is higher than that in Figure 3d. But the values of the spatial gradient show an opposite trend. The value of spatial gradient in Figure 3b is lower than that in Figure 3d. So in terms of ROTI, the ionospheric scintillation is stringer in Figure 3b than 3d. But in terms of spatial gradient, it looks opposite. How to explain it.**

**Response:**

The values of ROTI in each panel of Figure 3 (a-d), shown as representative cases, was greater than 0.5 TECU/min, a threshold value showing the presence of irregularities. However, they show difference in the strength of irregularities. These difference could be attributed to the difference in the strength of PRE, and other factors. Similarly, the value of the spatial gradient in TEC was different. For example, the value of the spatial gradient in TEC in Figure 3d is higher than that in Figure 3b, while ROTI value in Figure 3b is higher than that in Figure 3d. This might indicate the difference in the cause of the maximum enhancement/reduction in the gradient in TEC and ROTI, for example zonal neutral wind and ExB drift. (Page 10-11).

**5. Figure 3(c): It has almost similar +ve and -ve phases around 16-22 hours. What do the negative phase of the spatial gradient mean physically?**

**Response:**

When we apply Equation 1, we might obtain positive and/or negative values in the gradient of TEC, mostly around 16-22 hrs. Both the positive and negative phase indicates the gradient in TEC. These values are the relative TEC/plasma density between the two stations. The negative value in gradient of TEC indicates the higher TEC/plasma density over DEBK relative to ASAB, i.e., a reduction in TEC/plasma density over ASAB, showing a decrease in plasma density over ASAB compared to DEBK.

**6. The authors here have just shown 4 cases where ionospheric irregularities**

were present. **They also need to show cases when there were no ionospheric irregularities.**

**Response:**

Cases where absent in the occurrence of ionospheric irregularities were presented in the modified manuscript in Figure 4 (a-d). (Page 13)

**7. Figure 4 (a-c): The authors need to modify the color codes of the three figures to make it clear. Right now, the minimum limit of ROTI in Figure 4(a-b) is set at 0.5 which is considered as the onset of ionospheric irregularities. Set it to zero so that a clear picture can be seen.**

**Response:**

The color codes of Figure 4 (a-c) given in the manuscript were modified, and we can find in the revised manuscript in Figure 5 (a-c). (Page 15)

**8. The manuscript do not show any evidence yet, to prove the spatial gradient can be used as an indicator of ionospheric irregularities. It is in a stage where it actually investigates the relationship between spatial gradient of TEC and ionospheric irregularities. So I will suggest the authors suitably change the title of the manuscript. Technical:**

**Response:**

To show the relation between the spatial gradient in TEC and the occurrence of ionospheric irregularities over the equatorial region, the authors present the correlation between them as shown in Figure 7. In this study, the relation between the spatial gradient of TEC and ROTI was presented with correlation coefficients (C = 0.58 for DEBK, C = 0.53 for ASAB). We got hits that could lead us to relate the spatial gradient of TEC and the occurrence of ionospheric irregularities. To obtain the gradient of TEC we used a pair of closely-spaced receiver stations such that the two receivers share the same GPS satellite (less than $2°$) to calculate the difference in TEC values between the two

neighboring ionospheric piercing points (IPP) at any given time. In our case, however, the two stations are separated by 5o. The moderate correlation obtained might be attributed to the wider longitudinal separation $(5°)$ between the two stations. The other factor for the moderate correlation between the gradient of TEC and occurrence of ionospheric irregularities might be the way ROTI was computed (since ROTI contains both the spatial and temporal variation in TEC). If we can reduce the contribution of the temporal part of ROTI, we may get a better relationship. (Page 15, Lines 7- 16; Page 16 Lines 1:16 )

The title of the manuscript is modified as:

Investigation of the relationship between the spatial gradient of total electron content (TEC) and the occurrence of ionospheric irregularities.

9. Page 8: line 3: Figure 2a and d should be Figure 2 (a-d). **Response:**

We modified the way we referred the figures, as suggested.

Please also note the supplement to this comment:
https://www.ann-geophys-discuss.net/angeo-2018-131/angeo-2018-131-AC2-supplement.pdf

―――――――――――――――

**Supplement:**

**Investigation of the relationship between the spatial gradient of total electron content (TEC) and the occurrence of ionospheric irregularities**

Teshome Dugassa[1,2], John Bosco Habarulema[3,4], and Melessew Nigussie[5]

[revised manuscript text omitted]

15   index (ROTI). From the definition, however, ROTI mixes both the spatial and temporal gradients of TEC variations. The longitudinal gradient of integrated Pederson conductivity in the E-region at sunset time play a fundamental role in the strengthening the PRE magnitude and affect the generation of ionospheric irregularities (Tsunoda, 1985). It is well known that PRE is a postsunset phenomena which uplift the ionosphere and create a conducive condition for irregularity formation. This implies the magnitude of the zonal electric field is an important parameter for real-time prediction. It is also known that PRE is due to

20   spatial gradient of electron density near solar-terminator. However, it is not easy to obtain the longitudinal gradient of electron density over Africa longitude sector as ionosondes are not available in nearby locations and study the relationship between the electron density gradient and occurrence of ionospheric irregularities. We know TEC is the integral of electron density, so a closely located GPS receivers would help us to estimate the strength of the zonal electric field and investigate the relation between the gradient of TEC and occurrence of irregularities. In this study, the relationship between the spatial gradient of

[revised manuscript text omitted]
-d) indicate representative cases for ROTI (red and black curves) and spatial gradient in TEC (blue curve) when the occurrence of ionospheric irregularities are absent. The value of $ROTI_{ave}$ was less than 0.4 TECU/min, showing non-occurrence of ionospheric irregularities. In these cases, the maximum enhancement/reduction in spatial gradient of TEC were observed during day and/or nighttime period. When the occurrence of irregularities are absent, an enhancement/reduction in the gradient of TEC in the postsunset period also observed (Figure 4 a). The pattern in the spatial gradient of TEC and ROTI during geomagnetic storm days were also illustrated in Fig.5 (a-d). These storm days are categorized as moderate magnetic storms (-100 nT $\leq$ Dst $\leq$-50 nT) (Loewe and Prölss, 1997; Echer et al., 2013). When the presence of ionospheric irregularities

[Figure]

**Figure 3.** Typical examples of diurnal variation in the spatial gradient of TEC (blue curve) and the ROTI over ASAB (red curve) and DEBK (black curves) on a) 30 March 2014, b) 10 April 2014, c) 20 September 2014 and d) 10 October 2014.

are observed (ROTIave≥ 0.4 TECU/min), the magnitude of spatial gradient of TEC in the post-sunset period is enhanced (during 13 September 2014) and when its presence suppressed (ROTIave< 0.4 TECU/min), the magnitude of TEC gradient in the nighttime period is reduced (19 February and 27 August 2014). On those days, the spatial gradient in TEC observed during the daytime hours show maximum enhancement/reduction.

5    Figures 6a and b show ionospheric irregularity occurrence using the phase fluctuation index ($ROTI_{ave}$) at ASAB and DEBK in the year 2014, respectively. The $ROTI_{ave}$ values are indicated in the color bar. As stated by Oladipo and Schuler (2013b), the value of $ROTI_{ave} \geq 0.4$ shows the presence of ionospheric irregularity. The occurrence of ionospheric irregularities at the two stations, as indicated by intensity level of $ROTI_{ave}$, was predominantly observed in the premidnight periods, mainly between 19:00 LT and 24:00 LT. The large-scale ionospheric irregularities, which are responsible for the scintillation of trans-

10  ionospheric signals at GNSS frequencies, are more pronounced during post-sunset hours. The observed phase fluctuation shows monthly variations and there is also a seasonal trend in the occurrence of ionospheric irregularity. Strong and weak ionospheric irregularities are observed in March equinox and in June/July solstices, respectively.

[Figure]

**Figure 4.** Examples showing the absence of ionospheric irregularities observed using ROTI and its relation with spatial gradient of TEC on (a) 12 April 2014 (b) 13 April 2014 (c) 16 October 2014 and (d) 18 October 2014.

Figure 6c shows the annual variation in the spatial gradient of TEC between DEBK and ASAB in the year 2014. The maximum enhancement/reduction in the value of the gradient of TEC were observed mostly during the post-sunset (18:00 - 24:00 LT) and postmidnight (24:00 - 06:00 LT) period. Equation (1) was applied to all days (364 days) of year 2014 in computing the spatial gradient of TEC. Out of the total observed daily maximum value of the gradient of TEC, about 194 days

5  (in percent about 53 %) of them fall in this time period. There was also a case where the maximum enhancement and reduction in the value of the gradient of TEC were observed in the early morning period.

Figure 6c shows the daily maximum positive (blue dots) and maximum negative (red dots) in values of TEC gradient. The maximum enhancement/reduction in the gradient of TEC were mostly observed during post-sunset and postmidnight period. They also show monthly and seasonal variations. The maximum peak/reduced values of the spatial gradient of TEC observed

10  in equinoctial months are greater than solstice months. Equinoctial asymmetry in the occurrence of TEC gradient was also noticed. TEC gradient in March equinox was greater than that in September equinox. The minimum values in the spatial gradient of TEC observed mostly during post-sunset and post-midnight periods show similar trend with that of the maximum enhancement. They also show an equinoctial asymmetry, where TEC gradient in March equinoxes were greater than the one obtained for September equinoxes.

[Figure]

**Figure 5.** Representative examples showing diurnal variation of the patterns of the spatial gradient of TEC and the ROTI over ASAB and DEBK stations during geomagnetic storm days (a) 19 February 2014 (b) 12 April 2014 (c) 27 August 2014 and (d) 13 September 2014.

Figures 6d and e present the daily maximum values of the phase fluctuation index ($ROTI_{ave}$) over ASAB and DEBK stations, respectively and Fig. 6f shows the daily maximum and minimum value of the gradient of TEC in the year 2014. It is clearly observed from Figures 6 (d-f) that the enhancement in $ROTI_{ave}$ and gradient of TEC shows monthly and seasonal variations, and also equinoctial asymmetry is observed. The daily maximum value of the spatial gradient of TEC shows a kind
5   of similar trends with the daily maximum value of $ROTI_{ave}$ observed over ASAB and DEBK stations. The trend they show has similarity with the time of occurrence of maximum enhancement, monthly and seasonal variations. Moreover, the seasonal variation observed in both variables exhibits equinoctial asymmetry, where the March equinox was greater than September equinoxes. The mechanism of generation of the enhancement in vertical drift just after sunset was detailed by Farley et al. (2008). The magnitude of peak vertical drift is known to control the seasonal and day-to-day variations in the occurrence of
10   equatorial spread F (Manju et al., 2009; Tulasi Ram et al., 2006).

Figure. 7 shows the relation between the spatial gradient of TEC and ROTI (over ASAB and DEBK) computed in the time period of 19:00-24:00 LT for geomagnetic quiet days of the year 2014. For every quiet day of the year 2014, mean of $\Delta TEC/\Delta lon$ and mean of ROTI during the time period between 19:00 LT and 24:00 LT were computed. In this time period, peak in value of TEC gradient about 2.5 TECU/deg and ROTI about 1 TECU/min were observed. To analyze the effect of

[Figure]

**Figure 6.** Ionospheric irregularity occurrence at (a) Asab (ASAB), Eritrea (b) Debark (DEBK), Ethiopia. (c) spatial gradient of TEC between ASAB and DEBK, in year 2014. Daily maximum value of $ROTI_{ave}$ (d) over ASAB, (e) over DEBK and (f) Daily maximum (blue) and minimum (red) values of spatial gradient of TEC, in year 2014. The $ROTI_{ave}$ value in TECU/min is indicated in color code.

the horizontal gradient of TEC on navigation and communication during geomagnetic storms, Radicella et al. (2004) considered the absolute value of TEC gradient. Here, we took the absolute value of the spatial gradient of TEC to describe the relationship between the TEC gradient and ROTI. The correlation coefficient between $\Delta TEC/\Delta lon$ and ROTI is about 0.58 (in ASAB) and 0.53 (in DEBK), respectively. It is well known that ROT is the combination of the spatial and temporal gra-
5  dients. However, by giving less attention to the spatial gradient effect, previous authors often use $\Delta TEC/\Delta t$ to explain the TEC fluctuation determination. It is not only the temporal variation of TEC that contribute to the fluctuation in the phase and amplitude of the signals but also the spatial gradient of TEC. The computed correlation coefficient between the TEC gradient and ROTI, here, gives an indication of the contribution of the spatial gradient of TEC to ROTI (or ROT) usage. This can give the case where the spatial gradient of TEC between two nearby located stations can be used as an indicator of occurrence of
10  ionospheric irregularities. Every night time enhancement/reduction in the gradient of TEC may not be a guarantee to indicate the occurrence/non-occurrence of ionospheric irregularities. However, there are indications which shows the occurrence of irregularities over both stations (ASAB and DEBK) when the night time enhancement/reduction in the TEC gradient were observed. By comparing the ROTI index, ionospheric TEC gradient and vertical TEC, Hua and Chunbo (2009) reported the correlation between them and their variation characteristics resulted from plasma instability in the ionosphere. Cesaroni et al.
15  (2015) described the importance of the information provided by the TEC gradients variability and the role of the meridional TEC gradients in driving scintillation. By comparing the zonal and the meridional components of average and standard deviation of $\Delta TEC$, Cesaroni et al. (2015) reported that the North-South (N-S) gradients of TEC are significantly larger than their East-West (E-W) counterparts, regardless of the season. Saito and Yoshihara (2017) associated extreme ionospheric total electron gradient with plasma bubbles for GNSS Ground-Based Augmentation System and they obtained a largest ionospheric gradient of about 3.38 TECU/km. It is suggested that when scintillation events are investigated ionospheric TEC gradient is also one of considerable parameters.

[Figure]

**Figure 7.** Relation between the spatial gradient of TEC and ROTI index derived from TEC over ASAB (blue) and DEBK (red) during 19:00-24:00 LT period of the magnetic quiet days of year 2014. The black and magenta lines indicate the linear fit between the spatial gradient of TEC and ROTI for ASAB and DEBK, respectively.

[revised manuscript text omitted]

**4    Conclusions**

In this study, we present for the first time the relationship between the spatial gradient of TEC between two nearby located stations (ASAB and DEBK) and the occurrence of ionospheric irregularities over Ethiopia, an equatorial region, based on GPS-TEC data. The following features were observed in the study. The daytime equatorial electrojet (EEJ) derived from H-component of geomagnetic field and the real-time electric field (EEF) model correlates linearly and positively with correlation coefficient of C = 0.6. Most of the daily maximum enhancement and/or reduction in the spatial gradient of TEC was observed in the pre-midnight and post-midnight period. In terms of seasons and months, the nighttime pattern of most of the the spatial

[Figure]

**Figure 8.** Percentage of occurrence of ionospheric irregularities over ASAB (red) and DEBK (blue) stations in the year 2014 based on $ROTI_{ave}$ index.

gradient of TEC and $ROTI_{ave}$ show similar trends. Both of them show maximum enhancement/reduction during the March and September months. Equinoctial asymmetry was also observed both 
[revised manuscript text omitted]

---

## Author Response (AR1)

**Response to reviewers' comments on the manuscript angeo-2018-131-RC1: Spatial gradient of total electron content (TEC) between two nearby stations as indicator of occurrence of ionospheric irregularity**

Authors: Teshome Dugassa, John Bosco Habarulema, and Melessew Nigussie

**General remarks**

We thank the two referees for their comments and suggestions which have led to an improved, revised manuscript. Those comments are all valuable and very helpful for revising and improving our paper, as well as the important guiding significance to our research. We have reacted to the comments carefully and have made correction which we hope meet with approval. We revised the introduction. We added graphs and discussions. The responds to the comments are as flowing:

**Referee #1**

**General comments**

1. **The study attempts to show how the difference between the TEC of two close GNSS stations can be used as a precursor of ionospheric irregularities in the post sunset period over both stations. The study being the first of its kind in the African sector is worthy of interest couple with the fact that it is well written and it gives insight about a possible relation between electric field and irregularities in the post sunset period. However, the authors first need to give a good justification why they want to use TEC gradient between two stations as a proxy of irregularities.**

   **Response**:
   Yes, it's a good comment. We have revised the introduction of the manuscript and added the justification why we used TEC gradient between two nearby located stations as indicator of occurrence of ionospheric irregularities **(Page 2: Lines 23-34 and Page 3: Lines 1-33)**.

   Previous studies explained the relation between the latitudinal (N-S) gradient of TEC surrounding the anomaly region and ionospheric scintillation over different sectors (Muella et al., 2008). Even though, the characteristics of ionospheric irregularities/plasma bubbles over equatorial/low-latitude region of Africa under different solar and geomagnetic activities were discussed Mungufeni et al. (2016), a limited number of studies have been carried out over the region relating the latitudinal/longitudinal gradient of TEC/plasma density and the occurrence of ionospheric irregularities.

   The longitudinal gradient of integrated Pederson conductivity in the E-region at sunset time play a fundamental role in the strengthening the PRE magnitude and affect the generation of ionospheric irregularities (Tsunoda, 1985). It is well known that PRE is a postsunset phenomena which uplift the ionosphere and create a conducive condition for irregularity formation. This implies the magnitude of the zonal electric field is an important parameter for real-time prediction. It is also known that PRE is due to spatial gradient of electron density near solar-terminator. However, it is not easy to obtain the longitudinal gradient of electron density over Africa longitude sector as ionosondes are not available in nearby locations and study the relationship between the electron density gradient and occurrence of ionospheric irregularities. We know TEC is the integral of electron density, so a closely located GPS receivers would help us to estimate the strength of the zonal electric field and investigate the relation between the gradient of TEC and occurrence of irregularities.

2. **Giving the fact that ROTI which can be easily estimated is already an indicator of irregularities why use the gradient between two stations to do the same work?**

   **Response**:
   It is well known that ROTI is a proxy for the occurrence of ionospheric irregularities and scintillations. From the definition, however, ROTI mixes both the spatial and temporal gradients of TEC variations. The relationship between the spatial gradient of TEC and ROTI was established to observe how the gradient of TEC affect radio signals, presents information that the horizontal electron density gradient as an important parameter to predict ionospheric scintillation. After establishing this relationship showing that both parameters give information about ionospheric irregularities, TEC gradient method may be an alternative as it is a simple computation of establishing the difference **(Page 3: Lines 6-33)**.

3. **What if the constraints "over the same latitude, but separated by a longitudinal of about 5 degree" is removed? What happens to the relation?**

   **Response**:
   We thank the reviewer for the important point raised. We have a plan to investigate the optimum separation between two GNSS receivers that can provide the best correlation between TEC gradient and irregularity. This will be done in a separate work.

**Specific comments**

1. **How does the relationship between TEC gradient and ROTI is established? The study never really specified or took into consideration the quietness and/or disturbed nature of the days used.**

   **Response**:
   Thank you for your good suggestions. To show the relationship between the TEC gradient and ROTI, the 10 quiet international days in the year 2014 were used. In this study period, there are about 364 days fully available in both stations simultaneously. The quiet international days are obtained from http://wdc.kugi.kyoto-u.ac.jp/qddays/index.html. In total, about 120 quiet days were used in investigating the relationship between TEC gradient and ROTI. The maximum value of the standard deviation of the spatial gradient of TEC was used to observe the correlation between the gradient of TEC and ROTI **(Page 18, Figure 8)**. Moreover, in the revised version of the manuscript, disturbed days were also examined. **(Page 15, Figure 5)**.

2. **In establishing a relation between equatorial electric field and spatial gradient, the authors used only four days in year 2014. Is this sufficient enough to show any kind of relationship between both quantities?**

   **Response**:
   In the previous version of the manuscript, we used four quiet days as a typical example to observe a relation between equatorial electric field and spatial gradient of TEC which is not sufficient enough. In the revised version of the manuscript, however, we modified Figure 2 by adding more data/months to establish the relationship between the equatorial electric field and spatial gradient. In the updated graphs, we took the monthly-quiet mean of the equatorial electric field and TEC gradient of the year 2014 **(Page 11, Figure 2)**.

3. **In Justifying the magnetic data gap (the H component of the Earth's magnetic field) in 2014, the authors performed a correlation between the equatorial electrojet $\Delta H$ and equatorial electric field (EEF) for quiet days in 2012. No information is given on these days and how they were selected and how many they are?**

   **Response**:
   Five quietest international days of each month of the year 2014 were utilized to perform the correlation between the equatorial electrojet $\Delta H$ and the equatorial electric field (EEF). These days are obtained from http://isgi.unistra.fr/data_download.php. Out of 60 days, only 38 days of data were available simultaneously in both magnetometer measuring stations **(Page 9, Figure 1b)**.

4. **The authors assumed the correlation between $\Delta H$ and EEF as obtained in 2012 (0.7) will be the same in 2014.**

   **Response**:
   We did the correlation between $\Delta H$ and EEF to justify the performance and reliability of the model over the equatorial region of Africa using available magnetometer data (the year 2012). As can be seen from their relation, the day time EEF model and day time $\Delta H$ correlated positively. Since the performance of the model was good, we could use the real-time electric field model for the year 2014 in the absence of $\Delta H$ data to observed the possible relation between electric field and irregularities/TEC gradient in the post-sunset period. Actually, the two selected years have a different solar condition. However, the average value of F10.7cm for the quiet days in the two years does not show significant variation (**Page 9, Figure 1b**).

**Listing of technical corrections**

   **Abstract**

1. **Line 9: Change correlation to relation. I did not see any correlation study between both variables in this work.**

   **Response**: As suggested, we have changed the term "correlation" to "relationship" (**Page 1, Line 13**).

2. **Line 11: Maximum positive/depletions. Why not use maximum enhancement and reduction. The spatial gradient will either be positive or negative. A negative gradient means reduction in electron density. Let's avoid using the word depletion since it can be mistaken for TEC depletion.**

   **Response**: Ok, we replaced (Words and phrases stated maximum positive/depletions was corrected to maximum enhancement and reduction). (**Page 1, Line 7, 16**).

3. **Line 15-16: The spatial gradient of TEC between the two nearby stations could be used as an indicator of the occurrence of ionospheric irregularities. Is it over both stations or it is a general statement?**

   **Response**: Over both stations (We did correlation for both stations) (**Page 1, Line 13 - 15**).

   **Introduction**

   **Page 2**

4. **Line 8: Attests**

   **Response**: Modified.

5. **Line 14. Remove mechanism**

   **Response**: Corrected (**Page 2, Line 11**).

6. **Line 18. ESF write in full. First time used.**

   **Response**: We accept it and corrected in the revised manuscript (**Page 2, Line 25**).

7. **Line 30. GPS write in full. The GPS scintillation index, S4 is not an instrument. The GPS is.**

   **Response**: We accept it and corrected in the revised manuscript (**Page 3 , Line 12**).

8. **Line 31. Global Navigation Satellite System (GNSS). Use either GNSS or GPS.**

   **Response**: We accept it and corrected in revised manuscript (**Page 1, Line 5, 19**).

   **Page 3**

9. **From lines 1-2, a mention of some work done over Africa has been made. However nothing was said about the scope of such studies, their limitations/gaps and how they relate to this study. Kindly address.**

    **Response**: It is a good suggestion, the limitations/gaps of the previous study that has been done over Africa equatorial/low-latitude region and how the current study related to this study was discussed (**Page 3 Lines 6-20**).

10. **Line 26. "and see". Change to as well as study.**

    **Response**: Modified.

11. **Line 27 "A closely found" change to closely located**

    **Response**: corrected (**Page 3, Line 23**).

12. **Line 28. I am not comfortable with the word 'longitudinal'. Change to spatial for uniformity with title.**

    **Response**: As suggested we changed 'longitudinal' to 'Spatial' (**Page 3, Line 29**).

13. **Line 27-28. What is the justification for the study of the relation between longitudinal (in this case spatial) gradient of TEC derived from two GPS receivers and occurrence of ionospheric irregularities still using GPS?**

    **Response**: Modified (**Page 2, 23-35, Page 3 Lines 21-29, )**

14. **Line 29. Same as in line 28.**

    **Response**: Corrected.

    **Page 4**

    **Data and methods**

15. **Line 2-3: Kindly read that statements and adjust for easy flow.**

    **Response**: It is a good suggestions. We adjust the statement for easy flow (**Page 4, Lines 4-13**).

16. **Line 5: Why 2014 only? Is there any particular justification for th choice of this year?**

    **Response**: It is because this year had sufficient simultaneous data for both stations for statistical values to be reliable.

17. **Line 6: Remove "of"**

    **Response**: We corrected it as suggested.

18. **Lines 7-8: Change the first average to "mean".**

    **Response**: As suggested we changed the "average" to "mean".

19. **Line 10: change were to "was".**

    **Response**: We corrected.

20. **Line 10: "then analyzed to show the possible indicator of". This statement is not correct. Adjust**

    **Response**: We adjusted the statement.

21. **Line 11: "The spatial gradient of TEC between the two nearby stations are located nearly along the same". Adjust statement. May be you should delete "are".**

    **Response**: Corrected.

22. **Line 14: Any reference for equation 1?**

    **Response**: Yes, we have references for the equation **(Page 4, Line 26)**.

23. **Line 16-17: I am not satisfied with your definition of $\Delta H$ the way it is and the way you associate it to the EEJ in these particular lines. In addition you need to add how the H was processed and corrected for baseline value and non cyclic variations.**

    **Response**: Yes, it is a good suggestion. In the revised manuscript we modified how H was processed **(Page 5, 6, and 7)**.

24. **Line 22: ...is a transfer function model which to models the daily variations...Check the sentence.**

    **Response**: The sentence was modified.

    **Page 5**

25. **Line 1: "which are mapped from interplanetary electric field (IEF) data". Change to ... which are mapped in the interplanetary electric field (IEF).**

    **Response**: corrected.

26. **Line 5: I think you need to clearly explain the various options that the model provides and then proceed to tell us exactly which of the three options you used and why.**

    **Response**: corrected.

27. **Line 16: add "s" to station.**

    **Response**: Corrected.

28. **Line 23: Put a comma after reliable.**

    **Response**: Corrected.

29. **Line 24: Change "from the model" to 'it'**

    **Response**: As suggested we changed "from the model" to 'it' in the revised manuscript.

30. **Line 32: Was ROTI introduced to quantify the ROT measurements or ionospheric irregularities? Clarify please.**

    **Response**: It is a good comment. We clarified what ROTI indicates.

    **Page 6**

31. **Line 6: Adjust to (Ma and Maruyama, 2006).**

    **Response**: As suggested we adjusted the way we cited the authors.

32. **Line 23: I thought the scope of the study was 2014. Why use data from 2012? Have you accounted for the yearly variation and solar activity influence in juxtaposing your 2012 and 2014 data? Please could you clarify this?**

    **Response**:
    Since magnetometer data during the year 2012 was available simultaneously both at dip equator (Addis Ababa)

and off-equator (Adigrat) we used these data to observe the performance of equatorial electric field model over equatorial region of Africa. The model correlates moderately over equatorial region of Africa, and hence we used the model over the region, during the year 2014 (**Page: 7-8** ).

**Results and Discussions**

33. **Line 22: For some of selected. Remove of**

    **Response**: We accept it and corrected in the revised manuscript

34. **Lines 22-23. How did you select those quiet days? How many where there? What is the temporal resolution of both $\Delta H$ and EEF. Is the correlation obtained from "some selected quiet days" be an adequate representation of all other quiet days in year 2012? Check Figure caption in Figure 1 and harmonize. Let's know whether you use some days or quiet days of month of year 2012.**

    **Response**:
    The quiet days used in this study are the international five quiet days of each month of year 2014 obtained from Kyoto website. Only 38 quiet days available. The temporal resolution of $\Delta H$ is 1 min and EEF is 5 min. We corrected the resolution of $\Delta H$ to 5 min resolution. We modified the correlation between the EEF and $\Delta H$ using the quiet days of year 2012. The graph is replotted. Caption of Figure 1 is also modified (**Page 9, Figure 1**).

    **Page 7**

35. **Lines 2-5. You gave us a beautiful description of how $\Delta H$ can be derived between the two magnetometers just for you to come and tell us that the data were not available for year 2014. I think it should have been the other way round.**

    **Response**: The interest of showing the relationship between $\Delta H$ derived between the magnetometers and EEF derived from equatorial electric field model for the year 2014 was described in the revised manuscript.

36. **Line 3. Correct Adegrat to Adigrat.**

    **Response**: As suggested, we have corrected "Adegrat" to "Adigrat".

37. **Line 4-5. To solve this data gap we used the daytime information of equatorial electric field derived from the real-time prompt penetration electric field model as an option. Did you use the real time model of Ionospheric electric fields of the real-time prompt penetration electric field model?**

    **Response**: We have used the real time model of Ionospheric electric fields of the real-time prompt penetration electric field model.

38. **In Page 6 lines 21-23 the authors claimed the relation is for some selected quiet days of months of year 2012 but in Page 7, lines 5-7 they state that the same relation is for year 2012. This may be extremely misleading.**

    **Response**: Yes, it is a good suggestion. To show the relationship between $\Delta H$ and EEF, we update the considered days by considering the international quietest days of months of year 2012.

39. **We need a clear explanation on how you wish to use correlation result for 2012 to support some of your results in 2014. You must be aware of the solar activity influence on vertical drift. It will not be totally accurate to say that the correlation in 2012 will be the same as those in 2014. May be you did that as an indication of something. You need to clarify.**

    **Response**:
    The correlation between the equatorial electric field model and $\Delta H$ in the year 2012 was made to observe the

performance of the model over the equatorial region of Africa, East Africa. We choice the year 2012 just because of availability of magnetometer data at both stations (Addis Ababa and Adigrat). The performance of the model over this sector was moderate (C = 0.6).

**Page 8**

40. **Line 5: Be consistent. Is it Figure or Fig? (check in all texts and harmonize).**

    **Response**: We used Figure/Fig ways in referring figures according to the style of the journal.

41. **Line 7. Replace "but lags' with " but after ''...**

    **Response**: We accept it and corrected in the revised manuscript.

42. **Line 8: The depletions in the gradient. I am not comfortable with the word depletion. Kindly use the reduction in the gradient.**

    **Response**: As suggested we replaced "depletions in the gradient" by "reduction in the gradient" a new in new manuscript.

43. **Line 9: ...maximum positive of the spatial gradient of ... Replace with " the peak of spatial gradient" or "the maximum spatial gradient".**

    **Response**: As suggested we replaced "maximum positive of the spatial gradient" by "the maximum spatial gradient" a new in new manuscript.

44. **Line 11: Change depletion with reduction.**

    **Response**: As suggested, we replaced it.

45. **Lines 16-19: Why over Asab only?**

    **Response**: Graph of Debark also added (**Page 11-12, 14-19, Figure 2-9**).

    **Page 9**

46. **30 March 2014, 10 April 2014, 20 September 2014, 10 October 2014. Where these days selected randomly?**

    **Response**:
    Over equatorial/low-latitude regions of Africa, since a high rate of occurrence of irregularities observed in the equinoctial seasons, and we selected the equinoctial seasons of the year 2014. And, these days are quiet days of the selected months.

47. **The caption in Figure 3 should be self explanatory and should tell us the stations (Asab and Debark) that were used for the ROTI.**

    **Response**: It is a good comment. We modified the caption of Figure 3 (**Page 12, Figure 3**).

    **Page 10**

48. **Line 1: Do you have any reference for this?**

    **Response**: Yes (**Page 11, Line 32**).

49. **Line 1: From the figure, ... Which Figure? Specify.**

    **Response**: We specified the referred figure in the revised manuscript.

50. **Lines 2-3: A convincing and quantitative way to demonstrate inferences in lines 3-4 is by performing correlation between spatial gradient and irregularities.**

    **Response**: The graph was modified and replotted (**Page 18, Figure 8**).

51. **Lines 7-8: An ionosphere gradient of 518 mm/km was discovered, generated by a plasma bubble. Read the statement and rephrase.**

    **Response**: The statement was rephrased in the revised manuscript.

52. **Line 14: (see., Fig. 5). Change to as seen in Figure 5.**

    **Response**: As suggested we have changed "Change" to "as seen".

53. **Line 17: Change "a" by "the"**

    **Response**: We changed "a" by "the" as suggested.

54. **Line 18: Change "indicates" to "shows"**

    **Response**: We have changed the phrase as suggested.

55. **Line 19: ...in section (2)... which section 2? Change to as stated earlier.**

    **Response**: Thank you. We have changed the phrase as suggested.

56. **Line 23: Put 's' to period**

    **Response**: corrected.

57. **Line 23-24: Equation (1) was applied to all days of the year 2014? Including disturbed days? This is where it is important to separate disturbed days from quiet ones. We know that gradients can be significant during geomagnetic storms.**

    **Response**: The correlation between the TEC gradient and ROTI was done only for the quiet days (**Page 12, Figure 3; Page 18, Figure 8**).

58. **Line 28 - 32: Most of the observed features have not been discussed and plausible answers not given to explain them.**

    **Response**: Thank you for your good suggestions. The observed features of results are discussed (**Page 11, Lines 1-35, Page 16, Lines 1-16**).

59. **Line 32: Change depletions to reductions.**

    **Response**: As suggested, we changed "depletion" to "reduction".

60. **Figure 4: a) Diurnal variation of the spatial gradient of TEC over ASAB and DEBK , b)Daily maximum value of the spatial gradient of TEC variation, c) Diurnal variation of ROTIave over ASAB station and d) Daily maximum value variation of ROTIaave over ASAB station in the year 2014. Check this Figure caption and adjust according to your Figures (e) and (f) are missing.**

    **Response**: It is a good comment. The caption of the figures are adjusted and replotted (**Page 16, Figure 6**).

    **Page 11**

61. **Lines 1-2. If you can show it don't say it.**

    **Response**:: Yes, we adjusted the statement .

62. **Lines 10-11. "The trend they show has similarity with" The trend is already a similarity. Adjust the statement.**

    **Response**: The statement is adjusted.

63. **The caption of Figure 4 is misleading. Please check and let it conform with what you have in the texts.**

    **Response**: It is a good comment. The caption of the figure is modified **(Page 16, Figure 6)**.

64. **Why not add a correlation plot between spatial gradient and ROTI over each station? This is a better way of obtaining quantitative information between both quantities.**

    **Response**: Yes, it's a good suggestion. The relationship between spatial gradient and ROTI over each stations were illustrated by showing the correlation between them **(Page 18, Figure 8)**.

    **Page 12**

65. **Line 3: What about Debark? Why is it not presented? Besides, is this Figure for quiet and disturbed periods? How did you segregate the effect of transient disturbances?**

    **Response**:
    Percentage of occurrence of ionospheric irregularities for Debark is plotted **(Page 20, Figure 9)**. The figures are both for quiet and disturbed periods. We didn't segregate the effect of transient disturbances, since it is not the objective of the study. This may be worked in the future.

66. **Line 26: Basu et al., the year is missing.**

    **Response**: We corrected the missed part of the cited reference.

    **Page 13**

67. **Line 6: Change "has not been seen" to something suitable.**

    **Response**: Modified.

    **Page 14**

    **Conclusions**

68. **Lines 2-3: This is inconclusive and cannot feature in this section given the fact the relation between EEF and TEC gradient was investigated for just for 4 days (Figure 2).**

    **Response**: Well, we update Figure 2 of the previous version of the manuscript. Graphs of quiet-monthly mean of each month of year 2014 for both EEF and TEC gradient was plotted to observe features showing the relation between EEF and TEC gradient **(Page 11, Figure 2)**.

    **Acknowledgments**

69. **Line 6: Remove and.**

    **Response**: corrected.

70. **Lines 7-8: We acknowledge http://www.geomag.org/models/PPEFM/RealtimeEF.html for providing the data the Prompt penetration equatorial electric field model. Give proper acknowledgement please.**

    **Response**: Thank you very much. We update the acknowledgement of the manuscript **(Page 21)**.

71. **Line 8: Provide adequate acknowledgement for using the AMBER data (Visit AMBER website for adequate acknowledgement).**

    **Response**: Thank you for your good suggestion. We acknowledged adequately **(Page 21)**.

**Referee #2**

(a) **Figures 2 and 3 needs a bit more clarification. I understand that the blue curves in Figure 3 shows the spatial gradient and the blue curve in Figure 2 shows enhancement in spatial gradient. What does the authors mean by enhancement/depletion in spatial gradient? Please mention how the enhancement/depletion in spatial gradient is calculated in Figure 2. I am just confused if the blue line in Figure 2 represent spatial gradient in TEC or enhancement in spatial gradient in TEC. The two sentences in lines 3-4 on page 8 are confusing. Considering Figures 2 and 3 are the most important figures of this manuscript they must be explained properly.**
    **Response**:
    In both of the graphs (Figures 2 and 3), the blue curves show the diurnal variation in the gradient of TEC. In Figure 2, the authors try to relate the electric field derived from equatorial electric field model and the spatial gradient in TEC. Whereas, Figure 3 was presented to show the trends of the time variation of ROTI

and the gradient in TEC. In both figures, the maximum enhancement/reduction in the gradient of TEC was observed. **(Page 11 and 12, Figures 2 and 3)**

On the computation of the spatial gradient of TEC between the two nearby located stations (ASAB and DEBK), the gradient of TEC may be either positive/negative value. Both the negative and positive values obtained from the differences of TEC between the two stations show the gradient of TEC. It is obvious that the positive/negative gradient in TEC is obtained when the minuend is larger/smaller than the subtrahend. A positive gradient in TEC denotes a higher in TEC/electron density over ASAB relative to DEBK and a negative gradient in TEC indicates a lower in TEC/electron density over ASAB than DEBK, i.e, a higher in TEC/electron density over DEBK than ASAB.

In the manuscript, we have used the term maximum enhancement/reduction in the gradient of TEC (in terms of magnitude) when the nighttime value of gradient of TEC was larger than the daytime value. Since fluctuation in ionospheric plasma density was prevalent during the nighttime hours, we have used the state of the night time gradient of TEC compared to daytime to describe the relationship between the gradient in TEC and the occurrence ionospheric irregularities.

The authors modified the term enhancement/depletion in the spatial gradient of TEC to maximum enhancement/reduction in the gradient of TEC. In the current study, there was no quantitative method that we have used to calculate the maximum enhancement/reduction in the gradient of TEC. We rather described the state of the gradient of TEC (which shows maximum positive/negative in TEC/electron density during nighttime relative to the daytime) by observing the values of gradient of TEC. The gradient in TEC was calculated by taking the difference of TEC between two stations (ASAB and DEBK) and divided by their longitudinal differences. If the gradient of TEC during nighttime is positive/negative values, the state of gradient of TEC show enhancement/reduction in gradient of TEC.

(b) **Page 8 Line # 9-10: The authors state: "The spatial gradient of TEC observed during the day time was relatively small compared to the evening time values for most of the days." If you take a look at Figure 3c, the daytime peak and the post sunset peak of the spatial gradient are almost of same values.**
**Response**:
The statement was modified as, "most of the day time value of spatial gradient in TEC was relatively small in magnitude compared to the night values."

(c) **Also the authors need to explain what do the increase or decrease in the spatial gradient in TEC mean physically. For example, in case of ROTI, it is very straight forward. If you look at Figure 3, the day time ROTI stays around zero. Postsunset, the ROTI values increase showing fluctuations in TEC (hence density). In case of ROTI, a standard value of 0.5 is considered to identify ionospheric irregularities.**
**However, in case of spatial gradient of TEC, you can see daytime fluctuations as well. So what you see postsunset, may not entirely be due to fluctuations in TEC or irregularities. It may have a significant contribution due to zonal plasma drift. How do you eliminate that possibility?**
**Response**:
Thank you for the suggestion. In the manuscript we modified the term increase/decrease in the spatial gradient of TEC to maximum enhancement/reduction in the gradient of TEC.

At a basic level, there are two independent ways of estimating the TEC gradient values using ground based GPS receiver data (Lee et al., 2006). The first method uses a pair of closely-spaced receiver stations. The second method uses a single GPS receiver station to infer the spatial TEC gradient values based on the observed temporal rate of change in TEC. The two methods actually have their own merit and demerits.

In our study, we have used the first method (i.e., station-pair method). This station-pair method gives us an instantaneous estimate of the TEC gradient along a fixed direction determined by the line segment connecting the two stations. It is true that the zonal drift might have an effect on TEC gradient. As the two stations used to get TEC gradient are close to each other, the same irregularity have a chance to be observed at these stations almost at the same time and hence difference of TEC at these stations can eliminate the contribution of zonally drifting irregularity on TEC gradient variation. However, in case of the single station-method we must note that the background ionospheric plasma drift/circulation speed can potentially inflate as well as deflate the estimated TEC gradient values obtained.

(d) **If you look at Figures 3b and 3d, the peaks of the red and blue curve matches, but their values say a different story. The value of ROTI-index in Figure 3b is higher than that in Figure 3d. But the values of the spatial gradient show an opposite trend. The value of spatial gradient in Figure 3b is lower than that in Figure 3d. So in terms of ROTI, the ionospheric scintillation is stringer in Figure 3b than 3d. But in terms of spatial gradient, it looks opposite. How to explain it.**

**Response**:
The values of ROTI in each panel of Figure 3 (a-d), shown as representative cases, was greater than 0.5 TECU/min, a threshold value showing the presence of irregularities. However, they show difference in the strength of irregularities. These difference could be attributed to the difference in the strength of PRE, and other factors. Similarly, the value of the spatial gradient in TEC was different. For example, the value of the spatial gradient in TEC in Figure 3d is higher than that in Figure 3b, while ROTI value in Figure 3b is higher than that in Figure 3d. This might indicate the difference in the cause of the maximum enhancement/reduction in the gradient in TEC and ROTI, for example zonal neutral wind and ExB drift. **(Page 12, Figure 3)**.

(e) **Figure 3(c): It has almost similar +ve and -ve phases around 16-22 hours. What do the negative phase of the spatial gradient mean physically?**

**Response**:
When we apply Equation 1, we might obtain positive and/or negative values in the gradient of TEC, mostly around 16-22 hrs. Both the positive and negative phase indicates the gradient in TEC. These values are the relative TEC/plasma density between the two stations. The negative value in gradient of TEC indicates the higher TEC/plasma density over DEBK relative to ASAB, i.e., a reduction in TEC/plasma density over ASAB, showing a decrease in plasma density over ASAB compared to DEBK.

(f) **The authors here have just shown 4 cases where ionospheric irregularities were present. They also need to show cases when there were no ionospheric irregularities.**

**Response**:
Thank you for the comment. Cases where absent in the occurrence of ionospheric irregularities were presented in the modified manuscript in Figure 4 (e-h). **(Page 14, Figure 4)**

(g) **Figure 4 (a-c): The authors need to modify the color codes of the three figures to make it clear. Right now, the minimum limit of ROTI in Figure 4(a-b) is set at 0.5 which is considered as the onset of ionospheric irregularities. Set it to zero so that a clear picture can be seen.**

**Response**:
It is a good comment. The color codes of Figure 4 (a-c) given in the manuscript were modified. **(Page 16, Figure 6 (left panel))**

(h) **The manuscript do not show any evidence yet, to prove the spatial gradient can be used as an indicator of ionospheric irregularities. It is in a stage where it actually investigates the relationship between spatial gradient of TEC and ionospheric irregularities. So I will suggest the authors suitably change the title of the manuscript. Technical:**

**Response**:
It is good comment. We presented the relationship between the spatial gradient in TEC and the occurrence

of ionospheric irregularities (Figure 7) as suggested. The spatial gradient of TEC and ROTI correlates with correlation coefficients (C = 0.58 for DEBK, C = 0.53 for ASAB). This could lead us to relate the spatial gradient of TEC and the occurrence of ionospheric irregularities. Studies indicate that the gradient of TEC can be computed from a pair of closely-spaced receiver stations such that the two receivers share the same GPS satellite (less than $2^o$). In our case, however, the two stations are separated by $5^o$. The moderate correlation obtained might be attributed to the wider longitudinal separation ($5^o$) between the two stations. The other factor for the moderate correlation between the gradient of TEC and occurrence of ionospheric irregularities might be the way ROTI was computed (since ROTI contains both the spatial and temporal variation in TEC). If we can reduce the contribution of the temporal part of ROTI, we may get a better relationship. **(Page 1-16; Page 17-18: Figure 7 and 8)**

The title of the manuscript is modified as:

**Investigation of the relationship between the spatial gradient of total electron content (TEC) between two nearby stations and the occurrence of ionospheric irregularity**

(i) **Page 8: line 3: Figure 2a and d should be Figure 2 (a-d)**.
**Response**:
Thank you. We modified, as suggested.

**List of all relevant changes made in the manuscript**:

- We modified the title
- We modified the abstract based on the added figures
- We used new terminology, standard deviation of TEC, $\sigma(\Delta TEC/\Delta lon)$, to show the relation between spatial gradient of TEC and occurrence of ionospheric irregularities.
- We reformulated the introduction.
- We rearranged some of the statements in "Data and Analysis Method" section.
- Figures are added.
- Discussions part of the results are modified

**Thank you very much !**

**References**

[revised manuscript text omitted]

---

## Referee Report (RR1)

**Investigation of the relationship between the spatial gradient of total electron content (TEC) between two nearby stations and the occurrence of ionospheric irregularity**
**By**
Teshome Dugassa et al. (2019)

**General comment**

The manuscript has been well improved. The authors have answered all the queries raised satisfactorily. However, in doing so they introduced some minor misconceptions and errors which I believe should be addressed before publication.

In addressing my concern in general comment 2, the authors claimed that that PRE is due to the spatial gradient of electron density near solar terminator and that the TEC gradient would help them to estimate the strength of the zonal electric field. They also make reference to the changes in Page 3, Lines 7 - 15). However, I could not unfortunately see these changes in Lines 7-15 of Page 3 as stated.
I wish to mention that it is not the gradient in TEC that enhances the PRE near solar terminator. The PRE is generated through the interaction of zonal neutral wind in the F-region and the conductivity gradient caused by the terminator as the authors clearly know. Anyway this not a problem given that the corrections have been pointed out in the specific comments.

Also, the authors did not discussed and explain well how the enhancement/reduction in spatial gradient of TEC is related to irregularities during geomagnetic storm. In Figure 5, they presented a good relationship between both parameters but did not really point out any mechanism that could have affected both. The only reason given so far as justification is the direction of IMF Bz.

In my opinion, it is important that the authors explain the practical significance of the standard deviation of the spatial gradient of TEC over 15 min (as used in the methodology section) and why they used it? Is it kind of related to definition of ROTI?

Finally in the conclusion, I find that the statement '*The spatial gradient of TEC/electron density near-solar terminator obtained from two nearby located GNSS receivers method may be an alternative method to estimate the strength of the zonal electric field''* is a little premature and too general. The authors should just mention that a relation was observed between both quantities at some particular period of the day.

**Specific comments**

**Page 2**
Line 8. to the presence of enhanced eastward electric field was thought to control the generation of plasma density irregularities.
Why was thought? It is established that the enhanced electric field is a crucial driver for irregularities generation.
Line 24-25. For accuracy I suggest you use ...'' This inhomogeneity, i.e spatial plasma density/TEC gradient, varies significantly at low-latitude region because of .........''. Remember that magnetic storm over your region of interest have the tendency of suppressing irregularities more than enhancing them.

**Page 3**
Line 2. Remove gradient (repetition)
Line 6. Change attempt to have attempted,
Line 19. Please change .. relating the **latitudinal/longitudinal gradient of TEC/Plasma density...''** **to .... on the relationship between TEC gradient and the occurrence of ionospheric irregularities.**
Lines 21-24. I suggest the authors remove this part. They have already talked about the PRE earlier on in Page 2, line 9. They could even take this part (Page 3, Lines 21-24) to Page 2 (line 9) if they want.
But they have to remove the phrase in Lines 24-25. [Longitudinal gradient of integrated Pederson conductivity that exists across sunset terminator affects the strength of PRE and the generation of ionospheric irregularities (Tsunoda, 1985).]. Eventhough the statement is true it has already been mentioned earlier at the beginning of the introduction. I think it is not needed hereas it could lead to some confusion in the flow of idea since it will be difficult to establish a clear link between that previous statement and **the longitudinal gradient** of electron density in the statement that followed. Remember Pederson conductivity can be estimated using electron density. But for the longitudinal gradient, I wouldn't know. What you need to do here is to emphasize on the fact that there is a lack of instrumentation for the direct investigation of the relationship between electron density gradient and occurrence of ionospheric irregularities over Africa and you have done that already. Just harmonize the statement for fluidity.
Lines 28- 29. [to examine the strength of the zonal electric field]. Closely located GPS can only help you get an insight into the behaviour of electric field especially at some particular period of the day. TEC is not only modulated by electric field, but by wind and solar radiation among others. If you want to examine directly the strength of electric field, you use magnetometers which the authors already did. So adjust that statement. I think what you want to say is that closely located GPS will help study the relation between gradient and electric field at post sunset.

Page 4, Line 20. I don't understand what you mean by ... was computed **for every time**. What is your temporal resolution?

Page 5 , line 26. It provides. What provides? Intermagnet or Amber or both? Adjust the statement.

Page 8, Line 17. Change was to is.
Page 8, Line 20. Change correlate to correlates

Page 9, Line 11. the positive/negative in the spatial gradient of TEC. Remove ''in the''

Page 10, Line 31-31. Kindly removed this statement. ''Sun et al. (2013) examined the relationship between the storm-enhanced plasma density (SED)-associated irregularities (ROTI) and TEC gradients over continental United States (CONUS) during the geomagnetic storms''. This mechanism has nothing to do with the low latitude ionosphere.

Line 4. Change could to can.
Line 5. Remove dynamically

Page 12, Line 2. Change to....observed the steepest TEC gradient .....

Line 2-3. .....change ''when the occurrence of ionospheric irregularity present (left panel) and absent (right panel)'' ''to during the occurrence of irregularities (left panel) and during their absence (right panel)''.
Line 6. Change indicate to indicates
Line 8. Please what do you mean by occurrence of irregularities was present/ absent? Why not just use when irregularities were present /absent?
Line 15 to 16. The density gradient controls the intensity of the PRE. This is not accurate. The PRE is generated through the interaction of zonal neutral wind in the F-region and the conductivity gradient caused by the terminator.
The density gradient affects the R-T instability growth rate thus, the generation of irregularities (Ossakow, 1982; Mendillo et al., 1992 JGR)
Line 22. Change to Figure 5 (a-d) illustrates typical examples of the .........
Line 24. Change are to is

Figure 4. Replace showing by of.
Line 3. Change are to is
Lines 10 to 11. Gradient in TEC is positive when TEC/electron density over ASAB is higher, and is negative when TEC/electron density over DEBK is higher. This is a repetition.

Page 15, Figure 5. Replace showing with of. Representative examples of ........

Page 17, Line 3. Change ''their correlation'' to ''the correlation''.

Page 19, Line 26, change covey to convey.

Page 20, Line 1. Change present to presented
Page 20, Line 4. What is summery? Do you mean summary?
Line 6. Add the year of your correlation.
Line 7. Change was to were
Line 8. Change seasons and months to month and season
Line 11. Change are to were also
Line 16. Change are to was

Page 21, lines 2-3 I think it is a little bit premature to state that ''The spatial gradient of TEC/electron density near-solar terminator obtained from two nearby located GNSS receivers method may be an alternative method to estimate the strength of the zonal electric field''. You could just report that you observed a relation between both and you mention the period when this relation was very obvious.

---

## Referee Report (RR2)

**Investigation of the relationship between the spatial gradient of total electron content (TEC) between two nearby stations and the occurrence of ionospheric irregularity**

By Teshome Dugassa et al

**General comment**

The paper is technically ok and has been significantly improved. I however, believe there are some clarifications that still need to be made.

I draw to attention of the authors to some confusion that may arise in the processing of magnetic data section mainly on the use of ΔH to represent hourly departures of H and the electrojet at the same time.

I have an issue with the way the storm cases have been analyzed in this paper. At first look there is no information on the storm except for the fact that they were moderate. We do not know when they started the main phase and even the intensity of the ring current etc... Also, it will be more appropriate to present the storm response in terms of variation of ROTI and spatial gradient with respect to variations during quiet background condition or before and after the storm or to the least study the storm effect (on irregularities and spatial gradient) with respect to some identified quiet days.

Again, it is evident that the storm will have the same effect on irregularities and spatial gradient given that a relationship between both quantities has been shown previously (section 3.2 down to Figures 4). What one might be interested in seeing is the effect of the storms on the relationship between irregularities and spatial gradient. For example how does the storm affect such relation and how the relation varies during the various storm phases?

I still find that the discussion on storm mechanism is not convincing. The authors discussed Bz polarities (page 14 lines 6-8) and effect without giving any concrete evidence (at least not from their Figures). Also, nothing has been said on the 12 April 2014 event.

Evidence of the storm time electric field using EEF was only discussed for the event of 18 February 2014 (reduction in PRE). Was there no storm effect on the PRE during the other events? If no what modulated irregularities and spatial gradient behavior?

Finally, I do not understand why the authors mixed their estimation of percentage irregularities occurrences for both quiet and disturbed days (''**all days of the year 2014 including both quiet and disturbed days**''). They had earlier presented that storm could enhance or reduce irregularites. Thus, estimating percentage occurrences during both quiet and disturbed days (without segregating the effect of the disturbances) implies a kind of ''pollution'' to the results especially, if several large and long lasting perturbations had occurred during the period of study, 2014.

**Specific comment**

**Title**
In the title I suggest ''ionospheric irregularity'' should be changed to **ionospheric irregularities.**

**Abstract**

Line 3-4: ''Different instruments and techniques have been applied to study the behavior of ionospheric irregularities''. What are those different instrument the authors are talking about? I do not think that these lines are necessary in the abstract.
Line 6. Kindly change irregularity to irregularities.
Line 7. ''derived from GPS-TEC''. Could you please delete it?
Line 8. Change are to were
Line 10. The enhancement in the intensity of $\sigma(\Delta TEC/\Delta lon)$. I do not understand what you mean by enhancement in intensity.
Line 12. Same as in line 10
Line 13-14. ''the relation between the spatial gradient of TEC**/electron density** obtained from two nearby located Global Navigation Satellite System (GNSS) receivers and equatorial electric field (EEF) was observed''. Remove **electron density,** What happened to the observed relation? Where you expecting not to find a relation between both quantities?
Line 15-16 ''The gradient in TEC and ROTIave observed during the evening time period shows similar trends with EEF but after 1-2 hrs.'' This statement is not concise. Tell us the trend as it is I fnd it difficult to understand. Also just let know that they have similar variations (which you must hint us about) with a delay of about 1-2 hours between both.
Line 17. Remove vast.
Line 19. Remove spatial. One seems to be confused with your use of spatial gradient, latitudinal gradient and longitudinal gradient. Be consistent. I think your study is concerned with longitudinal gradient
Line 20. remove computation.

**Introduction**
Line 26. Appleton ionospheric anomaly. why not just Appleton Anomaly????
Line 28, remove the in front of literature.
Line 30, change for to from
Line 26. Add ''at those stations'' at the end of the sentence.
Line 28. a closely located GPS stations. Remove a.
Line 34. Change application to applications

**Data and analysis method**

I appreciate the fact that the authors gave a complete description of the magnetic processing data. However in doing so they did not make it as precise as possible. For example the reader might be confused with the hourly departure of H denoted $\Delta H$ expressed by Eq. (7) and the $\Delta H$ in equation 10. Obviously both do not mean the same things but how do we differentiate? I think

the author should clearly define ΔH after equator 10 and add that this is what will be used in the rest of the paper.
Line 18. epoch time???? Which one suit best epoch or time?
Line 22. why not just say 1 minutes VTEC values for all satellites in view were averaged.
Line 31, change applied to used
Lines 3-4/ Read the statement and see if it is coherent.
Line 5. replace kinds with a suitable word.
Page 7-8
Line 23 and lines 1-2. Isn't this a repetition of lines 26 -28 of Page 5?
Line 5. change were to was. Same with the other 'were' in the next sentence of same line.
Line 12. be consistent with the usage of day time. is it daytime or day time?

**3. Results and discussion**
Line 19. the word reliability is not adequate here. As a matter of fact the relationship between two quantities cannot be use as a measure of the reliability of one of the quantities. Kindly use an appropriate word that describes exactly the idea you wish to pass across.
Line 20 -21. ''In the analysis, we considered the daytime (07:00 - 17:00 LT) value of ΔH and haven't you  said this earlier?
Line 30. Not just any pair of magnetometers please!!!!!!!!
What is the significance of lines 25 to 31 to your results???
Line 23. Remove the ''a''.
Line 27. replace nighttime period (after 18:00 LT) with post sunset period
Lines 22-23. What informed the choice of the storms? Were they selected randomly? What phase of the storm is represented in Figure 5?
Lines 26-28. On the other hand, when the occurrence of ionospheric irregularities is suppressed. (ROTIave < 0.4 TECU/min), the magnitude of  $\sigma(\Delta TEC/\Delta lon)$ shows reduction (for example, 19 February 2014 and 27 August 2014). The irregularities were suppressed with respect to what day? We do not know the behavior of irregularities before or after the storm. It is true there is a reduction of ROTI which might connotes absence of irregularities due to the storm but are the reduction in spatial gradient really significant? We need to now.
Line 28-29. When did the storm start?
Lines 31-32. The enhancement/reduction in the spatial gradient of TEC in the daytime period during geomagnetic storm day appears to show inhibition of ionospheric irregularities. I do not understand this.
Line 33-35. ''In the presence of ionospheric irregularities, the enhancement/reduction in the spatial gradient of TEC observed during post-sunset period during geomagnetic quiet/disturbed conditions was higher than when ionospheric irregularities are suppressed''. From which Figures?

Lines 2-3. As can be seen from Figs. 5, the geomagnetic storm appears to show a similar effect on the spatial gradient of TEC as it has on ionospheric irregularities. Were you expecting the storm effect to be different on both?

Line 6. Change storm to storms

Line 8. ''or local time at which the maximum negative excursion of Dst occurs''. We didn't see that.

Line 9-10. ''When the z-component of interplanetary magnetic field (IMF Bz) turns towards northward (for example, during 19 Feb 2014) in the post-sunset period, reduction in the spatial gradient of TEC''. We didn't see this.

Line 12. What do you mean by occurrence variation?

Lines 9-15. Why mix the percentage irregularities during both quiet and disturbed days?

Lines 12-16. I clearly do not see the importance of these lines.

---

## Author Response (AR2)

**Response to reviewers' comments on the manuscript angeo-2018-131-RC1 and RC2:**
**Investigation of the relationship between the spatial gradient of total electron content (TEC) between two nearby stations and the occurrence of ionospheric irregularity**

Authors: Teshome Dugassa, John Bosco Habarulema, and Melessew Nigussie

**General remarks**

We appreciate the time and guidance of the anonymous reviewers whose suggestions have been very helpful during the correction of this manuscript. The responses to the comments are as follows:

**Referee #1**

**General comments**

1. **The authors did not discussed and explain well how the enhancement/reduction in spatial gradient of TEC is related to irregularities during geomagnetic storm. In Figure 5, they presented a good relationship between both parameters but did not really point out any mechanism that could have affected both. The only reason given so far as justification is the direction of IMF Bz.**

   **Response**:
   How the enhancement/reduction in spatial gradient of TEC is related to irregularities during geomagnetic storm was discussed (**Page 13: Line 29-36 and Page 16: Lines 1-5**).

2. **In my opinion, it is important that the authors explain the practical significance of the standard deviation of the spatial gradient of TEC over 15 min (as used in the methodology section) and why they used it? Is it kind of related to definition of ROTI?**

   **Response**:
   The observed spatial gradient of TEC show positive and negative values. To see the intensity of spatial gradient of TEC, it is important to apply the standard deviation of gradient of TEC. Yes, it is related to definition of ROTI. For the current study, we considered the standard deviation in the gradient of TEC over 15 min. In the future, we may investigate other time intervals over which the standard deviation should be taken so that the gradient of TEC correlate with ROTI. We have explained why we took standard deviation of the spatial gradient of TEC over 15 min (**Page 4: Lines 26-31**).

3. **Finally in the conclusion, I find that the statement** *"The spatial gradient of TEC/electron density near-solar terminator obtained from two nearby located GNSS receivers method may be an alternative method to estimate the strength of the zonal electric field"* **is a little premature and too general. The authors should just mention that a relation was observed between both quantities at some particular period of the day.**

   **Response**:
   As suggested by the reviewer, we modified the statement (**Page 22: Lines 19-20**).

**Listing of technical corrections**
**Specific comments**

1. **Line 8. to the presence of enhanced eastward electric field was thought to control the generation of plasma density irregularities. Why was thought? It is established that the enhanced electric field is a crucial driver for irregularities generation.**

   **Response**:
   "was thought to control" is modified to "is a critical driver which control" (**Page 2: Line 8**).

2. **Line 24-25. For accuracy I suggest you use ... "This inhomogeneity, i.e spatial plasma density/TEC gradient, varies significantly at low-latitude region because of ........". Remember that magnetic storm over your region of interest have the tendency of suppressing irregularities more than enhancing them.**

**Response**:
"is higher at low-latitude region" is modified to "varies significantly at low-latitude region" (**Page 2: Line 25**).

**Page 3**

3. **Line 2. Remove gradient (repetition)**

**Response**:
"gradient" is removed (**Page 3: Line 2**)

4. **Line 6. Change attempt to have attempted,**

**Response**:
"attempt" is changed to "have attempted" (**Page 3: Line 6**)

5. **Please change .. relating the latitudinal/longitudinal gradient of TEC/Plasma density..." to .... on the relationship between TEC gradient and the occurrence of ionospheric irregularities.**

**Response**:
"... relating the latitudinal/longitudinal gradient of TEC/Plasma density" is changed to .... "on the relationship between TEC gradient" (**Page 3: Line 19-20**).

6. **Lines 21-24. I suggest the authors remove this part. They have already talked about the PRE earlier on in Page 2, line 9. They could even take this part (Page 3, Lines 21-24) to Page 2 (line 9) if they want.**

**Response**:
As suggested the lines are removed.

7. **Remove the phrase in Lines 24-25. What you need to do here is to emphasize on the fact that there is a lack of instrumentation for the direct investigation of the relationship between electron density gradient and occurrence of ionospheric irregularities over Africa and you have done that already. Just harmonize the statement for fluidity.**

**Response**:
As suggested we have removed the lines and adjusted the statement (**Page 3: Lines 21-24**).

8. **Lines 28- 29. [to examine the strength of the zonal electric field]. Closely located GPS can only help you get an insight into the behaviour of electric field especially at some particular period of the day. TEC is not only modulated by electric field, but by wind and solar radiation among others. If you want to examine directly the strength of electric field, you use magnetometers which the authors already did. So adjust that statement. I think what you want to say is that closely located GPS will help study the relation between gradient and electric field at post sunset.**

**Response**:
We thank you for the suggestion. We adjusted the statement (**Page 3: Lines 25-29**).

9. **Page 4, Line 20. I don't understand what you mean by ... was computed for every time. What is your temporal resolution?**

**Response**:
TEC data we have used has 1 minute resolution (**Page 4: Line 22**).

The computation of the spatial gradient of VTEC were carried out for epoch of time (1 to 1440 data points). The term "for every time" was used to indicate the epoch of time (**Page 4: Line 18**).

10. **Page 5 , line 26. It provides. What provides? Intermagnet or Amber or both? Adjust the statement.**

**Response**:
"It provides" is changed to "Both of the instruments provide" **(Page 5: Line 28 - Page 6: Line 1)**

11. **Page 8, Line 17. Change was to is**.

**Response**:
"was" is changed to "is" **(Page 8: Line 19)**

12. **Page 8, Line 20. Change correlate to correlates**

**Response**:
"correlate" is changed to "correlates" **(Page 5: Line 21)**

13. **Page 9, Line 11. the positive/negative in the spatial gradient of TEC. Remove "in the"**

**Response**:
"in the" is removed **(Page 9: Line 13)**

14. **Page 10, Line 31-31. Kindly removed this statement. "Sun et al. (2013) examined the relationship between the storm-enhanced plasma density (SED)-associated irregularities (ROTI) and TEC gradients over continental United States (CONUS) during the geomagnetic storms". This mechanism has nothing to do with the low latitude ionosphere.**

**Response**:
We have removed the statement as suggested.

**Page 11**

15. **Line 4. Change could to can.**

**Response**:
"could" is changed to "can" **(Page 11: Line 4)**

16. **Line 5. Remove dynamically**

**Response**:
We than you for the suggestion. "dynamically" is removed as suggested.

17. **Page 12, Line 2. Change to....observed the steepest TEC gradient .....**

**Response**:
"a most steep in" is changed to "the steepest" **(Page 12: Line 2)**

**Page 13**

18. **Line 2-3. .....change "when the occurrence of ionospheric irregularity present (left panel) and absent (right panel)" to "during the occurrence of irregularities (left panel) and during their absence (right panel)".**

**Response**:
"when the occurrence of ionospheric irregularity present (left panel) and absent (right panel)" is changed to "during the occurrence of irregularities (left panel) and during their absence (right panel)" **(Page 13: Lines 2-3)**

19. **Line 6. Change indicate to indicates**

**Response**:
"indicate" changed to "indicates" **(Page 13: Line 6)**

20. **Line 8. Please what do you mean by occurrence of irregularities was present/ absent? Why not just use when irregularities were present /absent?**

    **Response**:
    "when the occurrence of irregularity present (Fig. 4, left panels) than when the occurrence of irregularity of irregularity absent (Fig. 4, right panels)" changed to "when irregularities were present (Fig. 4, left panels), left panels) than when irregularities were absent (Fig. 4, right panels)" **(Page 13: Lines 8-9)**.

21. **Line 15 to 16. The density gradient controls the intensity of the PRE. This is not accurate. The PRE is generated through the interaction of zonal neutral wind in the F-region and the conductivity gradient caused by the terminator. The density gradient affects the R-T instability growth rate thus, the generation of irregularities (Ossakow, 1982; Mendillo et al., 1992 JGR)**

    **Response**:

    Thank you. As suggested we have modified the statement **(Page 13: Lines 15-17)**.

22. **Line 22. Change to Figure 5 (a-d) illustrates typical examples of the ………**

    **Response**:
    "Figure 5 (a-d) illustrates examples" is changed to "Figure 5 (a-d) illustrates typical examples" **(Page 13 Line 22)**.

23. **Line 24. Change are to is**

    **Response**:
    "are" is changed to "is" **(Page 13: Line 24)**

    **Page 14**

24. **Figure 4. Replace showing by of.**

    **Response**:
    In Figure 4,"showing" is changed to "of"

25. **Line 3. Change are to is**

    **Response**:
    "are" is changed to "is" **(Page 16: Line 10)**

26. **Lines 10 to 11. Gradient in TEC is positive when TEC/electron density over ASAB is higher, and is negative when TEC/electron density over DEBK is higher. This is a repetition.**

    **Response**:
    As suggested "Gradient in TEC is positive when TEC/electron density over ASAB is higher, and is negative when TEC/electron density over DEBK is higher" is removed **(Page 17 Lines 3-4)**.

27. **Page 15, Figure 5. Replace showing with of. Representative examples of ……..**

    **Response**:
    In the caption of Figure 5, "Representative examples showing" is changed to "Representative examples of"

28. **Page 17, Line 3. Change "their correlation" to "the correlation".**

    **Response**:
    "their correlation" is changed to "the correlation" **(Page 17: Line 31)**

29. **Page 19, Line 26, change covey to convey.**

    **Response**:
    "covey" is changed to "convey" **(page 21: Line 11)**

30. **Page 20, Line 1. Change present to presented**

    **Response**:
    "present" is changed to "presented" (**Page 22: Line 1**)

31. **Page 20, Line 4. What is summery? Do you mean summary?**

    **Response**:
    "summery" is changed to "summary" (**Page 22: Line 3**)

32. **Line 6. Add the year of your correlation.**

    **Response**:
    "in year 2014" is added (**Page 22: Line 6**)

33. **Line 7. Change was to were**

    **Response**:
    "was" is changed to "were" (**Page 22: Line 7**)

34. **Line 8. Change seasons and months to month and season**

    **Response**:
    "seasons and months" is changed to "month and season" (**Page 22: Line 9**)

35. **Line 11. Change are to were also**

    **Response**:
    "are" changed to "were also" (**Page 22: Line 11**)

36. **Line 16. Change are to was**

    **Response**:
    "are" is changed to "was" (**Page 22: Line 16**)

37. **Page 21, lines 2-3 I think it is a little bit premature to state that "The spatial gradient of TEC/electron density near-solar terminator obtained from two nearby located GNSS receivers method may be an alternative method to estimate the strength of the zonal electric field". You could just report that you observed a relation between both and you mention the period when this relation was very obvious.**

    **Response**:
    As suggested, we adjusted the statement (**Page 22: Lines 19-20**).

**Referee #2**

1. **Figure 4: This figure shows a comparison of spatial gradient in TEC on days with/without irregularities. If you take a closer look at Figures 4b, 4c, 4d, , the spatial gradient ($\sigma$) is around 0.5. These represent days with irregularities. On the other hand, Figures 4f, 4g, 4h represent days without ionospheric irregularities. But the value of the spatial gradient ($\sigma$) is still around 0.5. But ROTI shows clear difference on days with/without irregularities. This brings into question the entire ability of the spatial gradient ($\sigma$) to detect irregularities.**

   **Response**:
   The comments are very constructive.

   **Figure 4** show a comparison of spatial gradient in TEC on days with/without irregularities. Yes, some kind of clarity is needed on the magnitude of $\sigma(\Delta TEC/\Delta lon)$ in the presence/absence of ionospheric irregularities indicated by ROTI. When we investigate the relation between the spatial gradient of TEC and occurrence of ionospheric irregularities, we didn't present the threshold of $\sigma(\Delta TEC/\Delta lon)$ when the occurrence of ionospheric irregularities is present/absent. In the presence of ionospheric irregularities, we have observed cases when the magnitude of $\sigma(\Delta TEC/\Delta lon)$ are small. Investigating the threshold value of $\sigma(\Delta TEC/\Delta lon)$ used as a proxy of irregularities will be considered in the future work.

   To make clarity on the graphs, we have modified **Figure 4** (in the revised manuscript) such that one clearly observe examples comparing the spatial gradient in TEC represented by $\sigma(\Delta TEC/\Delta lon)$ on days with/without irregularities. In these examples, the intensity level of $\sigma(\Delta TEC/\Delta lon)$ is higher on the days with irregularities than without irregularities.

2. **Figures 3 and 4:**
   **Figure 3 shows the variation of spatial gradient for 4 different days a) 30 March 2014, b) 10 April 2014, c) 20 September 2014 and d) 10 October 2014. Figure 4(a-d) also represent the same set of days. However the spatial gradient ($\sigma$) looks entirely different in the 2 figures although they represent the same days. How is that possible?**

   **Response**:
   Thank you for the comments.

   **Figure 3** shows the diurnal variation in the spatial gradient of TEC for 4 different days. Figure 4, on the other hand, illustrate examples of the standard deviation of spatial gradient of TEC $\sigma(\Delta TEC/\Delta lon)$ for the same days of **Figure 3**. Yes, the two figures for the same days are different. These difference is due to the difference in the two parameters ($\Delta TEC/\Delta lon$ and $\sigma(\Delta TEC/\Delta lon)$) we have used.

   In the revised manuscript, to elaborate the difference between the spatial gradient of TEC ($\Delta TEC/\Delta lon$) and its standard deviation ($\sigma(\Delta TEC/\Delta lon)$), we have modified Figure 4 when the relation between $\sigma(\Delta TEC/\Delta lon)$ and ROTI was clearly observed.

   In this study, we have presented Examples on how spatial gradient in TEC and occurrence of irregularities are related (i) in the presence and absence of irregularity in **Figures 3 and 4**, and (ii) during disturbed days (Figure 5). We have also illustrated the relation between them using GPS-TEC data during 2014 (**Figures 3, 4, 5, 7, 8 and 9**). These Figures can lead us to relate spatial gradient in TEC and occurrences of ionospheric irregularities.

**List of all relevant changes made in the manuscript:**

- Page 3 Lines 21-31 are modified.

- Page 13: Lines 29-35, Page 14: Lines 1-11, Page 15: Lines 1-15, and Page 16: Lines 1- 6 are added.

- Figure 4 is modified.

- Figure 6 is added in response to reviewer 1's comments.

[revised manuscript text omitted]

---

## Author Response (AR3)

**Corrections made to manuscript angeo-2018-131:**

**Investigation of the relationship between the spatial gradient of total electron content (TEC)  between two nearby stations and the occurrence of ionospheric irregularities**

**Authors: Teshome Dugassa, John Bosco Habarulema, and Melessew Nigussie**

**General Remarks**

We appreciate the time and guidance of the anonymous reviewers whose suggestions have been very helpful during the correction of this manuscript. The responses to the comments are as follows:

**General comments:**

**Referee #1**

1. I draw to attention of the authors to some confusion that may arise in the processing of magnetic data section mainly on the use of ΔH to represent hourly departures of H and the electrojet at the same time.

   **Response:**

   To make clear the confusion between the hourly departure of H and the electrojet, we have modified ΔH  of Eq.7  to δH **(Page 6, Eq 7, 8, 9)**.

2. I have an issue with the way the storm cases have been analyzed in this paper. At first look there is no information on the storm except for the fact that they were moderate. We do not know when they started the main phase and even the intensity of the ring current etc...

   **Response:**

   Thank you for the comments.

   The information (the intensity of ring current, Dst; their SSC) concerning the selected storms are added **(Page 13, Lines 23-34)**.  We have also added graphs of Dst and IMF Bz **(Page 16, Figure 5).**

   Also, it will be more appropriate to present the storm response in terms of variation of ROTI and spatial gradient with respect to variations during quiet background condition or before and after the storm or to the least study the storm effect (on irregularities and spatial gradient) with respect to some identified quiet days.

   **Response:**

   As suggested, the storm response in terms of variation of ROTI and spatial gradient of TEC with respect to variations during quiet background conditions (quiet monthly mean) are presented **(Page 16, Figure 5)**.

3. Again, it is evident that the storm will have the same effect on irregularities and spatial gradient given that a relationship between both quantities has been shown previously (section 3.2 down to Figures 4). What one might be interested in seeing is the effect of the storms on the relationship between irregularities and spatial gradient. For example how does the storm affect such relation and how the relation varies during the various storm phases?

   **Response:**
   As suggested, we have used quiet monthly mean of ROTI and spatial gradient of TEC as a background condition to observe the storm effects on the relation between irregularities and spatial gradient **(Page 14, Lines 1-9; Page 16, Figure 5)**.

4. I still find that the discussion on storm mechanism is not convincing. The authors discussed Bz polarities (page 14 lines 6-8) and effect without giving any concrete evidence (at least not from their Figures). Also, nothing has been said on the 12 April 2014 event.

   **Response:**
   Thank you for the comments. We have modified discussion on the storm mechanism and their effect on ROTI and spatial gradient **(Page 15, Lines 15-31)**.

5. Evidence of the storm time electric field using EEF was only discussed for the event of 18 February 2014 (reduction in PRE). Was there no storm effect on the PRE during the other events? If no what modulated irregularities and spatial gradient behavior?

   **Response:**
   It can be seen from Fig. 6 (b) and  6 (d), ionospheric irregularities and spatial gradient of TEC is inhibited (triggered) during 12 April 2014 and 12 September 2014 storm main phase, respectively. The storm time electric field effect was not evident from Fig. 6 (b) and  6 (d). This could be due to difference in the intensity of the storm. While the storm on 19 February 2014 is strong (Dst = -120 nT), it is moderate on the other events (-87 nT, -79 nT, and -88 nT) on 12 April, 26 August and 12 September 2014, respectively. In addition, the mechanism can be explained by the local time at which the maximum negative excursion of Dst occurs (Aarons, 1991) **(Page 17, Lines 5-13)**.

6. Finally, I do not understand why the authors mixed their estimation of percentage irregularities occurrences for both quiet and disturbed days (''all days of the year 2014 including both quiet and disturbed days''). They had earlier presented that storm could enhance or reduce irregularities. Thus, estimating percentage occurrences during both quiet and disturbed days (without segregating the effect of the disturbances) implies a kind of ''pollution'' to the results especially, if several large and long lasting perturbations had occurred during the period of study, 2014.

**Response:**

We thank you for the comments and suggestions. We have modified Figure 10. The estimation of percentage occurrence of ionospheric irregularities over the stations was done only for quiet days of the month year 2014. **(Page 21, Lines 5-7, Figure 10).**

**Specific comment**

**Title**

In the title I suggest ''ionospheric irregularity'' should be changed to ionospheric irregularities.

**Response:** In the title ''ionospheric irregularity'' is changed to **ionospheric irregularities**.

**Abstract**

**Line 3-4:** ''Different instruments and techniques have been applied to study the behavior of ionospheric irregularities''. What are those different instrument the authors are talking about? I do not think that these lines are necessary in the abstract.

**Response:**

Thank you for the comments. We have removed the statement, ''Different instruments and techniques have been applied to study the behavior of ionospheric irregularities'' in the abstract **(Lines 3-4)**.

**Line 6.** Kindly change irregularity to irregularities.
**Response:** ''Irregularity'' is changed to ''irregularities'' **(Page 1, Line 5)**.

**Line 7.** ''derived from GPS-TEC''. Could you please delete it?
**Response:** As suggested ''derived from GPS-TEC'' is deleted **(Page 1, Line 7)**.

**Line 8.** Change are to were
**Response:** ''are'' is changed to ''were'' **(Page 1, Line 7).**

**Line 10.** The enhancement in the intensity of ($\Delta$TEC/$\Delta$lon). I do not understand what you mean by enhancement in intensity.
**Response:** Modified to ''The intensity level of ($\Delta$TEC/$\Delta$lon)'' **(Page 1, Line 9).**

**Line 12.** Same as in line 10
**Response:** Modified to ''The observed enhancement of ($\Delta$TEC/$\Delta$lon)'' **(Page 1, Line 9).**

**Line 13-14.** ''the relation between the spatial gradient of TEC/electron density obtained from two nearby located Global Navigation Satellite System (GNSS) receivers and equatorial electric field (EEF) was observed''. Remove electron density, What happened to the observed relation? Where you expecting not to find a relation between both quantities?

**Response:** We have removed "electron density" **(Page 1, Line 12).**

After postsunset period when PRE due to enhancement in zonal electric field is expected to occur, the spatial gradient of TEC also show enhancement **(in Figure 2)**. In our investigation since ROTI and spatial gradient of TEC show good relation, such relation was expected.

**Line 15-16.** "The gradient in TEC and ROTIave observed during the evening time period shows similar trends with EEF but after 1-2 hrs." This statement is not concise. Tell us the trend as it is I find it difficult to understand. Also just let know that they have similar variations (which you must hint us about) with a delay of about 1-2 hours between both.

**Response:**
The statement is modified as, "The gradient in TEC and ROTIave observed during the evening time period have similar variations with a delay of about 1-2 hrs between them. Both of them show similar trends with EEF". **(Page 1, Line 15-16).**

**Line 17.** Remove vast.
**Response:** "vast" is removed as suggested. **(Page 1, Line 16).**

**Line 19.** Remove spatial. One seems to be confused with your use of spatial gradient, latitudinal gradient and longitudinal gradient. Be consistent. I think your study is concerned with longitudinal gradient.
**Response:**
"spatial" is removed as suggested. **(Page 1, Line 18).**

**Line 20.** remove computation.

**Response:** As suggested "computation" was removed. **(Page 1, Line 18).**

**Introduction**
**Page 2**
**Line 26.** Appleton ionospheric anomaly. why not just Appleton Anomaly????

**Response:**
We have changed "Appleton ionospheric anomaly" to "Appleton  anomaly" **(Page 2, Line 26).**

**Line 28,** remove the in front of literature.

**Response:** We have deleted "in the literature" **(Page 2, Line 28).**

**Line 30**. change for to from

**Response:** We have changed "for" to "from" **(Page 2, Line 31).**

**Page 3**
**Line 26.** Add "at those stations" at the end of the sentence.

**Response:**
As suggested we have added "at those stations" at the end of the sentence **(Page 3, Line 26).**

**Line 28.** a closely located GPS stations. Remove a.

**Response:** We have removed "a" **(Page 3, Line 28).**

**Line 34.** Change application to applications
**Response:**
"Application" is changed to "applications" **(Page 3, Line 34).**
**Data and analysis method**

I appreciate the fact that the authors gave a complete description of the magnetic processing data. However in doing so they did not make it as precise as possible. For example the reader might be confused with the hourly departure of H denoted ΔH expressed by Eq. (7) and the ΔH in equation 10. Obviously both do not mean the same things but how do we differentiate? I think the author should clearly define ΔH after equator 10 and add that this is what will be used in the rest of the paper.

**Response:**
Thank you. To differentiate, the hourly departure of H denoted ΔH expressed by Eq. (7) and the ΔH in equation 10, we have modified Eq.(7) to δH. **(Page 6, Eq. 7-10).**

**Page 4**
**Line 18.** epoch time???? Which one suit best epoch or time?
**Response:** We have used time. **(Page 4, Line 18).**

**Line 22**. why not just say 1 minutes VTEC values for all satellites in view were averaged.

**Response:** As suggested we have modified the statement. **(Page 4, Lines 21-22).**

**Line 31,** change applied to used
**Response:** As suggested we have changed "applied" to "used" **(Page 4, Line 30).**

**Page 5**
**Lines 3-4.** Read the statement and see if it is coherent.
**Response:** Thank you. We have modified the statements **(Page 5, Lines 1-3).** .

**Line 5.** replace kinds with a suitable word.
**Response:** We have removed "kinds" **(Page 5, Line 3).**

**Page 7-8**
**Line 23 and lines 1-2.** Isn't this a repetition of lines 26 -28 of Page 5?
**Response:** Thank you. Yes, it is repetition and we have removed them. **(Page 7, Line 19).**

**Page 8**
**Line 5.** change were to was. Same with the other 'were' in the next sentence of same line.
**Response:** We have changed "were" to "was". **(Page 8, Line 2 and 3).**

**Line 12.** be consistent with the usage of day time. is it daytime or day time?
**Response:** We have changed "day time " to "daytime". **(Page 8, Line 9).**

**Results and discussion**
**Page 8**
**Line 19.** the word reliability is not adequate here. As a matter of fact the relationship between two quantities cannot be use as a measure of the reliability of one of the quantities. Kindly use an appropriate word that describes exactly the idea you wish to pass across.

**Response:**
We have modified the statement as to,  "To use the EEF model over the East African sector, it is important to show the performance EEF, by presenting the relationship between ΔH and EEF. Here, we presented the relation between the ΔH and EEF for five (5) international quiet days of the year 2012."  **(Page 8, Line 16).**

**Line 20 -21.** "In the analysis, we considered the daytime (07:00 - 17:00 LT) value of ΔH and haven't you said this earlier?

**Response:** Yes, it is stated earlier, and we have removed the statement. **(Page 8, Line 17).**

**Line 30.** Not just any pair of magnetometers please.

**Response:**
The statement is corrected to, "pair of magnetomer stations one located near magnetic equator and the other at off-equator ". **(Page 8, Line 27).**

What is the significance of lines 25 to 31 to your results???

**Response:**
Lines 25 to 31 show how ΔH and vertical velocity ExB drift are related with EEF. Since we are observing the relationship between ΔH and EEF, these results support the result we observed during study and hence EEFM could be used over low-latitude region of Africa and apply to electric field model to describe the physics of ionosphere in this region.

**Page 10**
**Line 23**. Remove the ''a''.
**Response:** We have removed "a". **(Page 10, Line 19).**

**Line 27.** Replace nighttime period (after 18:00 LT) with post sunset period

**Response:** Replaced as suggested. **(Page 10, Line 23).**

**Page 13**
**Lines 22-23.** What informed the choice of the storms? Were they selected randomly? What phase of the storm is represented in Figure 5?

**Response:**
We thank you for the comments. The days are not selected randomly. During the study period (2014), days with Dst <=-50 nT (http://wdc.kugi.kyoto-u.ac.jp/qddays/index.html) are selected. Based on Dst value, disturbed days on different months are thus selected to see how the gradient in TEC are related to ionospheric phase fluctuation. We have modified the statement by adding information about the history of the selected storm days **(Page 13, Lines 22-34)**.

Figure 5 is also modified by including days before and after the main storm **(Page 16)**.

**Lines 26-28**. On the other hand, when the occurrence of ionospheric irregularities is suppressed (ROTIave < 0.4 TECU/min), the magnitude of (ΔTEC/Δlon) shows reduction (for example, 19 February 2014 and 27 August 2014). The irregularities were suppressed with respect to what day? We do not know the behavior of irregularities before or after the storm. It is true there is a reduction of ROTI which might connotes absence of irregularities due to the storm but are the reduction in spatial gradient really significant? We need to now.

**Response:**
To clarify the storm response interms of variation of ROTI and spatial gradient, we modified **Figure 5** that one can easily observe the suppression and enhancement of ROTI and/or spatial gradient during the disturbed days. Here, the suppression /enhancement in the ionospheric

irregularities and/or gradient in TEC during disturbed days were reported based on the quiet monthly mean of ROTI and gradient of TEC **(Page 16)**.

**Line 28-29.** When did the storm start?
**Response:** The time when the storm started was presented and discussed **(Page 13, Lines 22-34).**

**Lines 31-32**. The enhancement/reduction in the spatial gradient of TEC in the daytime period during geomagnetic storm day appears to show inhibition of ionospheric irregularities. I do not understand this.

**Response:** The statement is removed **(Page 14).**

**Line 33-35.** "In the presence of ionospheric irregularities, the enhancement/reduction in the spatial gradient of TEC observed during post-sunset period during geomagnetic quiet/disturbed conditions was higher than when ionospheric irregularities are suppressed". From which Figures?

**Response:** Figure 4a and Figure 5d. **(Page 14, 16).**

**Page 14**
**Lines 2-3.** As can be seen from Figs. 5, the geomagnetic storm appears to show a similar effect on the spatial gradient of TEC as it has on ionospheric irregularities. Were you expecting the storm effect to be different on both?

**Response:** I do expect.

**Line 6.** Change storm to storms.

**Response:** storm is changed to storms. **(Page 15, Line 12).**

**Line 8.** "or local time at which the maximum negative excursion of Dst occurs". We didn't see that.
**Response:**
We than you for the comment. Effect of storms on the occurrence of ionospheric irregularities was indicated. **(Page 17, Lines 9-13)**.

**Line 9-10.** "When the z-component of interplanetary magnetic field (IMF Bz) turns towards northward (for example, during 19 Feb 2014) in the post-sunset period, reduction in the spatial gradient of TEC". We didn't see this.

**Response:** We can observe in thee modified version of the manuscript in Figure 5 a-d (v) **(Page 16, Figure 5).**

**Page 17**
**Line 12.** What do you mean by occurrence variation?

**Response:** Modified to variation. **(Page 18, Line 19).**

**Page 19**
**Lines 9-15.** Why mix the percentage irregularities during both quiet and disturbed days?

**Response:**
We thank you for the comments and suggestions. We have modified Figure 10. The estimation of percentage occurrence of ionospheric irregularities over the stations was done only for quiet days of the month year 2014. **(Page 21, Lines 5-7, Figure 10).**

**Page 21**
**Lines 12-16.** I clearly do not see the importance of these lines

**Response:** Thank you for the suggestion. We have removed the statements. **(Page 23, Line 4).**

[revised manuscript text omitted]